# Characteristics and sources of nonmethane volatile organic compounds (NMVOCs) and $O_3$–$NO_x$–NMVOC relationships in Zhengzhou, China

**Dong Zhang**[1,3], **Xiao Li**[2,3], **Minghao Yuan**[4], **Yifei Xu**[4], **Qixiang Xu**[2,3], **Fangcheng Su**[2,3], **Shenbo Wang**[2,3], **and Ruiqin Zhang**[2,3]

[1]College of Chemistry, Zhengzhou University, Zhengzhou, 450001, China
[2]School of Ecology and Environment, Zhengzhou University, Zhengzhou, 450001, China
[3]Institute of Environmental Sciences, Zhengzhou University, Zhengzhou, 450001, China
[4]Environmental Protection Monitoring Center Station of Zhengzhou, Zhengzhou, 450007, China

**Correspondence:** Ruiqin Zhang (rqzhang@zzu.edu.cn)

**Abstract.** [TS1]Nonmethane volatile organic compounds (NMVOCs) are important precursors of ozone ($O_3$) formation under sufficient nitrogen oxide conditions. Understanding the characteristics and emission sources of NMVOCs, as well as the relationship between NMVOCs and $O_3$, is of great significance for effective $O_3$ pollution control. In this study, continuous online monitoring of NMVOCs was carried out in Zhengzhou, Henan, from 1–30 June. Furthermore, the study provided recommendations for strategies aimed at reducing $O_3$ formation. During the observation period, the concentration of total NMVOCs (TNMVOCs) varied from 9.9 to 60.3 ppbv, with an average of $22.8 \pm 8.3$ ppbv. The average concentration of TNMVOCs during $O_3$ pollution events was higher than on clean days. Six major sources of NMVOCs were identified using the positive-matrix-factorization model. Vehicular exhausts (28 %), solvent usage (27 %), and industrial production (22 %) were the main sources. We explore the $O_3$–precursors relationship and propose observation-oriented $O_3$ control strategies. The results of the relative incremental reactivity (RIR) and the Empirical Kinetics Modeling Approach show that Zhengzhou was under an anthropogenic volatile organic compound (AVOC)-limited regime. NMVOCs had the largest RIR value, while $NO_x$ exhibited a negative RIR value. It is noteworthy that the sensitivity of $O_3$ formation to biogenic volatile organic compounds (BVOCs) was greater than that to AVOCs. Considering the reduction effect, it is recommended that the ratio of AVOCs to $NO_x$ be maintained at no less than 3 : 1 to effectively reduce $O_3$ formation.

## 1 Introduction

In recent years, ozone ($O_3$) pollution has become increasingly prominent in China, especially in urban areas (B. Liu et al., 2023; Zhao et al., 2021; Yan et al., 2023; Sicard et al., 2020). $O_3$ pollution is an important factor affecting ambient air quality (Zhang et al., 2023). Nonmethane volatile organic compounds (NMVOCs) are an important precursor of $O_3$ and secondary organic aerosols; they widely exist in the atmospheric environment and participate in many photochemical reactions, significantly affecting atmo-

spheric oxidation capacity and air quality (Zhu et al., 2021). Some NMVOCs are also air toxics (Billionnet et al., 2011), e.g., benzene, trichloroethylene, and chloroform (Lerner et al., 2012). Long-term exposure to higher concentrations of NMVOCs can lead to acute or chronic health risks (He et al., 2015). Therefore, it is necessary to continue carrying out NMVOC-monitoring activities in $O_3$ pollution areas to analyze the concentrations levels and sources of NMVOCs, as well as their effects on $O_3$ generation.

The concentration of NMVOCs is affected by background concentration levels, weather conditions (Mo et al., 2015),

emission sources, terrain conditions (Liu et al., 2016), and the extent of pollutant transport (Shao et al., 2009). In addition, under meteorological conditions with higher temperatures, NMVOCs exhibit photochemical losses during dispersion and regional transport (Zou et al., 2023; B. Liu et al., 2023; Liu et al., 2020). As a result, ambient NMVOC concentrations vary with locality and season. For example, in typical coastal areas of Ningbo, seasonal variation in NMVOC concentrations is of the following order: winter, spring, fall, and summer (Huang et al., 2023). The coastal areas of Shandong exhibit the highest values in winter ($28.5 \pm 15.1$ ppbv) and the lowest value in fall ($14.5 \pm 7.6$ ppbv) (Huang et al., 2023). The average summer concentration of total NMVOCs (TNMVOCs) in the suburbs of Jinan ($12.0 \pm 5.1$ ppbv) (Z. Liu et al., 2023b) is lower than that in the suburbs of Beijing ($18.3 \pm 8.9$ ppb) and much lower than that in the central city of Beijing ($44.0 \pm 28.9$ ppbv) (Wu et al., 2023a). The average TNMVOC concentration (21.7 ppbv) during the $O_3$ pollution period in Tianjin is 12 % higher than during the period with no $O_3$ pollution (B. Liu et al., 2023).

NMVOCs are emitted from various sources, including both anthropogenic sources and biogenic sources (Chameides et al., 1992), as well as through secondary generation via photochemical reactions (Yuan et al., 2012). The main sources of NMVOCs include motor vehicle emissions, industrial processes, solvent usage, fuel evaporation, combustion, and biogenic emissions (Wu et al., 2016; Prendez et al., 2013; Watson et al., 2001). Biogenic emissions are mainly affected by temperature and radiation conditions (Li et al., 2020). Biogenic emissions are therefore higher during hotter months, especially in summer (Pacifico et al., 2009; Xu et al., 2023). Urban areas are greatly affected by anthropogenic sources (Zhang et al., 2023; Goldstein and Galbally, 2007). In different regions, the main contribution sources of NMVOCs vary. For example, the main sources of anthropogenic volatile organic compounds (AVOCs) in the Yangtze River Delta region of China are vehicle emissions and solvent evaporation (Xu et al., 2023). The Pearl River Delta region is mainly affected by solvent use, the use of liquefied petroleum gas, and vehicle exhaust emissions. Atmospheric NMVOCs in Beijing are greatly influenced by motor vehicle emission sources and combustion sources (Liu et al., 2021; Zhang et al., 2020). Huang et al. (2023) reported that plastic synthesis, industrial processes, organic solvents, dyeing, traffic emissions, and pesticides have been identified as the main sources of NMVOCs in the city of Ningbo, a coastal area (Z. Liu et al., 2023a). Since different emission sources contribute differently to NMVOCs and thus have different impacts on the generation of $O_3$ (Zhang et al., 2023), it is necessary to investigate the sources of NMVOCs in different cities.

Designing a reasonable and effective precursor emission control strategy is crucial for controlling the photochemical generation of $O_3$ (Yang et al., 2021). The relationship between $O_3$ and its precursors is nonlinear (Chameides et al., 1992), and precursor emission reduction strategies need to be dynamically adjusted based on the actual sensitivity of $O_3$ formation (Chu et al., 2023; Lin et al., 2005). The observation-based model (OBM) is a widely used tool for analyzing $O_3$–$NO_x$–NMVOC sensitivity (Zhang et al., 2008; Nelson et al., 2021; Cardelino and Chameides, 1995). Several studies in China have analyzed the sensitivity of $O_3$ to precursors and control scenarios. For example, $O_3$ in the central area of the Yangtze River Delta is under an NMVOC-limited regime, and AVOCs play a leading role in the formation of $O_3$ (Z. Liu et al., 2023a). Chengdu is in a typical NMVOC-restricted area, meaning NMVOC emission reduction helps to prevent and control $O_3$ pollution. The emission reduction scenario based on NMVOC sources showed that the emission reduction ratio of NMVOCs to $NO_2$ needs to exceed $3 : 1$ to prevent $O_3$ pollution (Chen et al., 2022b). Xie et al. (2021) found that controlling NMVOCs in Leshan, a non-provincial capital city in southwestern China, can effectively reduce the photochemical generation of $O_3$; they also pointed out that the best emission reduction strategy for NMVOCs and $NO_x$ involves a ratio of $3 : 1$. In addition, the generation of $O_3$ in areas like Shanghai (Lu et al., 2023), Rizhao (Zhang et al., 2023), and Nanjing (Mozaffar et al., 2021) is generally limited by NMVOCs. However, in the United States and European countries, $O_3$ formation has gradually transitioned from an NMVOC-limited regime to an $NO_x$-limited regime (Nopmongcol et al., 2012; Ring et al., 2018; Goldberg et al., 2016).

Zhengzhou is the capital city of the province of Henan and an important transportation hub in China. High population density levels, extensive vehicle ownership (MPS, 2022), and complex industrial structures determine the complexity of NMVOC emission sources. In recent years, $O_3$ pollution in Zhengzhou has intensified, making it one of the cities with the highest $O_3$ pollution levels in central China (M. Wang et al., 2023; Min et al., 2022). In each year from 2020 to 2022, the annual 90th percentile of the mean maximum daily 8 h average of $O_3$, published by the Zhengzhou Ecological Environment Bureau, was 182, 177, and 178 $\mu g\,m^{-3}$, respectively – i.e., 10 % to 13 % higher than Grade II of the National Ambient Air Quality Standards (160 $\mu g\,m^{-3}$) (https://sthjj.zhengzhou.gov.cn/, last access: June 2023). Some studies have analyzed the concentration levels, sources, and impacts of NMVOCs on $O_3$ in Zhengzhou (Zeng et al., 2023; M. Wang et al., 2023; Min et al., 2022). Wang et al. (2022) analyzed the sensitivity of $O_3$ to precursors and found that in July, with low $O_3$ levels in Zhengzhou, $O_3$ formation is under an NMVOC-limited regime, while on days with $O_3$ pollution accumulation and persistence, $O_3$ formation is in a transitional state. Yu et al. (2021) showed that Zhengzhou is under an NMVOC-sensitive regime in September. These studies show that it is important to study emission reduction of precursors to control $O_3$ generation. However, there is still a lack of relevant research with respect to June, the month with the highest $O_3$ pollution levels in Zhengzhou. In

order to effectively address the increasingly serious trend of $O_3$ pollution in Zhengzhou, it is necessary to prioritize and strengthen research in the Zhengzhou area, especially during periods of high $O_3$ pollution. Therefore, it is necessary to continue to pay attention to the pollution levels of $O_3$ and precursors in Zhengzhou and further explore the relationship between them.

In this study, we conducted online measurements of NMVOCs in June, when $O_3$ pollution is severe in Zhengzhou. The concentrations of, composition of, and diurnal variation in NMVOCs in the atmosphere were analyzed. The main sources of NMVOCs were discussed using the ratio method and the positive-matrix-factorization (PMF) model. The OBM was used to analyze the sensitivity of $O_3$–$NO_x$–NMVOC relationships, and consequently, we proposed an emission reduction strategy for precursors to control $O_3$ concentrations. This study establishes a collaborative control strategy for atmospheric NMVOCs, which is of great significance for managing atmospheric $O_3$ pollution in Zhengzhou.

## 2  Materials and methods

### 2.1  Sampling site

The monitoring site is located on the roof (about 20 m above ground) of the building at the Environmental Protection Monitoring Center Station of Zhengzhou (34.75° N, 113.60° E) (Fig. S1 in the Supplement). The sampling site is a typical urban site that is surrounded by residential areas, commercial areas, and office buildings. There are no point sources of air pollution within a radius of 1 km. The sampling site may be affected by motor vehicle and plant emissions.

### 2.2  Sample collection and chemical analysis

The sampling campaign was conducted from 1–30 June 2023. NMVOC concentrations were observed using a gas chromatography–mass spectrometer (GC–MS; TH-PKU 300B, Wuhan Tianhong Instrument Co. Ltd, China), which employs detection technology with an ultralow-temperature preconcentration combined with a GC–MS/flame ionization detector (FID). The time resolution of the instrument is 1 h, and the flow rate is $60\,\mathrm{mL\,min^{-1}}$. The air samples were collected for the first 5 min of each hour and then preconcentrated through a cold trap to remove $H_2O_2$ and $CO_2$. The samples were captured using an empty capillary column. After preconcentration, the samples were desorbed by rapid heating and introduced into an analytical system. After separation via a chromatographic column, the samples were detected using the FID (for $C_2$–$C_5$ hydrocarbons) and the MS (for $C_5$–$C_{12}$ hydrocarbons, halocarbons, and oxygenated volatile organic compounds (OVOCs)). The correlation coefficient of the standard curve for the target compound was greater than or equal to 0.99, and the detection limit of the instrument method was less than or equal to $0.1\,\mathrm{nmol\,mol^{-1}}$. A

total of 115 NMVOCs were monitored, including 29 alkanes, 11 alkenes, 1 alkyne, 17 aromatic hydrocarbons, 35 halogenated hydrocarbons, 21 OVOCs, and 1 sulfide (carbon disulfide). Details of the device can be found in our previous study (Zhang et al., 2021). The individual NMVOC concentrations measured during the observation period are shown in Table S1 in the Supplement. Additionally, the study conducted simultaneous online measurements of hourly concentrations of particulate matter ($PM_{2.5}$ and $PM_{10}$), other trace gases (CO, $O_3$, NO, and $SO_2$), and meteorological data (temperature ($T$), relative humidity (RH), atmospheric pressure, wind speed (WS), and wind direction (WD)).

### 2.3  PMF model

The PMF 5.0 model is an advanced multivariate factor analysis tool (USEPA, 2014) that can be used to identify the sources of NMVOCs (Norris et al., 2014). The PMF model is expressed as follows:

$$X_{ij} = \sum_{k=1}^{p} g_{ik} f_{kj} + e_{ij}, \tag{1}$$

where, $i$, $j$, and $k$ represent the $i$th sample, the $j$th chemical species, and the $k$th factor, respectively. $X$ represents the concentrations of the chemical species measured in the sample, $g$ is the species contribution, $f$ is the species fraction, and $e$ is the residual matrix TS2 TS3.

The number of factors is obtained by minimizing the objective residual function $Q$ as follows:

$$Q = \sum_{i=1}^{n} \sum_{j=1}^{m} \left[ \frac{X_{ij} - \sum_{k=1}^{p} g_{ik} f_{kj}}{u_{ij}} \right]^2, \tag{2}$$

where $\mu^{ij}$ is the sample data uncertainty.

The sample data uncertainty (Unc) is calculated using Eqs. (3) and (4). If the data concentration is less than the method detection limit (MDL), Eq. (3) is used. Otherwise, Eq. (4) is used. The equations are as follows:

$$\mathrm{Unc} = \frac{5}{6} \times \mathrm{MDL}, \tag{3}$$

$$\mathrm{Unc} = \sqrt{(\mathrm{Error\ Fraction} \times \mathrm{concentration})^2 + (0.5 \times \mathrm{MDL})^2}, \tag{4}$$

where "Error Fraction" represents the precision (%) of each species.

Species with a high proportion of missing samples or with concentration values more than 25 % below the MDLs were excluded, while NMVOCs serving as typical tracers of emission sources were included (USEPA, 2014). NMVOCs with short atmospheric lifetimes were also excluded (Callén et al., 2014; Guo et al., 2011). In this study, 29 out of 115 NMVOCs collected over the sampling period were analyzed

by the PMF model. In this study, a seven-factor solution ($Q_{\text{true}}/Q_{\text{theoretical}} = 3.42$; Fpeak $= 0$, where Fpeak refers to a parameter used to control the rotation of factor-loading matrices in factor analysis CE1) was chosen (Fig. S2 in the Supplement).

## 2.4 Conditional-probability-function analysis

The conditional probability function (CPF) is a source identification tool that can be used to identify local emission sources of pollutants (Uria-Tellaetxe and Carslaw, 2014). CPF analysis methods were employed to determine the potential direction of emission sources by utilizing wind directions and source contributions calculated through the PMF model (Kim and Hopke, 2004). The CPF is defined as

$$\text{CPF} = \frac{m_{\Delta\Theta}}{n_{\Delta\Theta}}, \tag{5}$$

where $m_{\Delta\theta}$ represents the frequency of occurrences from the wind sector $\Delta\theta$ for the top 75 % of the contributions from each identified NMVOC source, while $n_{\Delta\theta}$ represents the total occurrences from the same wind sector. The CPF analyses were conducted using the "openair" package (Carslaw and Ropkins, 2012) in the statistical software R (R Foundation for Statistical Computing, Vienna, Austria).

## 2.5 OBM

An OBM configured with the Master Chemical Mechanism (MCM v3.3.1; https://mcm.york.ac.uk/MCM/, last access: May 2024) was employed to estimate the effect of changes in precursors on $O_3$ (Liu et al., 2022). Detailed information about the OBM can be viewed in previous studies (Chu et al., 2023; Ling et al., 2011). Briefly, the OBM is a zero-dimensional model that assumes a well-mixed atmosphere and, combined with atmospheric chemical mechanisms, simulates the $O_3$ production rate and the corresponding $O_3$ concentration at a given time (Kleinman, 2000).

The OBM used in this study iteratively solves a set of ordinary differential equations (ODEs) that describe the evolution of species concentrations over time. For species with observed concentrations (normally consisting of primary NMVOCs and $NO_x$), horizontal convection and emissions are normally significant. In a zero-dimensional model, these processes are incorporated into $R_{\text{other}}$. Within each iteration, $R_{\text{other}}$ is determined using Eq. (6):

$$R_{\text{other}} = \left(\frac{\partial C_i}{\partial t}\right)_{\text{obs}} - \left[P_i - L_i C_i - \frac{1}{H}v_d C_i \right.$$
$$\left. - \frac{1}{H}\frac{dH}{dt}(C_i - C_{i,\text{bg}}) + R_{\text{aero},i} + R_{\text{aq},i}\right], \tag{6}$$

where $P_i$ and $L_i C_i$ represent the total production rate and total loss rate, respectively. Moreover, $\frac{1}{H}v_d C_i$ represents the sum of the mixing and deposition rates; $\frac{1}{H}\frac{dH}{dt}(C_i - C_{i,\text{bg}})$

represents the mass exchange rate with the background atmosphere; $R_{\text{aero},i}$ and $R_{\text{aq},i}$ are the rates of aerosol and aqueous processes, respectively; and $\left(\frac{\partial C_i}{\partial t}\right)_{\text{obs}}$ denotes the real rate of change in concentration, which is interpolated from hourly observed data points.

With the value of $R_{\text{other}}$ explicitly determined from Eq. (6), the concentrations of all species are predicted by integrating the governing Eq. (7), which is expressed as

$$\frac{\partial C_i}{\partial t} = P_i - L_i C_i - \frac{1}{H}v_d C_i - \frac{1}{H}\frac{dH}{dt}(C_i - C_{i,\text{bg}})$$
$$+ R_{\text{aero},i} + R_{\text{aq},i} + R_{\text{other}}. \tag{7}$$

New iterations start with updated $R_{\text{other}}$ values based on the concentrations predicted in the previous step, continuing until a converged solution is obtained.

In this model, the net production rate of $O_3$ ($P(O_3)$) is determined as the difference between $O_3$ production (the oxidation of NO by $HO_2$ and $RO_2$) and $O_3$ destruction (i.e., $O_3$ photolysis, reactions of $O_3$ with OH and $HO_2$, reactions of OH with $NO_2$, and reactions of $O_3$ with alkenes). This method for estimating $O_3$ production and removal rates has been utilized in several previous studies (Wang et al., 2017, 2022). The constant ($k$) serves as the rate coefficient in the following reactions:

$$P(O_3) = k_{HO_2+NO}[HO_2][NO] + \sum k_{RO_{2i}+NO}[RO_{2i}][NO]$$
$$- k_{HO_2+O_3}[HO_2][O_3] - k_{OH+O_3}[OH][O_3]$$
$$- k_{O(^1D)+H_2O}[O(^1D)][H_2O] \tag{8}$$
$$- k_{OH+NO_2}[OH][NO_2]$$
$$- k_{\text{alkenes}+O_3}[\text{alkenes}][O_3].$$

The relative incremental reactivity (RIR) was computed through the OBM to evaluate the sensitivity of the photochemical production of $O_3$ to changes in the concentration of individual precursors within a given region (Ling et al., 2013; Cardelino and Chameides, 2000). It is calculated as follows:

$$\text{RIR}(X) = \frac{[P_{O_3}(X) - P_{O_3}(X - \Delta X)]/P_{O_3}(X)}{\Delta S(X)/S(X)}, \tag{9}$$

where $X$ is a specific precursor of $O_3$, $P_{O_3}(X)$ and $P_{O_3}(X - \Delta X)$ denote the net production of $O_3$ simulated by the OBM, and $\Delta S(X)/S(X)$ represents the change in the concentration of $S(X)$. Large changes in primary pollutants ($> 20$ %) deviate greatly from the base scenario and are not representative of the current situation. Therefore, the concentration changes in $\Delta S(X)/S(X)$ were assumed to be 20 %. In this study, the values of $S$ for NMVOCs and $NO_x$ were reduced by 0 %–100 %. The relative change in $P_{O_3}(X)$ with $S(\text{NMVOCs})$ and $S(NO_x)$ can be expressed as an isogram of $P_{O_3}(X)$.

The concentrations of trace gases ($SO_2$, $O_3$, CO, and NO) and meteorological parameters ($T$, RH, and WS) with a 1 h time resolution were used as constraints in this model. At

the same time, the concentrations of 75 NMVOCs observed using a 1 h time resolution were selected for input into the model as these 75 NMVOCs were included in MCM v3.3.1. The values for the photolysis frequency ($J(H_2O_2)$, $J(NO_2)$) and the planetary boundary layer are set to their default values. The setup and parameters of the OBM are summarized in Table S2 in the Supplement.

To evaluate the performance of this model, the index of agreement (IOA) was used in this study (Huang et al., 2005):

$$\text{IOA} = 1 - \frac{\sum_{i=1}^{n}(O_i - M_i)^2}{\sum_{i=1}^{n}(|O_i - \overline{O}| + |M_i - \overline{O}|)^2}, \qquad (10)$$

where $O_i$, $M_i$, and $\overline{O}$ represent the hourly values of the observation, the simulation, and the average of the observations, respectively. In various studies, model simulation results are often considered acceptable when the IOA value falls within the range of 0.68 to 0.89 (Wang et al., 2018). To evaluate the reliability of our model simulations, we conducted an analysis of $O_3$ concentration in the atmosphere and calculated the IOA values. Our model does not directly incorporate $O_3$ observations. Instead, it utilizes concentrations of trace gases ($SO_2$, CO, and NO) and 75 NMVOCs, as well as meteorological parameters ($T$, RH, and WS), to simulate the concentration of $O_3$ in the atmospheric environment. The IOA values for $O_3$ were calculated from 07:00 to 19:00 LT during the day a result of 0.8 was obtained. Therefore, the results simulated by our model are reliable.

## 3 Results and discussions

### 3.1 General characteristics

#### 3.1.1 NMVOC concentrations and composition

According to the National Ambient Air Quality Standards (NAAQS) of China from 2012 (Ministry of Environmental Protection of China, 2012), the Grade-II threshold for the maximum daily 8 h average (MDA8) of $O_3$ was 160 µg m$^{-3}$ ($\sim$ 75 ppbv). Two $O_3$ pollution events exceeding 160 µg m$^{-3}$ were identified: Case 1 (8–17 June) and Case 2 (20–27 June). Additionally, there were $O_3$ pollution events on 6 June and from 29–30 June. However, for better data coverage, this study focuses on periods of $O_3$ pollution lasting at least a week, and processes with fewer days of pollution are not discussed in this study. The remaining observation periods were classified as clean days. Figure 1 shows time series illustrating the concentration of TNMVOCs, the 8 h moving average of $O_3$, $SO_2$, $PM_{2.5}$, $NO_x$, CO, and meteorological parameters (WD, WS, $T$, and RH) from 1–30 June 2023. The gray areas in Fig. 1 represent $O_3$ pollution events, and the white areas represent clean days. During the observation period, there were 21 d of $O_3$ pollution, accounting for 70 % of the period.

During the observation period, the average wind speed ($1.3 \pm 0.9$ m s$^{-1}$) was relatively low, which was not con-

ducive to dispersion. The mean RH ($52\% \pm 19\%$) was low, and the mean temperature ($28.9 \pm 4.6\,°C$) was high. The meteorological conditions of high temperature and low RH are conducive to the occurrence of photochemical pollution. The maximum daily 8 h moving average (MDA8) of $O_3$ reached 229 µg m$^{-3}$. The hourly average concentrations of $SO_2$, $NO_2$, CO, and $PM_{2.5}$ were $4.4 \pm 3.3$ µg m$^{-3}$, $26.5 \pm 17.9$ µg m$^{-3}$, $0.6 \pm 0.2$ mg m$^{-3}$, $59.6 \pm 26.5$ µg m$^{-3}$, and $22.9 \pm 7.1$ µg m$^{-3}$, respectively. The concentrations of these pollutants were 97 %, 87 %, 94 %, and 35 % lower than the Grade-I threshold of the 2012 NAAQS, respectively. The average concentration of the TNMVOCs was $22.8 \pm 8.3$ ppbv.

During the Case-1 process, $O_3$ pollution lasted for 10 d. The average RH and temperature were $41\% \pm 16\%$ and $29.9 \pm 4.1\,°C$, respectively, and the average WS was $1.3 \pm 0.8$ m s$^{-1}$. The concentration of MDA8 $O_3$ reached a maximum of 228 µg m$^{-3}$ (11 June) during the pollution period, exceeding the Grade-II threshold of MDA8 $O_3$. In Case 1, the mean concentrations of $SO_2$, $NO_2$, CO, $PM_{10}$, and $PM_{2.5}$ were $6.1 \pm 4.1$ µg m$^{-3}$, $27.4 \pm 19.5$ µg m$^{-3}$, $0.6 \pm 0.1$ mg m$^{-3}$, $69.1 \pm 31.5$ µg m$^{-3}$, and $25.6 \pm 6.8$ µg m$^{-3}$, respectively. The average concentration of TNMVOCs during this process was $24.1 \pm 8.9$ ppbv. In Case 2, $O_3$ pollution occurred continuously for 8 d. The average RH and average temperature were $50\% \pm 14\%$ and $31.2 \pm 2.9\,°C$, respectively. The average concentrations of TNMVOCs ($22.5 \pm 7.4$ ppbv), $SO_2$ ($2.7 \pm 2.1$ mg m$^{-3}$), $NO_2$ ($24.9 \pm 12.3$ mg m$^{-3}$), CO ($0.6 \pm 0.1$ mg m$^{-3}$), $PM_{10}$ ($61 \pm 19$ mg m$^{-3}$), and $PM_{2.5}$ ($24 \pm 7$ mg m$^{-3}$) in Case 2 were all lower than those in Case 1.

The average concentrations of TNMVOCs, $NO_2$, $PM_{10}$, and $PM_{2.5}$ on clean days were lower than those during the $O_3$ pollution events. The average RH ($65\% \pm 17\%$) on clean days was higher than during the Case-1 and Case-2 events, while the average temperature ($26.0 \pm 4.8\,°C$) was lower than during the Case-1 and Case-2 events. According to the analysis in Fig. S3a and b in the Supplement, $O_3$ concentrations show a significant correlation with temperature and RH, with correlation coefficients of 0.7 and $-0.61$, respectively. Therefore, conditions of high temperatures and low RH are more conducive to $O_3$ pollution. Figure S3c indicates that $O_3$ concentrations exceeding the secondary standard mainly occur under meteorological conditions of high temperatures (greater than 30 °C) and low RH (less than 55 %). It can be noted that when $35 < T < 40\,°C$ and $20\% < \text{RH} < 40\%$, $O_3$ concentrations consistently exceed the Grade-II threshold of the 2012 NAAQS. High temperatures and low RH are more conducive to $O_3$ pollution (Chen et al., 2020; Zhang et al., 2015). Meng et al. (2023) argue that most reactions involved in $O_3$ formation accelerate with temperature and that the rate of $O_3$ production exceeds that of $O_3$ loss by a large margin. Therefore, during the study period, the meteorological conditions of high temperature and low RH were also important factors affecting the occurrence of $O_3$ pollution.

Additionally, the average concentration of $NO_2$ on clean days ($24.4 \pm 16.1$ ppbv) was lower than that in Case 1 and Case 2, while the average concentration of NO on clean days ($4.8 \pm 5.5$ ppbv) was higher than during Case 1 (3.9 ± 3.75 ppbv) and Case 2 (3.9 ± 2.4 ppbv). A higher concentration of $NO_2$ can promote the formation of $O_3$, whereas the titration reaction between NO and $O_3$ consumes $O_3$ (Sillman, 1999). Therefore, the higher concentration of $NO_2$ and the lower concentration of NO during pollution events are key factors contributing to the occurrence of $O_3$ pollution events.

The means and standard deviations of NMVOC groups during different processes are listed in Table 1. During the entire period, the concentration of TNMVOCs varied from 10 to 60 ppbv, with an average concentration of $23.0 \pm 8.0$ ppbv. Similar levels of NMVOC concentrations were observed for Case 1 ($24.0 \pm 9.0$ ppbv) and Case 2 ($23.0 \pm 7.0$ ppbv). The TNMVOC concentrations on clean days were relatively low ($21 \pm 7.2$ ppbv). Furthermore, nearly all NMVOC groups were more abundant during $O_3$ pollution events than on clean days.

Throughout the entire sampling period, alkanes ($10.0 \pm 4.4$ ppbv), OVOCs ($4.5 \pm 1.3$ ppbv), and halocarbons ($4.3 \pm 1.9$ ppbv) were the most abundant NMVOC groups, accounting for 44 %, 20 %, and 19 % of the TNMVOCs, respectively. These groups were followed by alkenes (9 %), aromatics (5 %), alkenes (5 %), OVOCs (7 %), alkynes (7 %), and sulfides (1 %). During the two $O_3$ pollution events, alkanes, being the most abundant NMVOC group, contributed 41 % (Case 1) and 43 % (Case 2) to the TNMVOCs. Alkanes were the most abundant NMVOC group during the observation period, partly due to the presence of alkane emission sources around the observation site (e.g., civilian combustion and motor vehicle emissions) and the low photochemical reactivity of alkanes (Mozaffar et al., 2020). Even on clean days, alkanes ($9.6 \pm 3.9$ ppbv) were still the most abundant group (46 %), and halocarbons (19 %) and OVOCs (19 %) were the two other major groups.

Figure 2 illustrates the 15 NMVOCs with the highest average mixing ratios during the two $O_3$ pollution events and on clean days. Ethane, propane, $n$-butane, isopentane, isobutane, $n$-hexane, and $n$-pentane were the most abundant alkanes throughout the entire observation period. Ethane is a major component of natural gas (NG) (Thijsse et al., 1999), while propane, $n$-butane, and isobutane are important tracers of liquefied petroleum gas (LPG) (Tsai et al., 2006; An et al., 2014). Moreover, $n$-hexane mainly originates from solvent emissions. Ethylene, propylene, and isoprene were the most abundant alkenes. Ethylene and propylene mainly originate from biomass burning (Andreae and Merlet, 2001). Isoprene mainly comes from plants (Brown et al., 2007). There was also a high level of acetylene, which is a tracer of incomplete combustion (Blake and Rowland, 1995). Benzene and toluene were the most abundant aromatics; they mainly originate from solvent emissions, vehicular exhausts,

and industry processes (Seila et al., 2001; Mo et al., 2015). Dichloromethane, an important species in solvent usage, was the most abundant halohydrocarbon species (Huang et al., 2014). Acetone was the most abundant OVOC species; it has complex atmospheric sources and is mainly attributed to vehicular emissions and secondary formation (Guo et al., 2013; Watson et al., 2001). The concentration of acetone in the two pollution processes was significantly higher than on clean days, as reported in other studies (Guo et al., 2013), indicating a strong photochemical reaction during the pollution process, e.g., the photo-oxidation of $i$-butene to acetone (Guo et al., 2013). Therefore, vehicle exhaust emissions, solvent use, combustion, biogenic emissions, and industrial processes are important sources of NMVOCs at the observation sites, as further illustrated in the PMF source apportionment (in Sect. 3.2.2).

### 3.1.2 Diurnal variations in NMVOCs, $O_3$, and $NO_x$

The concentration characteristics of pollutants in the atmosphere are affected by the variation pattern of the atmospheric boundary layer, the intensity of photochemical reactions, and pollution source emissions (B. Wang et al., 2023). A selection of NMVOCs, $O_3$, and $NO_x$ were chosen, and their daily changes were analyzed, as shown in Fig. S4 in the Supplement. The diurnal variation in $O_3$ concentration shows unimodal characteristics. During the day, with the increase in temperature and light intensity, the concentration of $O_3$ gradually increased and reached a peak at about 14:00 LT, and then the concentration gradually decreased. This diurnal pattern was influenced by strong photochemical reactivity, boundary layer processes, and meteorological parameters. Higher $O_3$ production during the day indicates significant contributions from both photochemical reactions and atmospheric mixing processes. The diurnal variations in ethane, propane, isobutane, $n$-butane, isopentane, $n$-pentane, ethylene, propylene, acetylene, benzene, and toluene were similar, showing low concentrations in the daytime and high concentrations in the evening. This pattern is associated with a higher boundary layer and strong photochemical reactivity during the day (Tang et al., 2007). An elevated boundary layer is conducive to the dispersion of NMVOCs and other pollutants (Bon et al., 2011; Chen et al., 2022a), while strong photochemical reactions consume NMVOCs (Xia et al., 2014; Zhang et al., 2018). In addition, the peak concentrations of these NMVOCs occurred in the morning and evening (07:00–08:00 and 23:00–24:00 LT, respectively), showing a consistent daily pattern with $NO_x$. This suggests that the emissions of these NMVOCs were significantly influenced by motor vehicle emissions and fuel combustion. The higher NMVOC and $NO_x$ concentrations at night may have been caused by heavy-traffic emissions related to traditional nighttime activities in the city. Isoprene, a typical tracer of plant emissions, is highly dependent on temperature and solar radiation (Guenther et al., 1993; Sharkey et

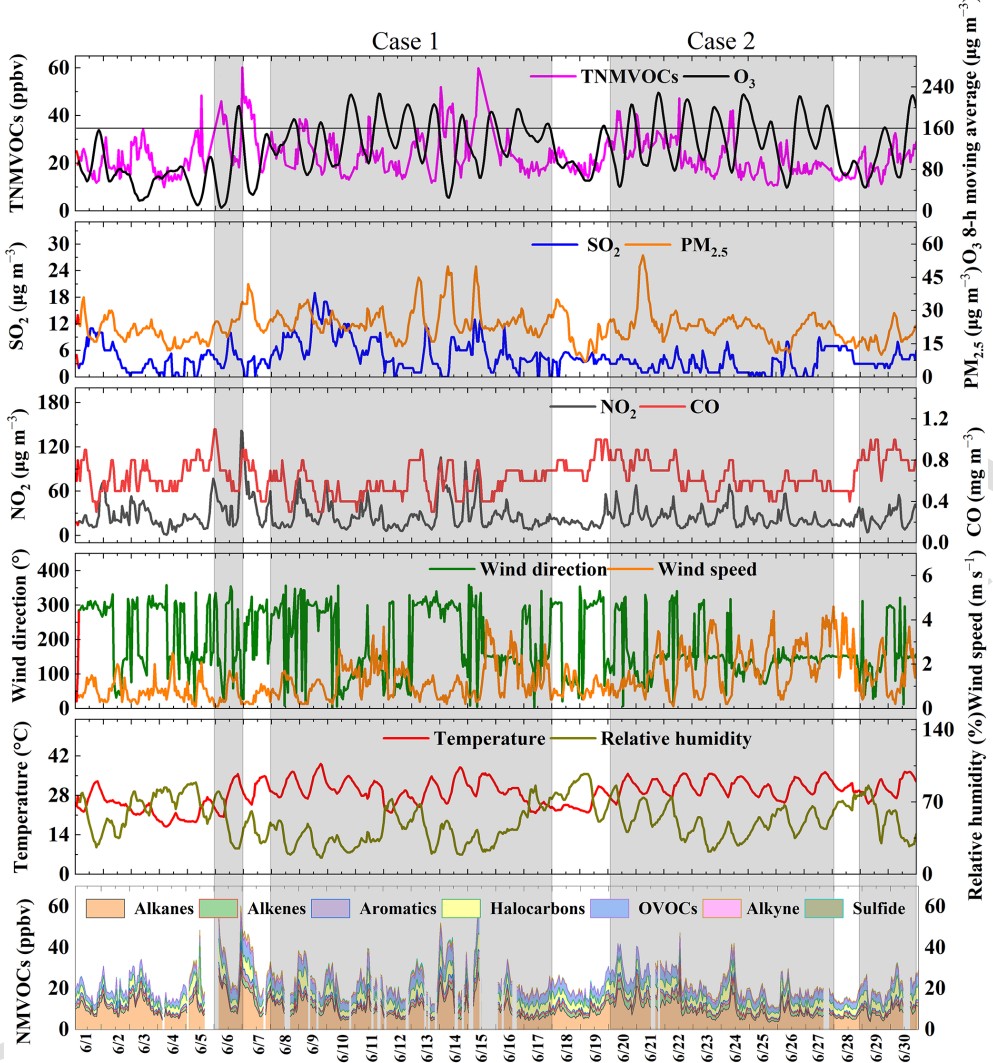

**Figure 1.** Time series illustrating hourly concentrations of TNMVOCs, the 8 h moving average of $O_3$, $SO_2$, $PM_{2.5}$, $NO_2$, CO, meteorological parameters (WD, WS, $T$, and RH), and NMVOCs during the sampling period. Gray regions represent $O_3$ pollution processes.

**Table 1.** Concentrations of NMVOCs (ppbv) during different processes in Zhengzhou.

| Species | Entire period (n = 652) | | Case 1: 8–17 June (n = 201) | | Case 2: 20–27 June (n = 184) | | Clean days (n = 224) | |
|---|---|---|---|---|---|---|---|---|
| | Range | Average ± SD | Range | Average ± SD | Range | Average ± SD | Range | Average ± SD |
| Alkanes | 3.6–30.7 | 10.0 ± 4.4 | 4.2–28.3 | 10.0 ± 4.6 | 3.6–24.6 | 9.6 ± 4.1 | 4.6–22.2 | 9.6 ± 3.9 |
| Alkenes | 0.4–10.7 | 2.0 ± 1.2 | 0.6–10.7 | 1.9 ± 1.2 | 0.6–10.7 | 2.5 ± 1.4 | 0.4–4.0 | 1.7 ± 0.7 |
| Aromatics | 0.3–5.0 | 1.1 ± 0.7 | 0.4–4 | 1.2 ± 0.8 | 0.3–3.1 | 1.1 ± 0.6 | 0.3–4.4 | 1.1 ± 0.6 |
| Halocarbons | 1.8–31.1 | 4.3 ± 1.9 | 2.0–10.6 | 4.5 ± 1.8 | 2.2–8.8 | 4.2 ± 1.4 | 1.8–31.1 | 3.9 ± 2.2 |
| OVOCs | 1.8–9.7 | 4.5 ± 1.3 | 3.4–9.7 | 5.3 ± 1.2 | 2.0–8.1 | 4.4 ± 1.1 | 1.8–8.6 | 3.9 ± 1.2 |
| Sulfide | 0.0–1.5 | 0.1 ± 0.2 | 0.0–1.5 | 0.2 ± 0.3 | 0.0–0.5 | 0.1 ± 0.1 | 0.0–1.0 | 0.1 ± 0.1 |
| Alkyne | 0.1–3.7 | 1.1 ± 0.6 | 0.2–3.2 | 1.1 ± 0.6 | 0.2–3.2 | 1.0 ± 0.5 | 0.1–3.7 | 1.0 ± 0.7 |
| TNMVOCs | 9.9–60.3 | 22.8 ± 8.3 | 0–60.0 | 24.1 ± 8.9 | 10.5–47.3 | 22.5 ± 7.4 | 9.9–48.5 | 20.8 ± 7.2 |

Note that $n$ stands for the total sampling number used for each process.

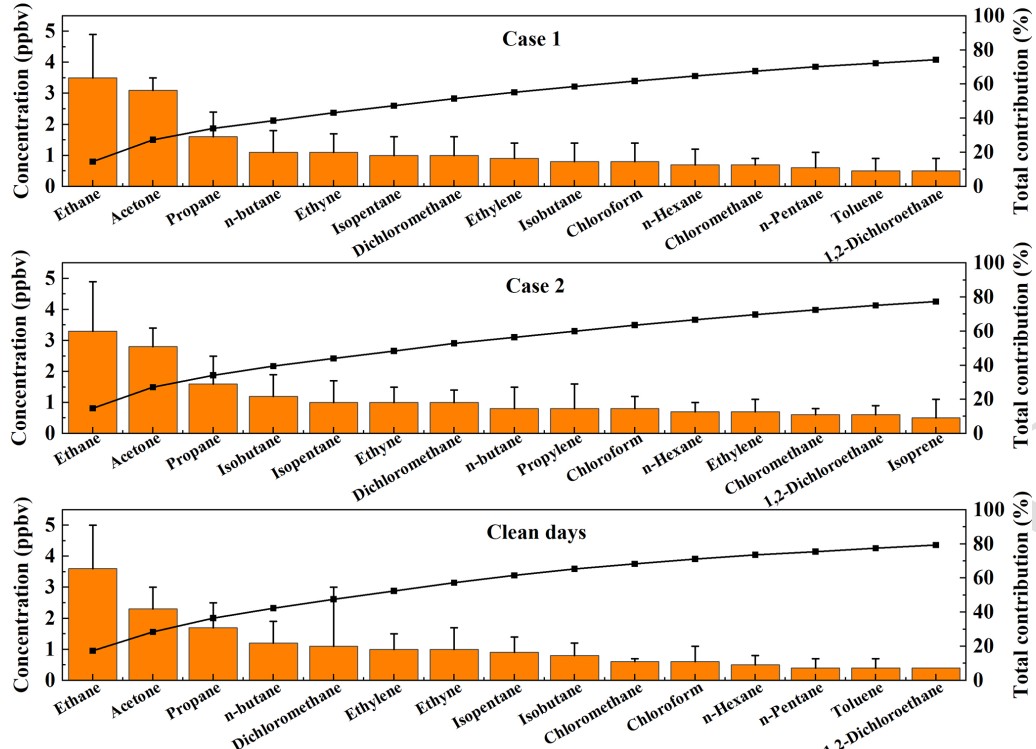

**Figure 2.** Comparisons of the concentrations of the 15 NMVOCs (ppbv) with the highest average mixing ratios for different processes. Error bars represent standard deviations.

al., 1996). Therefore, the concentration of isoprene increased significantly during the day (07:00–20:00 LT) and decreased significantly at night. It is worth noting that the concentration of isoprene showed a bimodal characteristic, with peaks at 10:00 and 15:00 LT. Previous studies have shown that the rate at which plants emit isoprene decreases when temperatures exceed 40 °C (Guenther et al., 1993; Sharkey et al., 1996). Therefore, the drop in isoprene concentrations observed at noon may have been due to excessive temperatures affecting biogenic emissions. Additionally, the concentration of OH radicals peaked at noon (Fig. S5 in the Supplement), leading to the rapid oxidation of isoprene by OH radicals, further contributing to the observed bimodal pattern (Paulot et al., 2009). Acetone comes from a wide range of sources, including vehicle emissions, industrial production, and secondary formation (Sha et al., 2021). Acetone remained at high concentrations throughout the day, and there was no obvious diurnal variation, suggesting that primary acetone sources near the site may have concealed the daytime acetone peak produced by photochemical reactions (Guo et al., 2013). Dichloromethane, which mainly originates from solvent use, exhibited high concentrations mainly at night (23:00–05:00 LT). This might be related to the longer atmospheric lifetime of dichloromethane and its lower boundary layer height at night (Li et al., 2018; Chen et al., 2022a).

## 3.2   Sources of NMVOCs

### 3.2.1   Diagnostic ratios

Ratios of specific NMVOCs can be used to assess the initial emission source of NMVOCs or the degree of photochemical reaction (Miller et al., 2012; An et al., 2014). The ratios of isopentane / $n$-pentane, toluene / benzene (T / B), and ($m$-/$p$-xylene) / ethylbenzene (E / X) are discussed in this study (Fig. 3).

In Case 1 and Case 2 and on clean days, the Pearson coefficients for isopentane and $n$-pentane were 0.7, 0.94, and 0.6, respectively, indicating a strong correlation and suggesting a common emission source between the two substances. Isopentane / $n$-pentane ratios of 0.8–0.9, 2.2–3.8, 1.5–3.0, and 1.8–4.6 (Fig. 3a) indicate that isopentane and $n$-pentane originate from NG, vehicle emissions, liquid gasoline, and fuel evaporation, respectively (An et al., 2014; Watson et al., 2001). In this study, the ratios for Case 1, Case 2, and the clean days were 0.7, 2.5, and 1.1, respectively. This suggests that isopentane and $n$-pentane may originate from NG emissions, vehicular exhausts, and liquid gasoline, respectively.

The T / B ratio can be used to distinguish between coal and biomass combustion (0.2–0.6), motor vehicle emissions ($\sim$ 2.0) (Liu et al., 2008), industrial processes (3.0–6.9) (Zhang et al., 2016), and fuel evaporation ($\sim$ 4.1) (Dai et al., 2013). In this study, the T / B ratios for the two O$_3$ pollution events

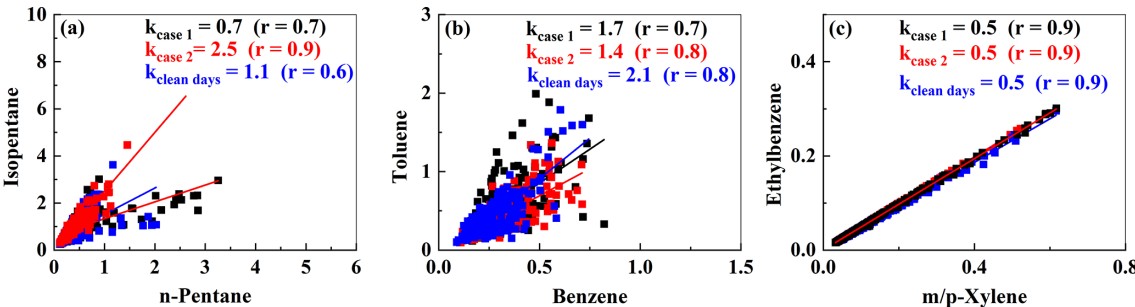

**Figure 3.** Correlations between compounds with different observation periods ($k$ denotes the slope).

were 1.7 and 1.4 (Fig. 3b), respectively, indicating that combustion and vehicle emissions were the main sources of benzene and toluene emissions (Hong et al., 2019).

Since $m$-xylene, $p$-xylene, and ethylbenzene share a common source but differ in terms of their OH radical reaction rate constants, the E/X ratio can be used to understand source characteristics (Miller et al., 2012; Yurdakul et al., 2018). During the pollution events and on clean days, $m$-xylene, $p$-xylene, and ethylbenzene showed a strong positive correlation ($r = 0.9$; Fig. 3c), indicating that they originated from a common emission source. Previous studies have shown that NMVOCs are transported from inner urban areas when the E/X ratio is 0.3–0.4 and that NMVOCs are transported from distant sources when the ratio is significantly higher than 0.3 (Monod et al., 2001). In this study, the E/X ratios for the two pollution events and the clean days came to 0.5, indicating that the air mass measured at the observation point was affected by air mass transport. We analyzed the relationship between ethylbenzene, $m$-xylene, $p$-xylene, the E/X ratio, wind direction, and wind speed. As shown in Fig. S6 in the Supplement, the concentrations of ethylbenzene, $m$-xylene, and $p$-xylene are mainly influenced by winds from the northwest, and their concentrations tend to increase with stronger wind speeds. Similarly, the E/X ratio also exhibits similar patterns of variation, further indicating the regional transport of ethylbenzene, $m$-xylene, and $p$-xylene from distant sources.

### 3.2.2 Source apportionment

In this study, the PMF 5.0 model from the Environmental Protection Agency (EPA) was used to analyze the source profile and species percentage of each source during the observation period to determine the relative contribution of each potential source, as shown in Fig. 4. Seven factors were determined by the model: combustion, industrial production, biogenic emissions, vehicular exhausts, LPG/NG, "solvent use 1", and "solvent use 2". A detailed analysis is given below.

Factor 1 was characterized by high percentages of acetylene (76 %), ethane, propane, ethylene benzene, and toluene. Acetylene is a typical tracer of coal burning (Barletta et al.,

2005). Ethane, propane, and ethylene are typical tracers of incomplete combustion (Guo et al., 2011; Ling et al., 2011). Therefore, Factor 1 was classified as combustion. The CPF plots indicate that the contributing direction was northwest at about $2\,\mathrm{m\,s^{-1}}$ (Fig. S7a in the Supplement).

Factor 2 was rich in $C_4$–$C_6$ alkanes, aromatics (toluene, ethylbenzene, $m$-xylene, $p$-xylene, $o$-xylene, and 1,2,4-trimethylbenzene), and halocarbons (1,2-dichloroethane and 1,2-dichloropropane). Previous studies have shown that these species are all related to industrial production. Therefore, Factor 2 was classified as industrial production. The CPF plots indicate that a local source with a low wind speed of $< 1\,\mathrm{m\,s^{-1}}$ was the dominant source (Fig. S7b).

Factor 3 was characterized by a high percentage (83 %) of isoprene, a typical tracer of biogenic emissions (Brown et al., 2007). High temperatures and strong radiation in summer are more conducive to the biogenic emissions of isoprene (Liu et al., 2016). Therefore, Factor 3 was classified as biogenic emissions. The CPF plots indicate that the dominant source direction was southwest when wind speeds were below $2\,\mathrm{m\,s^{-1}}$ (Fig. S7c).

Factor 4 was characterized by high percentages of $C_2$–$C_6$ alkanes (such as ethane, propane, isobutane, $n$-butane, isopentane, $n$-pentane, 2,2-dimethylbutane, and 2,3-dimethylbutane), benzene, toluene, ethylbenzene, $m$-xylene, and $p$-xylene), which are related to vehicular emissions (Jorquera and Rappenglück, 2004; Song et al., 2007; Chen et al., 2014). Therefore, Factor 4 was classified as vehicular exhausts. The CPF plots indicate that a local source with low wind speeds was the dominant source, likely related to the large amount of traffic on the main roads in the southern and western directions (Fig. S7d).

Factor 5 was characterized by high percentages of ethane, propane, isobutane, and propylene, which are the main components of LPG and NG (Shao et al., 2016; Song et al., 2007; Na et al., 2001). Therefore, Factor 5 was classified as an LPG and NG source. The CPF plots show that the dominant source direction of this factor was east at $1$–$2\,\mathrm{m\,s^{-1}}$ (Fig. S7e).

Factor 6 was characterized by high percentages of chloromethane, dichloromethane, tetrachloromethane, 1,2-dichloroethane, 1,2-dichloropropane, and ethyl acetate,

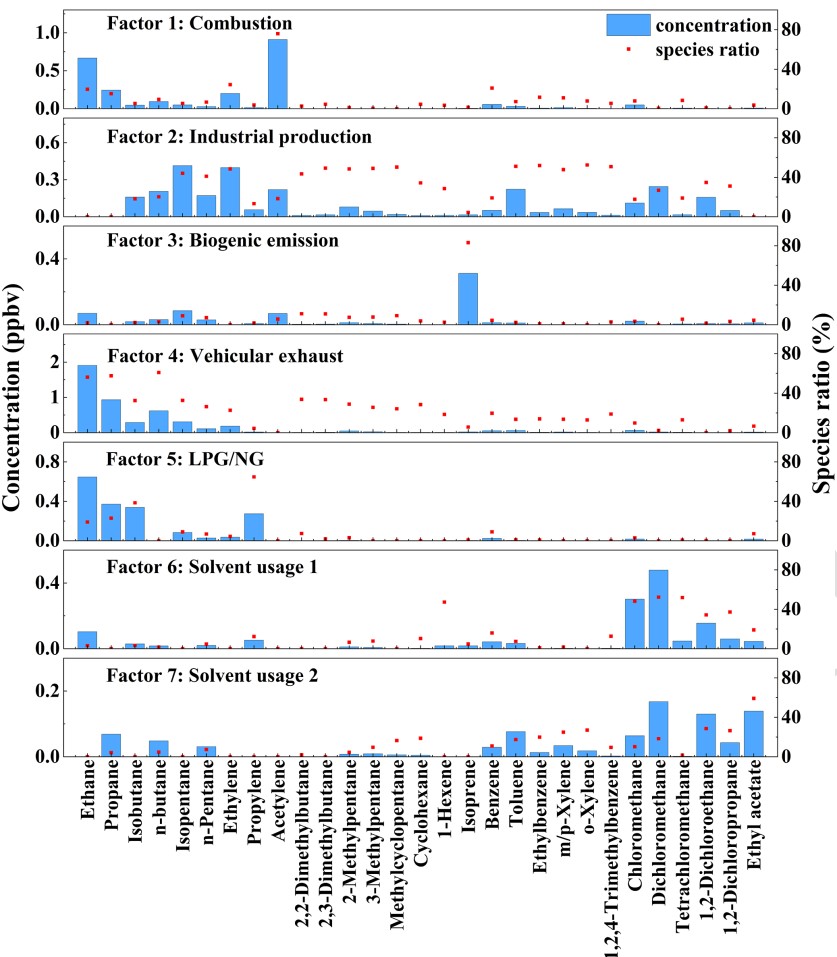

**Figure 4.** Source profiles and contributions of the NMVOCs during the observation period.

which are typical solvents for industrial applications (Li et al., 2020; Huang et al., 2014). Therefore, Factor 6 was classified as solvent usage 1. The CPF plots for this factor indicate that the dominant directions were northeast and southeast (Fig. S7f).

Factor 7 was dominated by methylcyclopentane, cyclohexane, TEXs (toluene, ethylbenzene, and xylenes (*m*-xylene, *p*-xylene, and *o*-xylene)), 1,2-dichloroethane, 1,2-dichloropropane, and ethyl acetate. Methylcyclopentane and cyclohexane are commonly used as solvents in industrial processes (Lyu et al., 2016; Yuan et al., 2013). TEXs are the main component of organic solvents (Guo et al., 2011; Watson et al., 2001). Therefore, Factor 7 was classified as solvent usage 2. The CPF plots for this factor indicate that the high CPF values were found near the center when the wind speed was low ($\leq 1\,\mathrm{m\,s^{-1}}$). This finding indicates that local emissions were the dominant source (Fig. S7g).

Figure 5 shows the proportion of each NMVOC source during the observation process. Throughout the entire observation period, vehicular exhausts were the main contributor, accounting for 28 %, followed by solvent usage

(27 %) and industrial production (22 %). Other sources, including LPG/NG (9 %), combustion sources (8 %), and biogenic emissions (6 %), contributed comparatively little. In Case 1, vehicular exhausts (30 %) were the largest contributor, followed by solvent usage (27 %) and industrial production (23 %). Compared with the Case-1 event, the contributions of solvent usage and industrial production in the Case-2 event did not change much, but the contribution of LPG/NG increased by 14 %, becoming an important source. On clean days, vehicular exhausts (35 %), solvent usage (25 %), and industrial production (21 %) were the most significant contributors. Compared to clean days, the contributions of solvent usage, industrial production, biogenic emissions, and LPG/NG increased during both pollution events, while the contributions of combustion sources and vehicular exhausts decreased. In summary, vehicular exhausts, solvent usage, and industrial production were major contributors to both O$_3$ pollution events and the clean days.

In summary, the observation sites are significantly influenced by vehicular exhausts, solvent usage, and industrial production. The results of this study show similarities in the

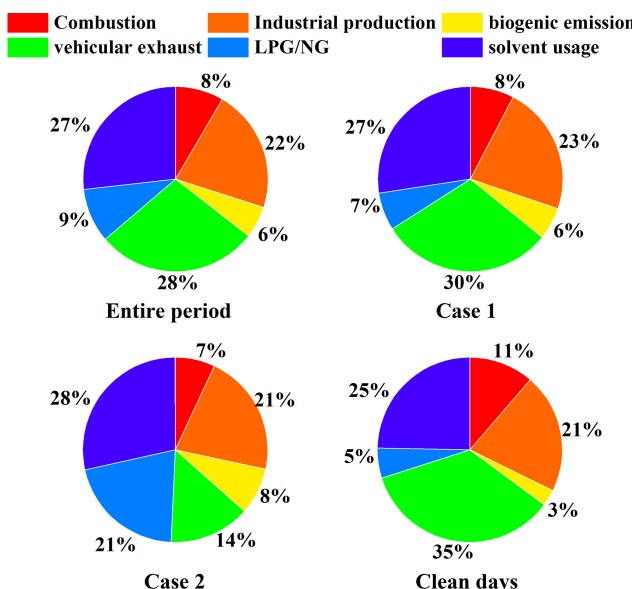

**Figure 5.** Source contributions to NMVOC concentrations during different periods.

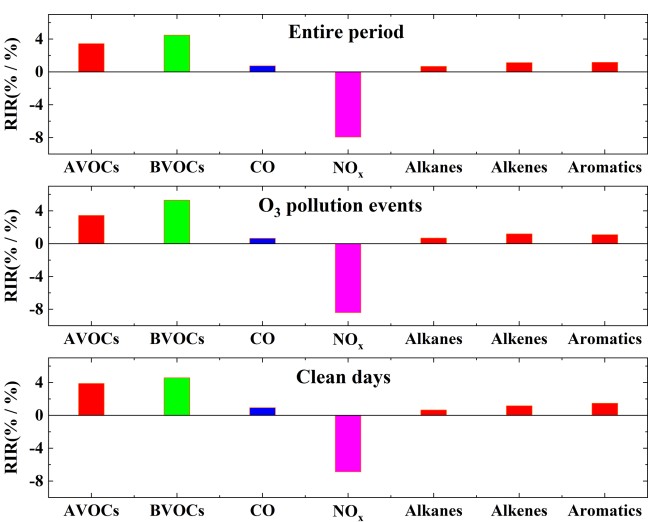

**Figure 6.** Average RIR values of $O_3$ for different species and groups during different processes in Zhengzhou.

source apportionment of NMVOCs in Zhengzhou during the summers of 2018 to 2021 (Yu et al., 2022; Guo et al., 2024). Yu et al. (2022) found that vehicular exhausts and industrial production contributed the most to NMVOC emissions in Zhengzhou from 2018 to 2020, with the main sources of NMVOCs in summer being vehicular exhausts, solvent usage, and industrial production. In contrast to the NMVOC source apportionment results from Li et al. (2020) for the $O_3$ pollution process in Zhengzhou that occurred in May 2018, the difference lies in the higher impact of solvent usage compared to that of vehicular exhausts and industrial production. This is mainly attributed to the fact that the observation site used in Li et al. (2021) was located at Zhengzhou University, making it more susceptible to the influence of chemical-reagent use. In comparison to the source apportionment of NMVOCs in Zhengzhou during winter (Zhang et al., 2021), combustion becomes an important contributor during winter due to the increased heating demand, while the contribution from solvent usage is relatively lower due to the cold temperatures. In comparison with other cities (Table S3 in the Supplement), vehicular exhausts in Zhengzhou contribute the most – more than in cities like Qingdao (Wu et al., 2023b), Xuchang (Qin et al., 2021), Guangzhou (Meng et al., 2022), Nanjing (Fan et al., 2021), Shijiazhuang (Guan et al., 2020), and Weinan (Hui et al., 2020) but less than in Changzhou (Z. Liu et al., 2023a) and on par with Beijing (Liu et al., 2020). Solvent usage is a larger contributor in Zhengzhou than in Qingdao (Wu et al., 2023b), Xuchang (Qin et al., 2021), Nanjing (Fan et al., 2021), Shijiazhuang (Guan et al., 2020), Weinan (Hui et al., 2020), Changzhou (Z. Liu et al., 2023a), and Beijing (Liu et al., 2020) but a smaller contributor than in Guangzhou (Meng et al., 2022). Industrial pro-

duction is a larger contributor in Zhengzhou than in Xuchang (Qin et al., 2021), Guangzhou (Meng et al., 2022), Nanjing (Fan et al., 2021), Weinan (Hui et al., 2020), and Changzhou (Z. Liu et al., 2023a) but a smaller contributor than in Shijiazhuang (Guan et al., 2020).

## 3.3 Contribution to $O_3$ formation

### 3.3.1 $O_3$ sensitivity analysis

In this study, the RIR of AVOCs, biogenic volatile organic compounds (BVOCs), CO, $NO_x$, alkanes, alkenes, and aromatics were calculated (Fig. 6). The RIR values for NMVOCs were all positive throughout the entire period, indicating that $O_3$ generation is most sensitive to reductions in NMVOCs. In comparison, the RIR value for $NO_x$ was negative, indicating that a reduction in $NO_x$ would lead to an increase in $O_3$ concentration. Among the AVOCs, aromatics exhibited the highest RIR values, followed by alkenes and alkanes. During both $O_3$ pollution events and on clean days, the RIR values for $NO_x$ were negative, while the RIR values for NMVOCs and CO were positive. In pollution events, except for those concerning BVOCs, the absolute values of RIR for each group and species were lower than those observed on clean days, indicating that the sensitivity of $O_3$ to NMVOCs, $NO_x$, and CO was higher on clean days than during the $O_3$ pollution events. Compared to clean days, during ozone pollution events, the RIR value for AVOCs decreased by 11 %, with aromatics showing the largest decrease (26 %), while alkanes and alkenes increased by 7 % and 3 %, respectively. Additionally, during pollution events, CO and $NO_x$ were reduced by 41 % and 18 %, respectively. Additionally, CO and NOx decreased by 29 % and 22 %.

Isoprene was the sole BVOC considered in this study and is an important tracer for indicating biogenic emissions (Xie

et al., 2021; Li et al., 2024; Qin et al., 2023). Throughout the entire period, especially during the pollution events, the RIR values for the AVOCs were lower than those for the BVOCs, indicating that $O_3$ formation was more sensitive to biogenic emissions. This may be due to increased BVOC emissions caused by higher temperatures and certain solar radiation conditions, combined with their high reactivity and $O_3$ formation potential. Studies in Yucheng (Zong et al., 2018), Leshan (Xie et al., 2021), and Nanjing (Fan et al., 2021; Ming et al., 2020) have shown that $O_3$ is highly sensitive to BVOCs. Studies in Zhengzhou (Wang et al., 2022), Hangzhou (Zhao et al., 2020), and Hong Kong (Wang et al., 2017) have suggested that $O_3$ exhibits greater sensitivity to BVOCs than AVOCS during hot seasons. In their study on $O_3$ source apportionment in the province of Henan, where Zhengzhou is located, Wang et al. (2019) found that BVOCs contribute approximately 23.9 % of the $O_3$ attributed to NMVOCs. Therefore, the contribution of BVOCs to $O_3$ is very important.

### 3.3.2 Results of the Empirical Kinetics Modeling Approach (EKMA)

Given the current limitations in implementing appropriate control measures for BVOCs, the following analysis focuses solely on the impact of AVOCs and $NO_x$ on $O_3$ formation. The EKMA curve based on the OBM is shown in Fig. 7. It can be seen from the EKMA curve that there is a highly nonlinear relationship between the generation of $O_3$ and its precursor compounds, AVOCs and $NO_x$. The same $O_3$ concentration can be generated using different combinations of AVOC and $NO_x$ concentrations. In the figure, AVOCs and $NO_x$ at 100 % represent the base case, and the $y$ axis and $x$ axis represent the percentages of AVOCs and $NO_x$, respectively, relative to the actual observed mixture ratio (100 %). The straight lines in the figure, known as ridge lines, are formed by the junctions of turning points on $O_3$ concentration lines (Dodge, 1977).

The ridge divides the graph into two regions: the upper-left region and the lower-right region, with large differences in $O_3$ generation between these two regions. In the lower-right region, the $O_3$ concentration lines and the horizontal coordinates show a parallel relationship. If the $NO_x$ concentration remains constant, the $O_3$ concentration does not change with the variations in AVOC concentration. When the AVOC concentration remains constant, the concentration of $O_3$ decreases as the $NO_x$ concentration decreases. Therefore, in this region, $O_3$ generation is controlled by $NO_x$. In the upper-left region, if only the concentration of AVOCs is reduced, the concentration of $O_3$ will decrease significantly, whereas if only the concentration of $NO_x$ is reduced, the concentration of $O_3$ will first rise and then decrease. In this region, $O_3$ generation is controlled by AVOCs. Near the ridge line, when $NO_x$ and AVOCs are reduced at the same time, the $O_3$ concentration decreases, and $O_3$ generation is in the cooperative control area of the AVOCs and $NO_x$.

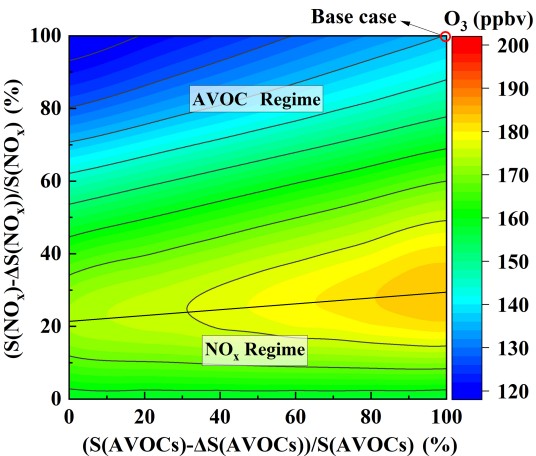

**Figure 7.** Isopleth diagram illustrating modeled $O_3$ using the remaining percentages of $S$(AVOCs) and $S$($NO_x$).

The slope of this ridge line on this EKMA curve is about $6:1$, meaning that reducing $NO_x$ and AVOCs along this ridge is the fastest way to reduce the $O_3$ concentration. As shown in the figure, Zhengzhou is a typical AVOC control area, and $O_3$ is very sensitive to changes in AVOCs. At the same time, Case 1, Case 2, and clean days are all above the ridge line and belong to the AVOC control region (Fig. S8 in the Supplement). Therefore, reducing AVOCs can effectively reduce the generation of $O_3$.

### 3.3.3 Control strategies of $O_3$

The above analysis, based on single species ($NO_x$ or AVOCs), was only used to discuss the sensitivity of $O_3$ concentration to precursors; however, such extreme control measures are difficult to achieve. Usually, in actual operations, the method of simultaneously controlling $NO_x$ and AVOC emissions is adopted to reduce the concentration of $O_3$. To establish a reasonable and effective AVOC and $NO_x$ emission reduction plan, we conducted a series of simulations to calculate the $O_3$ concentration by adjusting the ratio of input AVOCs to $NO_x$. The following analysis examines the reduction cases with regard to $O_3$ control between 10:00 and 16:00 LT during the observation period.

Figure 8 shows different reduction schemes. In Fig. 8, the $y$ axis and $x$ axis correspond to the reduction percentages of $NO_x$ (or $NO_x$ and AVOCs) and the percentage reduction in $O_3$ concentration, respectively. Positive and negative values represent increases and decreases in $O_3$ concentration compared to the base case, respectively. The results show that $O_3$ concentration will eventually decline regardless of the reduction method used, with the trend of change illustrated in Fig. 8a. As shown in Fig. 8b, if only $NO_x$ is reduced, when the emission reduction is less than 60 %, $O_3$ concentration increases, and when the emission reduction is greater than 60 %, $O_3$ concentration decreases. Therefore, reducing only

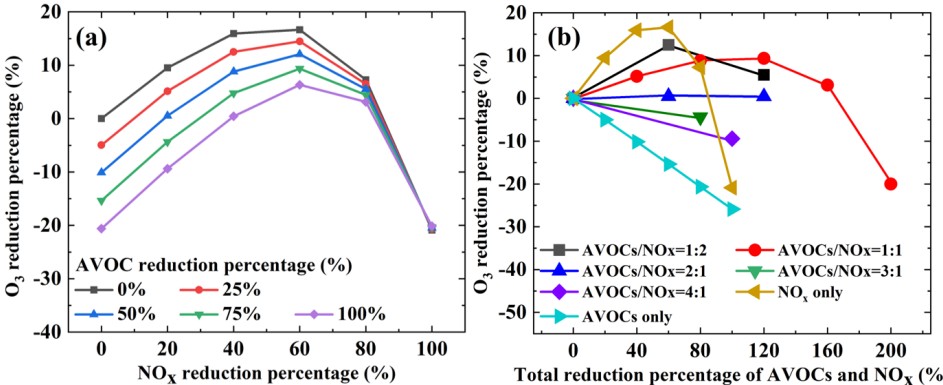

**Figure 8.** Response of $O_3$ concentration to different AVOC and $NO_x$ reduction percentages. Note that AVOCs / $NO_x$ represents the ratio of the percentage reduction in AVOCs to that in $NO_x$.

$NO_x$ emissions is not conducive to reducing $O_3$ concentration. When the AVOCs / $NO_x$ reduction ratio is 1 : 2 or 1 : 1, the variations in $O_3$ concentration exhibits a similar trend to the variations in $NO_x$ emission reduction alone, and $O_3$ concentration increases first and then decreases. When the AVOCs / $NO_x$ reduction ratio is 2 : 1, $O_3$ concentration increases to a certain extent. When the AVOCs / $NO_x$ emission reduction ratio is 3 : 1 or 4 : 1, $O_3$ concentration continues to decline, and the rate of decline in $O_3$ concentration is higher at 4 : 1 than at 3 : 1. When only AVOC emissions are reduced, there is a continuous downward trend in $O_3$ concentration, and the decline rate is very fast. However, with regard to actual production activities, simply reducing AVOC emissions cannot be achieved, which is not conducive to policy implementation. Therefore, from the perspective of comprehensive emission reduction effects, the AVOCs / $NO_x$ reduction ratio should be no less than 3 : 1 in order for it to be conducive to the reduction in $O_3$ concentration.

In addition, this study analyzed $O_3$ reduction schemes from 10:00 to 16:00 LT. It can be seen in Fig. S9 in the Supplement that as $NO_x$ decreases, $O_3$ concentration increases and then decreases. When the AVOC reduction ratio is fixed and the $NO_x$ reduction ratio is less than 60 %, $O_3$ concentration increases as $NO_x$ decreases. In this case, $O_3$ concentration increases by 30 %, 21 %, 16 %, 13 %, 13 %, 15 %, and 15 % from 10:00 to 16:00 LT – that is, under the AVOC scenario without reduction. When the $NO_x$ reduction ratio is greater than 60 %, $O_3$ concentration decreases as $NO_x$ decreases. When the reduction is at its greatest (i.e., 100 % reduction in $NO_x$ and AVOCs), $O_3$ concentration at 10:00 LT is still increased compared with the observed atmospheric concentration, which increases by 14 %. $O_3$ concentration at hourly intervals from 11:00 to 16:00 LT decreases by 2 %, 15 %, 25 %, 32 %, 36 %, and 36 %, respectively.

From 10:00 to 16:00 LT, when only $NO_x$ is reduced, $O_3$ concentration increases and then decreases. When only AVOCs are reduced, $O_3$ concentration continues to decrease. When the AVOCs / $NO_x$ reduction ratio is less than 2 : 1, $O_3$ concentration increases and then decreases. When the AVOCs / $NO_x$ reduction ratio is greater than 2 : 1, $O_3$ concentration continues to decrease. When the AVOCs / $NO_x$ reduction ratio is equal to 4 : 1, $O_3$ concentration decreases the most and the fastest. When the AVOCs / $NO_x$ reduction ratio equals 4:1, the maximum reduction in $O_3$ concentration at hourly intervals from 10:00 to 16:00 LT amounts to 3 %, 6 %, 10 %, 11 %, 13 %, and 13 %, respectively.

## 4 Conclusions

Summer $O_3$ pollution remains an important environmental issue in Zhengzhou. This study investigated the characteristics and emission sources of $O_3$ precursors from 1–30 June 2023. The OBM was used to analyze the influence of precursors on the formation of $O_3$, and an emission reduction strategy for these precursors was proposed to control the concentration of $O_3$. During the entire period, the concentration of TNMVOCs varied from 9.9 to 60.3 ppbv, with an average value of $22.9 \pm 8.3$ ppbv. The average concentrations of TNMVOCs during $O_3$ pollution events were higher than on clean days. Alkanes (44 %), OVOCs (20 %), and halocarbons (19 %) were the most abundant NMVOC groups. The most abundant species during the $O_3$ pollution events and on clean days were ethane, acetone, and propane. The average concentrations of $NO_2$ during pollution events were higher than on clean days, while the average concentrations of NO were lower than on clean days. Therefore, increasing concentrations of $O_3$ precursors were found to be a significant factor in the formation of $O_3$ pollution. At the same time, the unfavorable meteorological conditions of high temperatures and low RH in the observation process were also important factors in the formation of $O_3$ pollution. Further analysis of the sources revealed that vehicular exhausts (28 %), solvent usage (27 %), and industrial production (22 %) were the main emission sources of NMVOCs. The increase in solvent usage, biogenic emissions, and LPN/NG contributions is an important cause of $O_3$ pollution. An analysis of the sensitivity

of $O_3$ to precursors found that NMVOCs had the highest RIR values, while $NO_x$ had negative RIR values. Alkenes had the highest RIR values among the AVOCs. It should be noted that the RIR value of the BVOCs was greater than that of the AVOCs. The local $O_3$ formations occurred under an AVOC-limited regime, which means that reducing concentrations of AVOCs is an effective way to reduce $O_3$ concentration. We recommend a minimum AVOCs / $NO_x$ reduction ratio of no less than 3 : 1 to effectively reduce $O_3$ formation.

**Data availability.** The data set is available to the public and can be accessed upon request from Ruiqin Zhang (rqzhang@zzu.edu.cn).

**Supplement.** The supplement related to this article is available online at: https://doi.org/10.5194/acp-24-1-2024-supplement.

**Author contributions.** DZ performed chemical modeling analyses of the OBMs and wrote the paper. XL collected the data and contributed to the data analysis. RZ designed and revised the paper. QX, FS, and SW contributed to the discussions of the results. MY and YX provided part of the data from Zhengzhou.

**Competing interests.** The contact author has declared that none of the authors has any competing interests.

**Disclaimer.** Publisher's note: Copernicus Publications remains neutral with regard to jurisdictional claims made in the text, published maps, institutional affiliations, or any other geographical representation in this paper. While Copernicus Publications makes every effort to include appropriate place names, the final responsibility lies with the authors.

**Acknowledgements.** This work was supported by the Zhengzhou $PM_{2.5}$ and $O_3$ Collaborative Control and Monitoring Project (grant no. 20220347A), the Natural Science Foundation of Henan Province (grant no. 232300421395), and the National Key Research and Development Program of China (grant no. 2017YFC0212403).

**Financial support.** This research has been supported by the Zhengzhou $PM_{2.5}$ and $O_3$ Collaborative Control and Monitoring Project (grant no. 20220347A), the Natural Science Foundation of Henan Province (grant no. 232300421395), and the National Key Research and Development Program of China (grant no. 2017YFC0212403).

**Review statement.** This paper was edited by Rob MacKenzie and reviewed by two anonymous referees.

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

## Remarks from the language copy-editor

CE1 Please check and confirm the inserted definition is correct.

## Remarks from the typesetter

TS1 To avoid any errors (and the need to publish a corrigendum), please provide all corrections either as comments in the PDF (using the Adobe annotations tool) or a Word file listing the changes (as you did to answer our remarks). Potential changes you had added in the PDF file itself were not visible and could therefore not be considered in the proofreading; we could only insert the changes listed in your Word file. In case more corrections are necessary, please provide them as described.

TS2 Please check throughout the text that all vectors are denoted by bold italics and matrices by bold roman.

TS3 Please note that the changes were not found in the PDF. Please clearly explain which letters need to be adjusted

TS4 Please confirm "LT" for "local time" throughout the text.

TS5 Due to the requested changes, we have to forward your requests to the handling editor for approval. To explain the corrections needed to the editor, please send me the reason why these corrections are necessary. Please note that the status of your paper will be changed to "Post-review adjustments" until the editor has made their decision. We will keep you informed via email.

TS6 Please provide the exact date and location (city) of this conference.

TS7 Please provide a persistent identifier (ISBN or DOI preferred).