# Peer review of "Characteristics and sources of NMVOCs and the O3-NOx-NMVOCs relationships in Zhengzhou, China"

_EGUsphere, 2023_

## Referee Comment (RC1)

**Review for "Characteristics and sources of VOCs and the $O_3$-$NO_x$-VOC relationships in the central plain city, China.**

**Summary:**

This manuscript presents an interesting study, investigating what is driving ozone production in the region of Zhengzhou. The authors have investigated both the reactivity of individual/grouped VOCs as well as a source apportionment analysis, and a box model has been used to produce an ozone production isopleth. Whilst these are the key sections crucial to any exploration of the O3-NOx-VOC relationships in an urban centre, the authors could work on adding some more detail to each section. In addition, more work is needed to bring together all of the results sections, with more discussion on what the data is telling us and what this could mean for what is driving ozone production. With these changes in place, I would recommend this manuscript for publication in the Journal of Atmospheric Chemistry and Physics.

This manuscript is generally well written, with a small number of couple of grammatical mistakes (e.g. line 74: is in a VOCs-limited regimes, change to "is in a VOC-limited regime").

Some conclusions are made throughout the manuscript that are not so clearly backed up by the results. The authors could add some more quantification (e.g. percentage differences), when comparing Case 1, Case 2 and clean days. If the main message is that the sources are the same during both polluted and clean days, this needs to be made clearer. If this is the case, what could be driving the ozone production? Is it elevated concentrations? Could it be temperature? This can all be brought together in the conclusions.

The authors should include an expanded version of Table 1, which shows average mixing ratios for all the VOCs measured. It should also be made clear which of these VOCs are used in which part of the study (e.g. OBM, source apportionment etc.). The should be made available in the supplementary.

The authors should also consider changing the title from "central plain city, China" to the name of the city (Zhengzhou) for clarity.

To improve the coherency of the manuscript, the authors should bring together findings from previous sections into the next section. For example, there are sections on concentrations and sources of a variety of VOCs, but then the last results section ignores that knowledge and just focuses on varying AVOCs by bulk.

The author should incorporate the results from previous sections into this last section. For example, how does O3 production change when NOx is varied alongside reductions in each source respectively?

The conclusions should bring together all of the findings of each results section.

**Detailed comments:**

Remove "of course"

Although I recognise that additional details on the GC-MS instrumentation can be found in a previous publication, the authors should include a few more details about instrumentation in the sample collection section. What is the time resolution of samples? How long were samples captured for? What were they captured in, and how long were they left before being analysed by the instrumentation?

CPF is described as a "new source identification tool", but papers from 2006 and 2007 are referenced. Is this technique new? Have there been new applications of this?

The MCM is no longer hosted at the University of Leeds and hasn't been for a while. I suggest you provide the new working link, hosted by the University of York (https://mcm.york.ac.uk/MCM/).

The authors should also describe how they have calculated $P(O_3)$ from the model, if this was not a direct output. If the authors are using a specific box model (e.g. DSMACC, AtChem2), please specify which one and include the appropriate references.

Was any interpolation or averaging of the data performed to generate 1 h time resolution data for the model?

Why was WS constrained in your box model? This parameter is not used by the MCM. Was it used to calculate other factors in your model?

Please list all the measured VOCs, indicating which ones were selected, in a table. It would also be helpful to see average concentrations for these compounds. All of this should be presented in the supplementary.

What are the default values for $j(H_2O_2)$ and $j(O_1D)$ and planetary boundary layer? Do you think these are representative of the site? Have any dilution rates been applied to compounds generated in the model, and if so, are these varied based on the default PBLH? The authors need to highlight why these values were used. If measured or modelled values cannot be used, it is important to discuss the uncertainties associated with using default values for your results.

Please provide more context for why a value of 0.6 indicates the model is performing well. What is the acceptable range for this value to be? Which compound did you evaluate the performance of the model using? What was constrained in the model when you did this, and what was the model left to calculate. Much more detail is required here.

What is meant by "short pollution process"? Do you mean that the pollution event was too short, and you only looked at periods where pollution remained high for at least a week for better data coverage? Please clarify.

This is a very large proportion of halocarbons. It would be really interesting to see which VOCs are included in each category in your study, and what the average mixing ratios were across the sampling period. Please include this as an expanded version of Table 1 in the supplementary.

Remove "if"

Explain what you mean by the "top fifteen VOC species". Do you mean the highest average mixing ratio? If so, please replace "top" with something like "the fifteen VOCs with the highest average mixing ratio across the observation period."

"As illustrated in the following PMF source apportionment". Where is this? I cannot see any source apportionment in this section, just some discussion of the possible sources of different VOCs from the literature.

I don't understand what Figure 2 shows. What does the line represent and what are the bars? What are the authors trying to say using this figure?

"Major" VOCs – why exactly were these selected? You could either just say "a selection of VOCs", or if you are pointing out that these are the major ones, you need to explain why.

"photochemical reaction" – change to "photochemical reactivity", as there are multiple photochemical reactions happening.

Explain what you mean by "the daily change was consistent with $NO_x$".

What are "traditional nighttime activities"?

Acetone is a "common VOC" – what does this mean? Consider changing the language.

Could it be that the pentanes come from a mix of sources? The text implies that for Case 1, Case 2 and clean days, the sources are exclusively NG emissions, Vehicular exhaust emissions, and liquid gasoline emissions respectively.

Was this value 0.5 for all three case? How does this ratio vary with wind speed? Perhaps this could further support the argument of E/X rations indicating that air masses were affected by transport?

Generally for this section, more discussion of the potential locations of some of these sources would be good to see. There are two solvent factors. Why is this? Different industrial sources? Are there any industrial sources in the northeast and southeast of the site? For factor 7, are there any industrial sites near the observation site? Are the source apportionment factors consistent with the geography of the site?

This is really interesting. Are you saying that O3 pollution events appear to be independent of the VOC sources, and these remain broadly similar? If this is the case, then what is driving these O3 pollution events. Is it the meteorology?

This is the wrong way around - O3 formation was more sensitive to biogenic emissions, not "biogenic emissions was more sensitive to O3 formation".

From Figure 6, the RIRs look broadly similar between Case 1, Case 2, and clean. In the text, the authors have suggested there is a difference between clean and polluted days. Could you please quantify what this difference is? For example, what is the percentage difference in RIRs during polluted days, and is this significant?

Please clarify which data you have used to generate the Isopleth in Figure 7. Have you used the average peak $O_3$ across the entire observation period? Or is this the average daytime $O_3$ between particular hours? There is a point for "base case", but it would be interesting to see where Case1, Case2 and clean days would appear on this isopleth.

This seems quite general, after the detail you have gone into earlier on the RIRs for VOC groups and looking closely at different sources. I understand why biogenic compounds have not been included, but the impact of these should be further acknowledged in the text as changes to biogenic emissions present a large uncertainty to how O3 production might change in the future. It would have been interesting to see how O3 production might change when the different sources are reduced alongside NOx. This is more useful to policy than just investigating different percentage changes in bulk AVOCs, as it would be challenging to reduce AVOC emissions in this way.

You say P(O3) increases respectively, but what is this respective to? Is this per hour?

The authors could work on bringing all of there results together in the conclusion, rather than just summarising the key result of each section in turn. What is driving O3 production? Is it related to sources? Is it the meteorology?

Olefins mentioned here, earlier referred to as alkenes. Please change to alkenes.

---

## Referee Comment (RC2)

The study investigates the characteristics of VOCs and their importance in ozone formation during June in Zhengzhou City, China. The study focuses on the interesting relationship of O3-NOx-VOCs during summer, which is crucial for ozone control strategies. The study compares $O_3$ pollution events and clean days regarding different sources and $O_3$ formation sensitivity. However, the manuscript is poorly written and not up to the mark for consideration for publication in ACP. The authors fail to discuss crucial sections of the manuscript.

The authors do not include basic details about the instrumentation and dataset. Why is only a small part of the VOC measurement included in the PMF analysis? The reason is not mentioned properly in the manuscript or supplementary.

More in-depth comparisons with similar previous studies in Chinese/ Zhengzhou city would enhance the wider impact of the manuscript. Details about common factors and key trace VOC species can also be included. Any new or unique source or marker emerging from the region during the study may provide valuable insights.

Discussion related to the influence of meteorology and the transport of air masses needs to be included and explained.

Some of the statements require supporting references and proper reasoning. Also, VOCs can be changed to NMVOCs throughout the manuscript.

**Detailed comments:**

**Lines 114-115:** "The sampling site is surrounded by residential areas, commercial areas, and office buildings, and there are no obvious atmospheric pollution sources nearby, which is a typical urban site."

These lines should be changed to the following for better clarity.

'The sampling site is a typical urban site, surrounded by residential areas, commercial areas, and office buildings. There are no point sources of air pollution nearby (mention up to how much radius)."

**Line 99** 'heaviest' should be 'highest'

**Lines 116-117:** "The sampling site is surrounded by roads and vegetation, and the sampling may be affected by motor vehicle emissions and plant emissions." changed to "The sampling site may be affected by motor vehicle and plant emissions."

**Section 2.2. Sample collection and chemical analysis**

More details should be provided about the instruments used for supporting measurements. Details about the input of sampling air should be added. Details about sampling dates, calibration, sampling time resolution, etc, should be added.

**Section 2.3 PMF model**

Only 29 out of 115 VOCs have been used for PMF analysis. Why is that? Multiple studies have used more than ~90 VOCs in the PMF model and shown its advantages. The author mentioned that species with missing samples were excluded. A very high proportion of sampling is missed out. This questions the reliability of the collected samples and dataset. More details about the error matrix and uncertainty should be included in this section. Why were the 5-factor solution, 6-factor solution, and 8-factor solution not considered? Did the author observe any source tracers or markers mixing in these solutions?

**Section 2.4 Conditional bivariate probability function analysis**

More details in the section are required.

**Line 192** areas are non-polluting processes (clean days). Remove 'non-polluting processes'

**Line 192-193** During the observation, $O_3$ polluted days were 22 days, accounting for 73%. You mentioned cases 1 (8th-17th Jun.) and 2 (20th-27th Jun.) as pollution events, which is 18 days instead of 22 days. There is a discrepancy. Most of the days are included in ozone pollution events in June.

**Line 198** 'The mean concentrations' change to 'The daily mean concentrations'.

**Lines 199-200:** All were lower than the ambient air quality standard value (National or WHO). Add reference. Also, compare the values with national standards/ WHO standards for pollutant criteria.

The mean of each VOCs species (115 in number) or at least which species have been measured should be added to the supplementary in Table.

The variation of different families or groups of VOCs can be added as a time series in Figure 1. Also, the left and right y-axis are not aligned properly with the text.

**Section 3.2 Sources of VOCs**

**Figure 4:** The VOCs concentration is given in ug/m3, while it is mentioned as ppbv in the whole manuscript. Please check the discrepancy in the units.

An explanation of each factor concerning the sampling site is required. As you mentioned, there are no point sources nearby, so why is industrial pollution showing 22%? What could be the contributing factors for it?

Have you also performed PMF on clean days and polluted days simultaneously? One suggestion is to check if PMF gives different results in different cases (Case 1 and Case 2 and clean days) and compare the results.

Secondary VOCs, such as OVOCs and amines, play an important role in identifying VOCs sources. The ratio of OVOCs and other family groups should also be analysed in each source to determine their contribution. This will give great insights into the source characterisation.

**More explanation is required for Figure 5.**

**Liness 366-369** indicate that similar sources contribute to O3 pollution events and clean days. Does it mean that the emissions of primary and secondary VOCs do not influence $O_3$ formation? Then, what are the reasons for O3 formation in the area?

**Section 3.3.1 $O_3$ sensitivity analysis**

**Line 383** indicates that $O_3$ formation is more sensitive to biogenic emissions. Why is that? Add references to previous studies in urban settings, how AVOCs and BVOCs vary and their effect on ozone levels.

**Line 382-383** What could be the possible reasons for the RIR of BVOCs being higher than AVOCs? What are the biogenic VOCs species you have included in this analysis? Add these details in the section.

**Figure 6:** The RIR (%) looks similar for every case. Even for clean days, aromatics show higher values than polluted events. I suggest to check the values.

Another section should be added to compare the study results with the source apportionment studies in the city in different periods or seasons. Also, comparison with other Chinese cities studies can add extra value to the analysis.

**Section 4 Conclusions**:

The authors have just given a summary of the results obtained. No explanation is included about why and how any trend follows the study of how sources influence $O_3$ formation in the area. What could be the driving factors for the presence of any particular source? You should include more details in the section.

---

## Author Comment (AC1)

**Referee comments:**

This manuscript presents an interesting study, investigating what is driving ozone production in the region of Zhengzhou. The authors have investigated both the reactivity of individual/grouped VOCs as well as a source apportionment analysis, and a box model has been used to produce an ozone production isopleth. Whilst these are the key sections crucial to any exploration of the $O_3$-$NO_x$-VOC relationships in an urban centre, the authors could work on adding some more detail to each section. In addition, more work is needed to bring together all of the results sections, with more discussion on what the data is telling us and what this could mean for what is driving ozone production. With these changes in place, I would recommend this manuscript for publication in the Journal of Atmospheric Chemistry and Physics.

This manuscript is generally well written, with a small number of couple of grammatical mistakes (e.g. line 74: is in a VOCs-limited regimes, change to "is in a VOC-limited regime").

Some conclusions are made throughout the manuscript that are not so clearly backed up by the results. The authors could add some more quantification (e.g. percentage differences), when comparing Case 1, Case 2 and clean days. If the main message is that the sources are the same during both polluted and clean days, this needs to be made clearer. If this

is the case, what could be driving the ozone production? Is it elevated concentrations? Could it be temperature? This can all be brought together in the conclusions.

The authors should include an expanded version of Table 1, which shows average mixing ratios for all the VOCs measured. It should also be made clear which of these VOCs are used in which part of the study (e.g. OBM, source apportionment etc.). The should be made available in the supplementary.

The authors should also consider changing the title from "central plain city, China" to the name of the city (Zhengzhou) for clarity.

To improve the coherency of the manuscript, the authors should bring together findings from previous sections into the next section. For example, there are sections on concentrations and sources of a variety of VOCs, but then the last results section ignores that knowledge and just focuses on varying AVOCs by bulk.

The author should incorporate the results from previous sections into this last section. For example, how does $O_3$ production change when NOx is varied alongside reductions in each source respectively? The conclusions should bring together all of the findings of each results section.

Reviewers' comments:

Thank you for your careful reading of our paper and valuable comments

and suggestions. We believe that we have adequately addressed your comments. To facilitate your review, the comments are in black, and the responses are in blue. The major changes that have been made according to these responses were marked in yellow color in the highlighted copy of the revised manuscript. And our own minor changes were marked in red font. Note that the following line numbers are shown in the corrected version.

**Response:** Thank you for your careful reading of our paper and valuable comments and suggestions. We believe that we have adequately addressed your comments. To facilitate your review, the comments are in black, and the responses are in blue. The major changes that have been made according to these responses were marked in yellow color in the highlighted copy of the revised manuscript. And our own minor changes were marked in red font. Note that the following line numbers are shown in the corrected version.

We have made the corrections as per your suggestions by fixing the grammatical errors and changing the title from "central plain city, China" to the city name (Zhengzhou) for clarity. Thank you for your valuable feedback, these modifications will enhance the clarity and accuracy of the manuscript. Furthermore, in response to the comments of reviewer 1, we have revised the abbreviation in

the manuscript from "VOCs" to "NMVOCs" as recommended. But, in Section 3.3, we further differentiate NMVOCs into AVOC and BVOC. The abbreviations for AVOCs and BVOCs remain unchanged.

Major comments:

44    Remove "of course"

**Response:** Thanks for the suggestion. It has been deleted.

199    Although I recognise that additional details on the GC-MS instrumentation can be found in a previous publication, the authors should include a few more details about instrumentation in the sample collection section. What is the time resolution of samples? How long were samples captured for? What were they captured in, and how long were they left before being analysed by the instrumentation?

**Response:** Thanks for the comments.

I have included more details about the GC-MS instrumentation in the sample collection section, addressing the time resolution of samples. (Section 2.2. Line 120-128).

Line 120-128: The time resolution of the instrument is 1 hour, and the flow rate is 60 mL/min. The air sample was collected for the

first 5 minutes of each hour and then pre-concentrated through a cold trap to remove $H_2O_2$ and $CO_2$. The sample was captured using an empty capillary column. After pre-concentration, the sample was desorbed by rapid heating and introduced into an analytical system. After separation by chromatographic column, the sample was detected by FID (for C2-C5 hydrocarbons) and MS (for C5-C12 hydrocarbons, halocarbons and OVOCs). The correlation coefficient of the standard curve of the target compound was greater than or equal to 0.99, and the detection limit of the instrument method was less than or equal to 0.1 nmol/mol. A total of 115 NMVOCs were monitored, including 29 alkanes, 11 alkenes, 1 alkyne, 17 aromatic hydrocarbons, 35 halogenated hydrocarbons, 21 OVOCs and 1 sulfide (carbon disulfide).

147    CPF is described as a "new source identification tool", but papers from 2006 and 2007 are referenced. Is this technique new? Have there been new applications of this?

**Response:** Sorry for the confusing and thanks for suggestions.

CPF is not a new technology. CPF analysis was applied to show the relative location of potential emission sources by using the wind directions and source contributions calculated by PMF (Kim and Hopke, 2004). We have revised the CPF description and changed

the references in the relevant literature. (Line 155-158)

Line 155-158: The conditional probability function (CPF) is a source identification tool, which can be used to identify local emission sources of pollutants (Uria-Tellaetxe and Carslaw, 2014). CPF analysis methods were employed to determine the potential direction of emission sources by utilizing the wind directions and source contributions calculated through PMF (Kim and Hopke, 2004).

154    The MCM is no longer hosted at the University of Leeds and hasn't been for a while. I suggest you provide the new working link, hosted by the University of York (https://mcm.york.ac.uk/MCM/). (Line 160)

**Response:** Sorry for the mistake.

we have updated the link to the new working website hosted by the University of York (https://mcm.york.ac.uk/MCM/).

164    The authors should also describe how they have calculated $P(O_3)$ from the model, if this was not a direct output. If the authors are using a specific box model (e.g. DSMACC, AtChem2), please specify which one and include the appropriate references.

**Response:** Thanks for the suggestion.

In this study, the net production rate $O_3$ ($P(O_3)$) is the difference between the $O_3$ production (the oxidation of NO by $HO_2$ and $RO_2$) and $O_3$ destruction ($O_3$ photolysis, reactions of $O_3$ with OH and $HO_2$, reactions of OH with $NO_2$, and reactions of $O_3$ with alkenes). This method for estimating $O_3$ production and removal rates has been utilized in several previous studies (Wang et al., 2017;Wang et al., 2022). The constants (k) represent the rate coefficients of the respective reactions, as follows:

$$P(O_3) = [k_{HO_2+NO}[HO_2][NO] + \sum k_{RO_{2i}+NO}[RO_{2i}][NO] - k_{HO_2+O_3}[HO_2][O_3] - k_{OH+O_3}[OH][O_3] - k_{O(^1D)+H_2O}[O(^1D)][H_2O] - k_{OH+NO_2}[OH][NO_2] - k_{alkenes+O_3}[alkenes][O_3]$$

The details of this calculation method have been included in the revised manuscript along with the appropriate references. Thank you for highlighting this important aspect of our methodology. (Line 170-176)

We did not use a specific box model such as DSMACC or AtChem2 in our study. Our model is a 0-D box model incorporating the Master Chemical Mechanism version 3.3.1 (MCMv3.3.1). This model can simulate the concentration of radicals in the atmosphere and has been widely used to calculate the production of $O_3$, as well as budgets of radicals and intermediate species. Additionally, the

studies by Wang et al. (2022) and Guo et al. (2024) provide detailed descriptions of this model.

Line 172-178: In this model, the net production rate $O_3$ ($P(O_3)$) is the difference between the $O_3$ production (the oxidation of NO by $HO_2$ and $RO_2$) and $O_3$ destruction ($O_3$ photolysis, reactions of $O_3$ with OH and $HO_2$, reactions of OH with $NO_2$, and reactions of $O_3$ with alkenes). This method for estimating $O_3$ production and removal rates has been utilized in several previous studies (Wang et al., 2022;Guo et al., 2024). The constants (k) represent the rate coefficients of the respective reactions, as follows:

$$P\,(O_3) = [k_{HO_2+NO}[HO_2][NO] + \sum k_{RO_{2i}+NO}[RO_{2i}][NO] - k_{HO_2+O_3}[HO_2][O_3] - k_{OH+O_3}[OH][O_3] - k_{O(^1D)+H_2O}[O(^1D)][H_2O] - k_{OH+NO_2}[OH][NO_2] - k_{alkenes+O_3}[alkenes][O_3] \qquad (8)$$

170  Was any interpolation or averaging of the data performed to generate 1 h time resolution data for the model?

**Response:** Thanks for the suggestion.

All the species we monitored were at a 1 h time resolution. We did not interpolate or average the data, and the observation-based data we entered into the model were all at 1 h time resolution. At the same time, in the mode, we set the step length of the mode to be 1 hour, so the result of the mode output is also 1 hour time resolution.

171    Why was WS constrained in your box model? This parameter is not used by the

MCM. Was it used to calculate other factors in your model?

**Response:** Thanks for the comments.

WS was really constrained in our box model. In our model, we take into account the effect of horizontal transmission on pollution transmission, which makes our simulation results more accurate.

172    Please list all the measured VOCs, indicating which ones were selected, in a table. It would also be helpful to see average concentrations for these compounds. All of this should be presented in the supplementary.

**Response:** Thanks for the suggestion.

We have included a table (Table S1) in the supplementary material listing all the measured NMVOCs and indicating the selected compounds along with their average concentrations. We hope this additional information meets your requirements.

Table S1. Results of NMVOCs observed: concentrations with statistical analysis, ppbv.

| Groups | Species | Mean ± SD | Median | 25% | 75% | MCM v3.3.1 name |
|---|---|---|---|---|---|---|
| Alkanes | | | | | | |
| | Ethane | 3.52 ± 1.58 | 3.07 | 2.28 | 4.41 | C2H6 |
| | Propane | 1.68 ± 0.9 | 1.44 | 1.03 | 2.12 | C3H8 |
| | Isobutane | 0.9 ± 0.59 | 0.73 | 0.5 | 1.13 | IC4H10 |
| | n-butane | 1.1 ± 0.75 | 0.9 | 0.59 | 1.34 | NC4H10 |
| | Isopentane | 1.01 ± 0.64 | 0.84 | 0.6 | 1.23 | IC5H12 |

| | | | | | |
|---|---|---|---|---|---|
| | n-Pentane | 0.48 ± 0.39 | 0.38 | 0.26 | 0.55 | NC5H12 |
| | 2,2-Dimethylbutane | 0.02 ± 0.02 | 0.02 | 0.01 | 0.03 | M22C4 |
| | Cyclopentane | 0.08 ± 0.04 | 0.07 | 0.05 | 0.09 | |
| | 2,3-Dimethylbutane | 0.03 ± 0.02 | 0.03 | | 0.04 | M23C4 |
| | 2-Methylpentane | 0.17 ± 0.11 | 0.14 | 0.1 | 0.2 | M2PE |
| | 3-Methylpentane | 0.1 ± 0.09 | 0.08 | 0.06 | 0.12 | M3PE |
| | n-Hexane | 0.62 ± 0.36 | 0.56 | 0.38 | 0.77 | NC6H14 |
| | Methylcyclopentane | 0.04 ± 0.03 | 0.03 | 0.02 | 0.04 | |
| | 2,4-Dimethylpentane | 0.01 ± 0.01 | 0.01 | 0.01 | 0.01 | |
| | Cyclohexane | 0.03 ± 0.02 | 0.02 | 0.01 | 0.03 | CHEX |
| | 2-Methylhexane | 0.01 ± 0.01 | 0.01 | 0.01 | 0.02 | M2HEX |
| | 2,3-Dimethylpentane | 0.01 ± 0.01 | 0.01 | 0.01 | 0.01 | |
| | 3-Methylhexane | 0.01 ± 0.01 | 0.01 | 0.01 | 0.02 | M3HEX |
| | 2,2,4-Trimethylpentane | 0.02 ± 0.02 | 0.02 | 0.01 | 0.02 | |
| | n-Heptane | 0.01 ± 0.01 | 0.01 | 0.01 | 0.02 | NC7H16 |
| | Methylcyclohexane | 0.02 ± 0.02 | 0.02 | 0.01 | 0.03 | |
| | 2,3,4-Trimethylpentane | 0.01 ± 0.01 | 0.01 | 0.01 | 0.01 | |
| | 2-Methylheptane | 0.01 ± 0 | 0 | 0 | 0.01 | |
| | 3-Methylheptane | 0.01 ± 0 | 0.01 | 0 | 0.01 | |
| | Octane | 0.02 ± 0.02 | 0.01 | 0.01 | 0.02 | NC8H18 |
| | n-Nonane | 0.01 ± 0.01 | 0.01 | 0.01 | 0.01 | NC9H20 |
| | n-Decane | 0.01 ± 0 | 0.01 | 0.01 | 0.01 | NC10H22 |
| | Undecane | 0.01 ± 0 | 0.01 | 0.01 | 0.01 | NC11H24 |
| | Dodecane | 0.01 ± 0 | 0.01 | 0.01 | 0.01 | NC12H26 |
| Alkenes | | | | | | |
| | 1-Hexene | 0.13 ± 0.3 | 0.04 | 0.03 | 0.06 | HEX1ENE |
| | Ethylene | 0.89 ± 0.52 | 0.82 | 0.49 | 1.14 | C2H4 |
| | Propylene | 0.45 ± 0.57 | 0.22 | 0.12 | 0.66 | C3H6 |
| | trans-2-Butene | 0.03 ± 0.02 | 0.02 | 0.02 | 0.03 | TBUT2ENE |
| | 1-Butene | 0.05 ± 0.03 | 0.04 | 0.04 | 0.06 | BUT1ENE |
| | cis-2-Butene | 0.02 ± 0.01 | 0.02 | 0.01 | 0.03 | CBUT2ENE |
| | trans-2-Pentene | 0 ± 0.01 | 0 | 0 | 0 | TPENT2ENE |
| | 1-Pentene | 0.01 ± 0.01 | 0.01 | 0 | 0.01 | PENT1ENE |
| | cis-2-Pentene | 0.01 ± 0.01 | 0 | 0 | 0.01 | CPENT2ENE |
| | Isoprene | 0.38 ± 0.54 | 0.14 | 0.02 | 0.52 | C5H8 |
| | 1,3-Butadiene | 0.01 ± 0.01 | 0.01 | 0.01 | 0.02 | C4H6 |
| Alkynes | | | | | | |
| | Ethyne | 1.07 ± 0.58 | 0.95 | 0.63 | 1.42 | C2H2 |
| Aromatics | | | | | | |
| | Benzene | 0.29 ± 0.14 | 0.25 | 0.19 | 0.36 | BENZENE |
| | Toluene | 0.47 ± 0.33 | 0.36 | 0.25 | 0.57 | TOLUENE |
| | Ethylbenzene | 0.07 ± 0.05 | 0.05 | 0.03 | 0.08 | EBENZ |
| | m/p-Xylene | 0.14 ± 0.11 | 0.11 | 0.07 | 0.18 | MXYL |
| | Styrene | 0.01 ± 0.01 | 0.01 | 0 | 0.01 | STYRENE |

| | | | | | | |
|---|---|---|---|---|---|---|
| | o-Xylene | 0.07 ± 0.06 | 0.05 | 0.03 | 0.09 | OXYL |
| | Isopropyl benzene | 0 ± 0 | 0 | 0 | 0 | IPBENZ |
| | n-Propyl benzene | 0.01 ± 0 | 0.01 | 0 | 0.01 | PBENZ |
| | 3-Ethyltoluene | 0.01 ± 0.01 | 0.01 | 0.01 | 0.02 | METHTOL |
| | 4-Ethyltoluene | 0.01 ± 0.01 | 0.01 | 0 | 0.01 | PETHTOL |
| | 1,3,5-Trimethylbenzene | 0.01 ± 0 | 0.01 | 0 | 0.01 | TM135B |
| | 2-Ethyltoluene | 0.01 ± 0 | 0.01 | 0 | 0.01 | OETHTOL |
| | 1,2,4-Trimethylbenzene | 0.02 ± 0.02 | 0.02 | 0.01 | 0.03 | TM124B |
| | 1,2,3-Trimethylbenzene | 0.01 ± 0.01 | 0.01 | 0 | 0.01 | TM123B |
| | 1,3-Diethylbenzene | 0 ± 0 | 0 | 0 | 0 | |
| | 1,4-Diethylbenzene | 0 ± 0 | 0 | 0 | 0.01 | |
| | Naphthalene | 0.01 ± 0.01 | 0.01 | 0.01 | 0.02 | |
| Halohydrocarbons | | | | | | |
| | Freon12 | 0.13 ± 0.02 | 0.13 | 0.12 | 0.13 | |
| | Freon114 | 0.12 ± 0.02 | 0.12 | 0.1 | 0.14 | |
| | Chloromethane | 0.64 ± 0.17 | 0.61 | 0.52 | 0.73 | CH3CL |
| | Vinyl chloride | 0.02 ± 0.04 | 0.01 | 0.01 | 0.02 | VINCL |
| | Bromomethane | 0.01 ± 0 | 0.01 | 0.01 | 0.01 | CH3BR |
| | Chloroethane | 0.02 ± 0.01 | 0.02 | 0.02 | 0.03 | CH3CH2CL |
| | Freon11 | 0.21 ± 0.02 | 0.21 | 0.21 | 0.22 | |
| | 1,1-Dichloroethylene | 0 ± 0 | 0 | 0 | 0 | CCL2CH2 |
| | Freon113 | 0.07 ± 0.01 | 0.07 | 0.07 | 0.08 | |
| | Dichloromethane | 1.06 ± 1.2 | 0.89 | 0.67 | 1.23 | CH2CL2 |
| | 1,1-Dichloroethane | 0.08 ± 0.08 | 0.06 | 0.04 | 0.09 | CHCL2CH3 |
| | cis-1,2-Dichloroethylene | 0 ± 0 | 0 | 0 | 0 | CDICLETH |
| | Chloroform | 0.74 ± 0.55 | 0.58 | 0.35 | 0.95 | CHCL3 |
| | 1,1,1-Trichloroethane | 0 ± 0 | 0 | 0 | 0 | CH3CCL3 |
| | Tetrachloromethane | 0.09 ± 0.01 | 0.09 | 0.09 | 0.1 | |
| | 1,2-Dichloroethane | 0.51 ± 0.33 | 0.42 | 0.31 | 0.62 | CH2CLCH2CL |
| | Trichloroethylene | 0.01 ± 0 | 0.01 | 0 | 0.01 | TRICLETH |
| | 1,2-Dichloropropane | 0.19 ± 0.16 | 0.15 | 0.11 | 0.22 | CL12PROP |
| | Bromodichloromethane | 0 ± 0 | 0 | 0 | 0 | |
| | trans-1,3-Dichloropropene | 0.01 ± 0.01 | 0.01 | 0 | 0.01 | |
| | cis-1,3-Dichloropropene | 0 ± 0 | 0 | 0 | 0 | |
| | 1,1,2-Trichloroethane | 0.03 ± 0.03 | 0.03 | 0.02 | 0.04 | CH2CLCHCL2 |
| | Tetrachloroethylene | 0.32 ± 0.36 | 0.21 | 0.1 | 0.41 | TCE |
| | 1,2-Dibromoethane | 0 ± 0 | 0 | 0 | 0 | DIBRET |
| | Chlorobenzene | 0.01 ± 0 | 0.01 | 0.01 | 0.01 | |

| | | | | | | |
|---|---|---|---|---|---|---|
| | Bromoform | 0 ± 0 | 0 | 0 | 0 | |
| | 1,1,2,2-Tetrachloroethane | 0 ± 0 | 0 | 0 | 0 | CHCL2CHCL2 |
| | 1,3-Dichlorobenzene | 0 ± 0 | 0 | 0 | 0 | |
| | 1,2-Dichlorobenzene | 0 ± 0 | 0 | 0 | 0 | |
| | trans-1,2-Dichloroethylene | 0.01 ± 0 | 0 | 0 | 0.01 | |
| | Dibromochloromethane | 0 ± 0 | 0 | 0 | 0 | |
| | 1,4-Dichlorobenzene | 0.02 ± 0.01 | 0.02 | 0.01 | 0.02 | |
| | Benzyl chloride | 0 ± 0 | 0 | 0 | 0 | |
| | 1,2,4-Trichlorobenzene | 0 ± 0 | 0 | 0 | 0 | |
| | Hexachloro-1,3-butadiene | 0.01 ± 0 | 0.01 | 0.01 | 0.01 | |
| OVOCs | | | | | | |
| | acetaldehyde | 0.2 ± 0.15 | 0.16 | 0.1 | 0.24 | CH3CHO |
| | n-butyraldehyde | 0.07 ± 0.1 | 0.05 | 0.04 | 0.06 | |
| | 1,4-Dioxane | 0.01 ± 0 | 0.01 | 0 | 0.01 | |
| | Acrolein | 0.14 ± 0.07 | 0.12 | 0.09 | 0.16 | ACR |
| | propanal | 0.33 ± 0.1 | 0.33 | 0.26 | 0.4 | C2H5CHO |
| | Acetone | 2.74 ± 0.7 | 2.78 | 2.25 | 3.2 | CH3COCH3 |
| | MTBE | 0.04 ± 0.04 | 0.03 | 0.02 | 0.04 | MTBE |
| | methylacrolein | 0.1 ± 0.09 | 0.08 | 0.06 | 0.12 | MACR |
| | valeraldehyde | 0.05 ± 0.02 | 0.05 | 0.04 | 0.06 | C4H9CHO |
| | caproaldehyde | 0.08 ± 0.06 | 0.07 | 0.02 | 0.11 | |
| | Isopropanol | 0.02 ± 0.02 | 0.02 | 0.01 | 0.03 | IPROPOL |
| | Vinyl acetate | 0 ± 0 | 0 | 0 | 0.01 | |
| | Ethyl acetate | 0.24 ± 0.2 | 0.19 | 0.13 | 0.3 | ETHACET |
| | 2-Butanone | 0.24 ± 0.1 | 0.21 | 0.17 | 0.29 | MEK |
| | Tetrahydrofuran | 0.03 ± 0.02 | 0.02 | 0.01 | 0.03 | |
| | Methyl methacrylate | 0.01 ± 0 | 0.01 | 0 | 0.01 | |
| | 4-Methyl-2-pentanone | 0.01 ± 0.01 | 0.01 | 0.01 | 0.02 | MIBK |
| | 2-Hexanone | 0.22 ± 0.22 | 0.19 | 0.03 | 0.32 | HEX2ONE |
| | Crotonaldehyde | 0.02 ± 0.02 | 0.02 | 0.01 | 0.03 | |
| | benzaldehyde | 0.01 ± 0 | 0.01 | 0.01 | 0.01 | |
| | m-methylbenzaldehyde | 0 ± 0 | 0 | 0 | 0 | |
| Sulfide | | | | | | |
| | Carbon disulfide | 0.13 ± 0.22 | 0.06 | 0.03 | 0.12 | |

173    What are the default values for $j(H_2O_2)$ and $j(O_1D)$ and planetary

boundary layer? Do you think these are representative of the site?

Have any dilution rates been applied to compounds generated in the

model, and if so, are these varied based on the default PBLH? The authors need to highlight why these values were used. If measured or modelled values cannot be used, it is important to discuss the uncertainties associated with using default values for your results.

**Response:** Thanks for the suggestion.

We acknowledge the error in our manuscript where we incorrectly referred to $J(NO_2)$ instead of $J(O_1D)$. We have since made the necessary corrections in the revised version of the manuscript.

In this study, the values of PBL, $J(H_2O_2)$, and $J(NO_2)$ we use are the results obtained from the model simulation. Although we did not have actual observational data, setting default values still helps improve the accuracy of our simulations. Compared to results without default values, the results with default values are more accurate and reliable. As mentioned in section 2.5 of the manuscript, we assessed the reliability of the simulation results by simulating the concentration of $O_3$. We calculated the index of agreement (IOA) to be 0.8. Since the real atmospheric boundary layer is variable, changes in the boundary layer can lead to variations in pollutant concentrations. An increase in the boundary layer height results in dilution of pollutant concentrations, and the dilution ratio in the model is based on changes in the planetary boundary layer.

180 Please provide more context for why a value of 0.6 indicates the model is performing well. What is the acceptable range for this value to be? Which compound did you evaluate the performance of the model using? What was constrained in the model when you did this, and what was the model left to calculate. Much more detail is required here.

**Response:** Thanks for the suggestion.

The performance of the OBM was evaluated by applying the index of agreement (IOA). The calculation formula is as follows:

$$IOA = 1 - \frac{\sum_{i=1}^{n} (O_i - M_i)^2}{\sum_{i=1}^{n} (|O_i - \overline{O}| + |M_i - \overline{O}|)^2}$$

We evaluate the reliability of the model simulation results by simulating the concentration of $O_3$ in the atmosphere and calculating the IOA value. In the model, we do not input observations of $O_3$, but by input the concentrations of trace gases ($SO_2$, CO, and NO) and 75 NMVOCs, and meteorological parameters (T, RH, and WS) to simulate the concentration of $O_3$ in the atmospheric environment. Finally, the IOA value is calculated from the simulated and observed values.

We have consulted a lot of literature and made a comparative analysis with the IOA value in the literature. In many studies, when IOA

ranges from 0.68 to 0.89, the simulation results are reasonable (Wang et al., 2018;Liu et al., 2022). We calculated an IOA value of 0.6 using data from all days. In fact, $O_3$ is not produced at night, so we recalculated the IOA value from 7:00 to 19:00 during the day and obtained a result of 0.8. Therefore, the results simulated by our model are reliable. At the same time, we also modified 0.6 to 0.8 in the manuscript. (Line 197-203)

Line 197-203: In various studies, model simulation results are often considered acceptable when the value of IOA falls within the range of 0.68 to 0.89 (Wang et al., 2018). To evaluate the reliability of our model simulations, we conducted an analysis of $O_3$ concentration in the atmosphere and calculated the IOA value. Our model does not directly incorporate $O_3$ observations. Instead, it utilizes concentrations of trace gases ($SO_2$, CO, and NO) and 75 NMVOCs, and meteorological parameters (T, RH, and WS) to simulate the concentration of $O_3$ in the atmospheric environment. The IOA values for $O_3$ was calculated from 7:00 to 19:00 during the day and obtained a result of 0.8. Therefore, the results simulated by our model are reliable.

189    What is meant by "short pollution process"? Do you mean that the pollution event was too short, and you only looked at periods where pollution remained high for at least a week for better data coverage?

Please clarify.

**Response:** We apologize for the confusion caused by our inappropriate description.

Our observations lasted 30 days. The $O_3$ pollution lasted for 10 days from June 8 to 17, and for 8 days from June 20 to 27. As you said, for better data coverage, we only discussed periods of $O_3$ pollution that lasted at least a week, and processes with relatively few days of pollution were not discussed. In the manuscript, we have also made modifications accordingly. (Line 210-212)

Line 210-212: Meanwhile, there were also $O_3$ pollution events on 6[th] Jun. and 29[th]-30[th] Jun. However, for better data coverage, we only discussed periods of $O_3$ pollution that lasted at least a week, and processes with relatively few days of pollution were not discussed in this study.

224 This is a very large proportion of halocarbons. It would be really interesting to see which VOCs are included in each category in your study, and what the average mixing ratios were across the sampling period. Please include this as an expanded version of Table 1 in the supplementary.

**Response:** Thank you for this very thoughtful suggestion. We have shown all the observed NMVOCs and their related statistics in

Table S1.

231    Remove "if".

**Response:** Thanks for the suggestion. It has been deleted.

236    Explain what you mean by the "top fifteen VOC species". Do you mean the highest average mixing ratio? If so, please replace "top" with something like "the fifteen VOCs with the highest average mixing ratio across the observation period."

**Response:** Sorry for this mistake.

What we mean is the 15 NMVOCs with the highest average mixing ratio across the observation period. According to your request, we have replaced the "Comparisons of the top fifteen NMVOCs during different processes, ppbv. Error bars are standard deviations." to "Comparisons of the fifteen NMVOCs with the highest average mixing ratio during different processes, ppbv. Error bars are standard deviations". (Line 293-294).

Line 293-294: **Figure 2.** Comparisons of the fifteen NMVOCs with the highest average mixing ratio during different processes, ppbv. Error bars are standard deviations.

253    "As illustrated in the following PMF source apportionment". Where is this? I cannot see any source apportionment in this section,

just some discussion of the possible sources of different VOCs from the literature.

**Response:** We apologize for the confusion caused by our inappropriate description.

Here our preliminary analysis found that vehicle exhaust, solvent use, combustion, biogenic emission, and industrial processes are important sources of NMVOCs at observation sites. In section 3.2.2, we used PMF to analyze the sources of NMVOCs and reached a similar conclusion. We try to express that the analysis here is consistent with the analysis in section 3.2.2.

Therefore, we have replaced the "as also illustrated in the following PMF source apportionment." to "as also illustrated in the following PMF source apportionment (in section 3.2.2)". (Line 289-291).

Line 289-291: Therefore, vehicle exhaust, solvent use, combustion, biogenic emission, and industrial processes are important sources of NMVOCs at observation sites, as also illustrated in the following PMF source apportionment (in section 3.2.2).

255    I don't understand what Figure 2 shows. What does the line represent and what are the bars? What are the authors trying to say using this figure?

**Response:** Thank you for your feedback.

In our study, the bars in Fig. 2 represent the concentration of each species, while the line represents the cumulative percentage of each species' concentration in the total volatile organic compounds (TNMVOCs) concentration. One of our aims is to show the top fifteen substances by concentration, as they have high concentrations and large percentages, making them key substances of interest. Additionally, by calculating the percentage of the top 15 NMVOCs in TNMVOCs, we aim to express the importance of these NMVOCs.

260    "Major" VOCs – why exactly were these selected? You could either just say "a selection of VOCs", or if you are pointing out that these are the major ones, you need to explain why.

**Response:** Thanks for the suggestion.

We have replaced the "Major VOCs" to "A selection of NMVOCs". (Line 298).

Line 298: A selection of NMVOCs, $O_3$, and $NO_x$ were selected, and their daily changes were analyzed, as shown in Fig. S4.

264    "photochemical reaction" – change to "photochemical reactivity", as there are multiple photochemical reactions happening.

**Response:** Thanks for the suggestion.

We have replaced the "photochemical reaction" to "photochemical reactivity". (Line 302 and Line 305).

Line 302: Higher $O_3$ production during the day indicates a strong photochemical reactivity.

Line 305: This is associated with a higher boundary layer and strong photochemical reactivity during the day.

271  Explain what you mean by "the daily change was consistent with $NO_x$".

**Response:** Thanks for the suggestion.

As shown in Fig. S4 of supplementary materials, the concentrations of ethane, propane, butane, pentane, ethylene, propylene, benzene, and toluene exhibit a bimodal diurnal pattern. Similarly, the diurnal variation of $NO_x$ concentration also displays a bimodal trend. Furthermore, the peak times of these NMVOCs and $NO_x$ concentrations align relatively closely. Therefore, in line 276, we state, 'Additionally, the peak concentrations of these NMVOCs occur in the morning and evening (7:00-8:00 and 23:00-24:00), showing a consistent daily pattern with $NO_x$.

We have carefully considered your suggestion and have made the necessary revisions in the manuscript. Specifically, we have amended the sentence as follows: In addition, the peak

concentrations of these NMVOCs were observed in the morning and evening (7:00-8:00 and 23:00-24:00), showing a consistent daily pattern with $NO_x$. This suggests that the emissions of these NMVOCs are significantly influenced by motor vehicle emissions and fuel combustion. (Line 307-309)

Line 307-309: In addition, the peak concentrations of these NMVOCs were observed in the morning and evening (7:00-8:00 and 23:00-24:00), showing a consistent daily pattern with $NO_x$.

273     What are "traditional nighttime activities"?

**Response:** Thanks for the suggestion.

In this context, the "traditional nocturnal activities of the city" refer to the typical or customary activities that take place at night. This could include increased traffic from people eating out, attending events or socializing. At the same time, due to the restrictions on large trucks during the day, many large trucks will travel intensively at night, resulting in higher levels of NMVOCs and $NO_x$ emissions.

279     Acetone is a "common VOC" – what does this mean? Consider changing the language.

**Response:** Sorry for the mistake.

We have replaced the "Acetone is a common NMVOCs and comes

from a wide range of sources" to "Acetone comes from a wide range of sources". (Line 318).

Line 318: Acetone comes from a wide range of sources, mainly from vehicle emissions, industrial production, and secondary formation.

297    Could it be that the pentanes come from a mix of sources? The text implies that for Case 1, Case 2 and clean days, the sources are exclusively NG emissions, Vehicular exhaust emissions, and liquid gasoline emissions respectively.

**Response:** Thanks for the suggestion.

Yes, it is possible that the pentanes come from a mix of sources. For example, pentanes can also be found in industrial emissions, evaporative emissions from fuel storage and handling, as well as natural sources such as vegetation and wildfires.

However, previous studies have tended to preliminarily determine the emission source of pentanes by calculating the ratios of isopentane/n-pentane. There are some differences in values of isopentane/n-pentane among natural gas emissions (NG, 0.8-0.9), vehicle emissions (2.2-3.8), liquid gasoline (1.5-3.0), and fuel evaporation (1.8-4.6) (Gilman et al., 2013;McGaughey et al., 2004;Watson et al., 2001). Therefore, the ratio method was used in

this study to calculate the ratios of isopentane/n-pentane, so as to preliminarily determine the source of pentane.

In this study, the ratios of Case 1, Case 2, and clean days were 0.7, 2.5, and 1.1, respectively. It suggests that isopentane and n-pentane may come from NG emissions, vehicular exhaust, and liquid gasoline, respectively.

311  Was this value 0.5 for all three cases? How does this ratio vary with wind speed? Perhaps this could further support the argument of E/X rations indicating that air masses were affected by transport?

**Response:** Thank you for your valuable suggestion.

Yes, in our study, this ratio was 0.5 for all three cases. Exploring the relationship between wind speed and the E/X ratios can provide additional evidence supporting our argument that air masses were influenced by transport. According to your requirements, we have analyzed the relationship between ethylbenzene, m/p-Xylene, E/X, and wind direction and speed. As shown in Fig. S5, the concentrations of ethylbenzene and m/p-Xylene are mainly influenced by winds coming from the northwest, and their concentrations tend to increase with stronger wind speeds. Similarly, E/X also exhibits similar patterns of variation. This further indicates that the regional transport of ethylbenzene and

m/p-Xylene from distant sources. We have included this analysis in our revised manuscript to strengthen our conclusions. (Line 351-354)

[Figure]

Figure S5. The rose diagrams of transport contributions from polluted sources for Ethylbenzene and m/p-Xylene which locate indifferent directions.

Line 351-354: As shown in Fig. S5, the concentrations of ethylbenzene and m/p-Xylene are mainly influenced by winds coming from the northwest, and their concentrations tend to increase with stronger wind speeds. Similarly, E/X also exhibits similar patterns of variation. This further indicates that the regional transport of ethylbenzene and m/p-Xylene from distant sources.

315 Generally for this section, more discussion of the potential locations of some of these sources would be good to see. There are two solvent factors. Why is this? Different industrial sources? Are there any industrial sources in the northeast and southeast of the site? For factor 7, are there any industrial sites near the observation site? Are the source apportionment factors consistent with the

geography of the site?

**Response:** Thank you for your comments.

In fact, we tried solutions with different factors. As shown in Fig. S2, we explored the number of PMF factors from 3 to 12 to obtain the best solution. Each model is run 20 times. The values of $Q_{true}/Q_{expected}$ in different solutions are discussed subsequently. Fpeak values from −2 to 2 are used in the model. We find that when the number of factors increases from 3 to 7, the change in the values of $Q_{true}/Q_{expected}$ is relatively stable. However, when the number of factors is increased to 8, the values of $Q_{true}/Q_{expected}$ fluctuate significantly. We also find that when Fpeak is 0, the values of $Q_{true}/Q_{expected}$ are lowest. Finally, we adopted a 7-factor solution ($Q_{true}/Q_{expected} = 3.42$; and Fpeak = 0). Therefore, choosing 7-factor solution is the most appropriate choice.

However, we found that substances contributing significantly to Factor 6 and Factor 7 are often used as organic solvents. We also consulted the literature and found similar occurrences in previous studies. Xu et al. (2023) used the PMF model to analyze the main sources of NMVOCs in the Yangtze River Delta region and identified two solvent usage sources. It is worth noting that the key contributing species in the two solvent usage sources in the Yangtze

River Delta region are quite consistent with our study. In their research, factor 3 and factor 6 were determined as solvent usage-toluene and solvent usage-C8 aromatics, respectively. Factor 3 was characterized by C5-C7 alkanes and toluene, while factor 6 was characterized by ethylbenzene, m/p-xylene, and o-xylene. As shown in Fig. 4 of our study, factor 6 and factor 7 were identified as solvent usage 1 and solvent usage 2. Factor 6 was characterized by chloromethane, dichloromethane, tetrachloromethane, 1,2-dichloroethane, 1,2-dichloropropane, and ethyl acetate. Although these species did not appear in Wang et al.'s study, it is evident that these substances are common organic solvents in industrial applications. Factor 7 was characterized by a high percentage of methylcyclopentane, cyclohexane, TEXs (Toluene, Ethylbenzene, m/p- Xylene, and o-Xylene), 1,2-Dichloroethane, 1,2-Dichloropropane, and Ethyl acetate. As we know, TEXs are related to the use of solvent cleaners in coatings, paintings, synthetic fragrances, adhesives, and solvents (Zhang et al., 2021). 1,2-Dichloroethane, 1,2-Dichloropropane, and Ethyl acetate are also commonly used scientific chemical reagents.

In addition, other studies by various scholars have also identified two solvent usage sources (Zhang et al., 2023;Wang et al., 2020;Song et al., 2019;Lyu et al., 2016).

Figure 1 shows the distribution of industrial sites around the observation points. It can be seen that there are a large number of industrial sites in Zhengzhou city. There are many industrial productions to the southeast and northeast of the observation points. We also found that there are indeed industrial facilities around Factor 7. We have carefully evaluated the source apportionment factors and believe that they are consistent with the geographical characteristics of the study site.

[Figure]

Figure 1. Distribution of industrial sites around observation sites.

369 This is really interesting. Are you saying that $O_3$ pollution events appear to be independent of the VOC sources, and these remain broadly similar? If this is the case, then what is driving these $O_3$ pollution events. Is it the meteorology? Please explain this in the manuscript

**Response:** Thank you for your comments.

In this study, the contributions of various pollution sources show relatively minor differences between $O_3$ pollution events and clean days, but there are still some distinctions. For instance, compared to clean days, in Case 1 events, industrial production, biogenic emission, LPG/Ng, and solvent usage increased by 2%, 3%, 2%, and 2% respectively. Compared to clean days, in Case 2 events, solvent usage, biogenic emission, and LPG/Ng increased by 3%, 5%, and 16% respectively. Therefore, the increased contributions of solvent usage, biogenic emission, and LPG/Ng may have a certain impact on the formation of $O_3$ pollution.

The reasons for $O_3$ formation in the area may involve various complex factors, including but not limited to emissions of nitrogen oxides, levels of solar radiation, meteorological conditions, etc.

First, we compared the average concentrations of nitrogen oxides in Case 1, Case 2, and clean days. The average concentrations of $NO_2$ in Case 1, Case 2, and clean days were 27.4 ± 19.5, 24.9 ± 12.3, and 24.4 ± 16.1 ppbv, respectively, while the average concentrations of NO were 3.9 ± 3.6, 3.9 ± 2.4, and 4.8 ± 5.5 ppbv, respectively. The average concentrations of $NO_2$ in pollution events were higher than those in clean days, while the average concentrations of NO were lower than those in clean days. Higher concentration of $NO_2$ can promote the formation of $O_3$, while the

titration reaction between NO and $O_3$ consumes $O_3$ (Sillman, 1999). Therefore, the higher concentration of $NO_2$ and lower concentration of NO during pollution events are one of the reasons for the occurrence of $O_3$ pollution events.

Second, we further explored the relationship between meteorology and $O_3$ concentration. According to Fig. S3a and Fig. S3b, it can be observed that $O_3$ concentration shows a linear increasing trend with temperature and a linear decreasing trend with RH. $O_3$ has a significant correlation with temperature and RH, with correlation coefficients of 0.7 and -0.61 respectively. Therefore, conditions of high temperature and low RH are more conducive to $O_3$ pollution. Figure S3c indicates that $O_3$ concentration exceeding the secondary standard mainly occurs under meteorological conditions of high temperature (greater than 30 °C) and low RH (less than 55%). It can be noted that when 35 °C < T < 40 °C and 20% < RH < 40%, the $O_3$ concentration consistently exceeds the secondary standard. Meng et al. (2023) argued that most of the reactions involved in $O_3$ formation increase with temperature, and the rate of $O_3$ production exceeds that of $O_3$ loss by a large margin.

In conclusion, in addition to the impact of solvent usage, biogenic emission, and LPG/Ng on $O_3$ pollution events, meteorological

factors are also significant factors in the occurrence of $O_3$ pollution events.

In addition, we have added an analysis of the correlation between $O_3$ and temperature and RH in the manuscript. (Line 235-253)

[Figure]

Figure S3. Correlation analysis of $O_3$, T, and RH.

Line 235-253: The average concentrations of TNMVOCs, $NO_2$, $PM_{10}$, and $PM_{2.5}$ on clean days were lower than those of the $O_3$ pollution events. The average RH (65 ± 17%) on clean days was higher than those during Case 1 and Case 2 events, while the average temperature (26.0 ± 4.8 °C) was lower than those during Case 1 and Case 2 events. According to the analysis in Fig. S3a and Fig. S3b, $O_3$ has a significant correlation with temperature and RH, with correlation coefficients of 0.7 and -0.61 respectively. Therefore, conditions of high temperature and low RH are more conducive to $O_3$ pollution. Fig. S3c indicates that $O_3$ concentration exceeding the secondary standard mainly occurs under meteorological conditions of high temperature (greater than 30 °C)

and low RH (less than 55%). It can be noted that when 35 °C < T < 40 °C and 20% < RH < 40%, the $O_3$ concentration consistently exceeds the grade II threshold of the NAAQS-2012. High temperature and low RH are more conducive to $O_3$ pollution(Chen et al., 2020;Zhang et al., 2015). Meng et al. (2023) argued that most of the reactions involved in $O_3$ formation increase with temperature, and the rate of $O_3$ production exceeds that of $O_3$ loss by a large margin. Therefore, during the study period, the meteorological conditions of high temperature and low RH are also important factors affecting the occurrence of $O_3$ pollution.

Besides, the average concentration of $NO_2$ in clean days (24.4 ± 16.1 ppbv) was lower than that in Case 1 and Case 2, while the average concentration of NO in clean days (4.8 ± 5.5 ppbv) was higher than that in Case 1 (3.9 ± 3.75 ppbv) and Case 2 (3.9 ± 2.4 ppbv). Higher concentration of $NO_2$ can promote the formation of $O_3$, while the titration reaction between NO and $O_3$ consumes $O_3$ (Sillman, 1999). Therefore, the higher concentration of $NO_2$ and lower concentration of NO during pollution events are one of the reasons for the occurrence of $O_3$ pollution events.

383   This is the wrong way around - $O_3$ formation was more sensitive to biogenic emissions, not "biogenic emission was more sensitive to $O_3$ formation".

**Response:** Sorry for this mistake.

We have replaced the "indicating that biogenic emission was more sensitive to $O_3$ formation" to "indicating that $O_3$ formation was more sensitive to biogenic emissions". (Line 451-452).

Line 451-452: During the entire period, especially in the pollution events, the RIR of AVOCs was lower than that of BVOCs, indicating that $O_3$ formation was more sensitive to biogenic emissions.

385  From Fig. 6, the RIRs look broadly similar between Case 1, Case 2, and clean. In the text, the authors have suggested there is a difference between clean and polluted days. Could you please quantify what this difference is? For example, what is the percentage difference in RIRs during polluted days, and is this significant?

**Response:** We appreciate the reviewer's insightful feedback.

As you pointed out, the RIRs appear broadly similar between Case 1, Case 2, and clean days. Due to the focus of this study on the summer month of June, characterized by strong sunlight, high temperatures, and low RH, conditions were highly conducive to photochemical reactions. As introduced in section 3.1.1, the concentrations of TNMVOCs during pollution events and clean

days were relatively low and showed minimal variations. Additionally, there were no significant changes in temperature and RH. Therefore, when analyzing $O_3$ sensitivity, the RIR values of various species in each process are relatively close, and the percentage differences in RIRs were not significant.

To address the difference between clean and polluted days in terms of RIRs, we have quantified this in detail. The percentage difference in RIRs during polluted days compared to clean days has been calculated. More analysis has been provided in the revised manuscript to clarify this aspect more effectively. (Line 446-448)

Line 446-448: Compared to clean days, the RIR value of AVOCs decreased by 11%, with Aromatics showing the largest decrease (26%), while Alkanes and Alkenes increased by 7% and 3% respectively. In pollution events, CO and $NO_x$ were reduced by 29% and 22%, respectively.

408    Please clarify which data you have used to generate the Isopleth in Fig. 7. Have you used the average peak $O_3$ across the entire observation period? Or is this the average daytime $O_3$ between particular hours? There is a point for "base case", but it would be interesting to see where Case 1, Case 2 and clean days would appear on this isopleth.

**Response:** We appreciate the reviewer's feedback.

Firstly, in OBM, we designed a total of 36 sets of simulated scenarios. The concentration of AVOCs decreases in 20% increments, as does the concentration of $NO_x$. In the Isopleth plot, the 'base case' represents scenarios where the concentrations of AVOCs and $NO_x$ are not reduced. The horizontal and vertical axes represent the proportions of the concentrations of AVOCs and $NO_x$ to their concentrations in the unreduced state, respectively. The isopleth represents the average $O_3$ concentration over the entire observation period. The isopleth represents the average $O_3$ concentration from 10 a.m. to 4 p.m. each day throughout the entire observation period. To create the $O_3$ EKMA curve, the data points of the maximum $O_3$ concentration generated by simulation between 10:00 and 16:00 are linearly fitted into colored surfaces. The data points of the same color on the colored surface are then connected to form the $O_3$ EKMA curve.

To see the positions of Case 1, Case 2, and clean days on the isopleth, we have redrawn an EKMA curve (Fig. S7). The horizontal and vertical axes represent AVOCs and $NO_x$ concentrations under different reduction scenarios, respectively, while the isopleth indicates the maximum $O_3$ concentration value in each scenario group. We have replaced Fig. 7 in the manuscript.

As can be seen from the Fig. 7, Case 1, Case 2, and clean days are all above the ridge line, indicating that the two pollution events and clean days are all in the AVOCs control area.

we have provided additional information on the positions of Case 1, Case 2, and clean days on the isopleth in the revised version of the manuscript. Thank you for highlighting this point. (Line 483-485)

[Figure]

Figure S7. EKMA curves of the $O_3$ max concentration.

Line 483-485: At the same time, Case 1, Case 2, and clean days are all above the ridgeline and belong to the AVOCs control region (Fig. S7). Therefore, reducing AVOCs can effectively reduce the generation of $O_3$.

451     This seems quite general, after the detail you have gone into earlier on the RIRs for VOC groups and looking closely at different

sources. I understand why biogenic compounds have not been included, but the impact of these should be further acknowledged in the text as changes to biogenic emissions present a large uncertainty to how $O_3$ production might change in the future. It would have been interesting to see how $O_3$ production might change when the different sources are reduced alongside $NO_x$. This is more useful to policy than just investigating different percentage changes in bulk AVOCs, as it would be challenging to reduce AVOC emissions in this way.

Response: Thanks for the valuable suggestion.

In section 3.3.1 of the manuscript, we have further acknowledged the impact of biogenic emissions on $O_3$ production. Through comparisons with studies from other cities, we have also observed the significance of biogenic emissions on urban $O_3$ production. (Line 449-459).

We acknowledge the importance of exploring the effects of reducing emissions from different sources in conjunction with nitrogen oxides on $O_3$ production. While we recognize the significance of this investigation, we agree that our current manuscript does not provide a clear and sufficient exploration of this aspect. Therefore, we plan to further elaborate on this topic in

future research to address your valuable suggestion.

Line 446-459: Isoprene was the sole BVOC considered in this study. Isoprene is an important tracer to indicate biogenic emissions (Xie et al., 2021;Li et al., 2024;Qin et al., 2023). During the entire period, especially in the pollution events, the RIR of AVOCs was lower than that of BVOCs, indicating that $O_3$ formation was more sensitive to biogenic emissions. This may be due to increased emissions of BVOCs at higher temperatures and solar radiation conditions, as well as their high reactivity and $O_3$ formation potential. Studies in Yucheng (Zong et al., 2018), Leshan (Xie et al., 2021), and and Nanjing (Fan et al., 2021;Ming et al., 2020) have shown that $O_3$ is highly sensitive to BVOCs. Studies in Zhengzhou (Wang et al., 2022), Hangzhou (Zhao et al., 2020), and Hong Kong (Wang et al., 2017) suggested that $O_3$ exhibits greater sensitivity to BVOCs than AVOCS during hot seasons. Wang et al. (2019) found in their study on $O_3$ source apportionment in Henan Province, where Zhengzhou is located, that BVOCs contribute to approximately 23.9% of the $O_3$ attributed to NMVOCs. Therefore, the contribution of BVOCs to $O_3$ is very important.

441   You say $P(O_3)$ increases respectively, but what is this respective to? Is this per hour?

**Response:** We apologize for the confusion caused by our inappropriate description.

First, we modified the term 'P(O$_3$)' to 'O$_3$ concentration' to represent the change in O$_3$ concentration with the reduction of AVOCs and NO$_x$ emissions. We also adjusted the vertical axes in Fig. 8 and Fig. S8.

Second, we have replaced the "In this case, P(O$_3$) increases by 30, 21, 16, 13, 13, 15, and 15% respectively" to "In this case, O$_3$ concentration increased by 30, 21, 16, 13, 13, 15, and 15% from 10 a.m. to 4 p.m.". (Line 519).

Line 519: In this case, O$_3$ concentration increased by 30, 21, 16, 13, 13, 15, and 15% from 10 a.m. to 4 p.m.

453   The authors could work on bringing all of these results together in the conclusion, rather than just summarising the key result of each section in turn. What is driving O$_3$ production? Is it related to sources? Is it the meteorology?

**Response:** Thank you for the valuable feedback on our study.

In the revised manuscript, we have integrated all results in the conclusion section, rather than just summarizing the key findings of each section separately. we have enhanced the discussion in the revised manuscript to elucidate the interplay between O$_3$

production and meteorological conditions. Finally, we found that the occurrence of $O_3$ pollution processes is related to the increase of pollutant concentration, the change of emission sources, and adverse meteorological factors. These revisions aim to provide a more thorough understanding of the mechanisms governing $O_3$ production. (Line 532-551)

Line 532-551: The summer $O_3$ pollution has always been an important environmental issue in Zhengzhou. This study investigated the characteristics and emission sources of $O_3$ precursors from 1st to 30th June 2023. The OBM was used to analyze the influence of precursors on the formation of $O_3$, and the emission reduction strategy of precursors was proposed to control the concentration of $O_3$. During the entire period, the concentration of TNMVOCs varied from 9.9 to 60.3 ppbv, with an average value of 22.9 $\pm$ 8.3 ppbv. The average concentration of TNMVOCs during $O_3$ pollution was higher than that during clean days. Alkanes (44%), OVOCs (20%), and halocarbons (19%) were the most abundant NMVOCs group. Ethane, acetone, and propane were always the most abundant species. The average concentrations of $NO_2$ in pollution events were higher than those in clean days, while the average concentrations of NO were lower than those in clean days. Therefore, the increasing concentration of $O_3$ precursors is

one of the reasons for the formation of $O_3$ pollution. At the same time, the unfavorable meteorological conditions of high temperature and low RH in the observation process are also important factors in the formation of $O_3$ pollution. Further analysis of the source of these precursors found that Vehicular exhaust (28%), solvent usage (27%), and industrial production (22%) were the main emission sources of NMVOCs. The increase of solvent usage, biogenic emission and LPN/NG contribution is an important cause of $O_3$ pollution. Sensitivity analysis of $O_3$ to precursors found that NMVOCs had the highest RIR value, while $NO_x$ had a negative RIR value. Alkenes have the highest RIR value among AVOCs. It should be noted that the RIR value of BVOCs was greater than that of AVOCs. The local $O_3$ formations were in the AVOCs-limited regimes, which means reducing the concentration of AVOCs was an effective way to reduce $O_3$ concentration. Meanwhile, we suggest that the minimum reduction ratio of AVOCs/$NO_x$ should be no less than 3:1 to reduce $O_3$ production.

465     Olefins mentioned here, earlier referred to as alkenes. Please change to alkenes.

  **Response:** Sorry for this mistake.

  It has been corrected.

**Reference:**

Chen, L., Zhu, J., Liao, H., Yang, Y., and Yue, X.: Meteorological influences on $PM_{2.5}$ and $O_3$ trends and associated health burden since China's clean air actions, Sci. Total. Environ., 744, 140837, https://doi.org/10.1016/j.scitotenv.2020.140837, 2020.

Fan, M., Zhang, Y., Lin, Y., Li, L., Xie, F., Hu, J., Mozaffar, A., and Cao, F.: Source apportionments of atmospheric volatile organic compounds in Nanjing, China during high ozone pollution season, Chemosphere, 263, https://doi.org/10.1016/j.chemosphere.2020.128025, 2021.

Gilman, J. B., Lerner, B. M., Kuster, W. C., and de Gouw, J. A.: Source Signature of Volatile Organic Compounds from Oil and Natural Gas Operations in Northeastern Colorado, Environ. Sci. Technol., 47, 1297-1305, https://doi.org/10.1021/es304119a, 2013.

Guo, J., Xu, Q., Yu, S., Zhao, B., and Zhang, M.: Investigation of atmospheric VOCs sources and ozone formation sensitivity during epidemic closure and control: A case study of Zhengzhou, Atmos. Pollut. Res., 15, https://doi.org/10.1016/j.apr.2023.102035, 2024.

Kim, E., and Hopke, P.: Comparison between conditional probability function and nonparametric regression for fine particle source directions, Atmos. Environ., 38, 4667–4673, https://doi.org/10.1016/j.atmosenv.2004.05.035, 2004.

Li, Y., Wu, Z., Ji, Y., Chen, T., Li, H., Gao, R., Xue, L., Wang, Y., Zhao, Y., and Yang, X.: Comparison of the ozone formation mechanisms and VOCs apportionment in different ozone pollution episodes in urban Beijing in 2019 and 2020: Insights for ozone pollution control strategies, Sci. Total. Environ., 908, https://doi.org/10.1016/j.scitotenv.2023.168332, 2024.

Liu, T., Hong, Y., Li, M., Xu, L., Chen, J., Bian, Y., Yang, C., Dan, Y., Zhang, Y., Xue, L., Zhao, M., Huang, Z., and Wang, H.: Atmospheric oxidation capacity and ozone pollution mechanism in a coastal city of southeastern China: analysis of a typical photochemical episode by an observation-based model, Atmos. Chem. Phys., 22, 2173-2190, https://doi.org/10.5194/acp-22-2173-2022, 2022.

Lyu, X. P., Chen, N., Guo, H., Zhang, W. H., Wang, N., Wang, Y., and Liu, M.: Ambient volatile organic compounds and their effect on ozone production in Wuhan, central China, Sci. Total. Environ., 541, 200-209, https://doi.org/10.1016/j.scitotenv.2015.09.093, 2016.

McGaughey, G. R., Desai, N. R., Allen, D. T., Seila, R. L., Lonneman, W. A., Fraser, M. P., Harley, R. A., Pollack, A. K., Ivy, J. M., and Price, J. H.: Analysis of motor vehicle emissions in a Houston tunnel during the Texas Air Quality Study 2000, Atmos. Environ., 38, 3363-3372, https://doi.org/10.1016/j.atmosenv.2004.03.006, 2004.

Meng, X., Jiang, J., Chen, T., Zhang, Z., Lu, B., Liu, C., Xue, L., Chen, J., Herrmann, H., and Li, X.: Chemical drivers of ozone change in extreme temperatures in eastern China, Sci. Total. Environ., 874, https://doi.org/10.1016/j.scitotenv.2023.162424, 2023.

Ming, W., Wentai, C., Lin, Z., Wei, Q., Yong, Z., Xiangzhi, Z., and Xin, X.: Ozone pollution characteristics and sensitivity analysis using an observation-based model in Nanjing, Yangtze River Delta Region of China, J. Environ. Sci., https://doi.org/10.1016/j.jes.2020.02.027, 2020.

Qin, Z., Xu, B., Zheng, Z., Li, L., Zhang, G., Li, S., Geng, C., Bai, Z., and Yang, W.: Integrating ambient

carbonyl compounds provides insight into the constrained ozone formation chemistry in Zibo city of the North China Plain, Environ. Pollut., 324, https://doi.org/10.1016/j.envpol.2023.121294, 2023.

Sillman, S.: The relation between ozone, $NO_x$ and hydrocarbons in urban and polluted rural environments, Atmos. Environ., 33, 1821–1845, https://doi.org/https://doi.org/10.1016/S1352-2310(98)00345-8, 1999.

Song, M., Liu, X., Zhang, Y., Shao, M., Lu, K., Tan, Q., Feng, M., and Qu, Y.: Sources and abatement mechanisms of VOCs in southern China, Atmos. Environ., 201, 28-40, https://doi.org/10.1016/j.atmosenv.2018.12.019, 2019.

Uria-Tellaetxe, I., and Carslaw, D. C.: Conditional bivariate probability function for source identification, Environ. Modell. Softw., 59, 1-9, https://doi.org/10.1016/j.envsoft.2014.05.002, 2014.

Wang, M., Qin, W., Chen, W., Zhang, L., Zhang, Y., Zhang, X., and Xie, X.: Seasonal variability of VOCs in Nanjing, Yangtze River delta: Implications for emission sources and photochemistry, Atmos. Environ., 223, https://doi.org/10.1016/j.atmosenv.2019.117254, 2020.

Wang, P., Chen, Y., Hu, J., Zhang, H., and Ying, Q.: Source apportionment of summertime ozone in China using a source-oriented chemical transport model, Atmos. Environ., 211, 79-90, https://doi.org/10.1016/j.atmosenv.2019.05.006, 2019.

Wang, X., Yin, S., Zhang, R., Yuan, M., and Ying, Q.: Assessment of summertime $O_3$ formation and the $O_3$-$NO_x$-VOC sensitivity in Zhengzhou, China using an observation-based model, Sci. Total. Environ., 813, 152449, https://doi.org/10.1016/j.scitotenv.2021.152449, 2022.

Wang, Y., Wang, H., Guo, H., Lyu, X., Cheng, H., Ling, Z., Louie, P. K. K., Simpson, I. J., Meinardi, S., and Blake, D. R.: Long-term $O_3$–precursor relationships in Hong Kong: field observation and model simulation, Atmos. Chem. Phys., 17, 10919-10935, https://doi.org/10.5194/acp-17-10919-2017, 2017.

Wang, Y., Guo, H., Zou, S., Lyu, X., Ling, Z., Cheng, H., and Zeren, Y.: Surface $O_3$ photochemistry over the South China Sea: Application of a near-explicit chemical mechanism box model, Environ. Pollut., 234, 155-166, https://doi.org/10.1016/j.envpol.2017.11.001, 2018.

Watson, J. G., Chow, J. C., and Fujita, E. M.: Review of volatile organic compound source apportionment by chemical mass balance, Atmos. Environ., 35, 1567-1584, https://doi.org/10.1016/S1352-2310(00)00461-1, 2001.

Xie, Y., Cheng, C., Wang, Z., Wang, K., Wang, Y., Zhang, X., Li, X., Ren, L., Liu, M., and Li, M.: Exploration of $O_3$-precursor relationship and observation-oriented $O_3$ control strategies in a non-provincial capital city, southwestern China, Sci. Total. Environ., 800, 149422, https://doi.org/10.1016/j.scitotenv.2021.149422, 2021.

Xu, Z., Zou, Q., Jin, L., Shen, Y., Shen, J., Xu, B., Qu, F., Zhang, F., Xu, J., Pei, X., Xie, G., Kuang, B., Huang, X., Tian, X., and Wang, Z.: Characteristics and sources of ambient Volatile Organic Compounds (VOCs) at a regional background site, YRD region, China: Significant influence of solvent evaporation during hot months, Sci. Total. Environ., 857, 159674, https://doi.org/10.1016/j.scitotenv.2022.159674, 2023.

Zhang, C., Liu, X., Zhang, Y., Tan, Q., Feng, M., Qu, Y., An, J., Deng, Y., Zhai, R., Wang, Z., Cheng, N., and Zha, S.: Characteristics, source apportionment and chemical conversions of VOCs based on a

comprehensive summer observation experiment in Beijing, Atmos. Pollut. Res., 12, 230-241, https://doi.org/10.1016/j.apr.2020.12.010, 2021.

Zhang, H., Wang, Y., Hu, J., Ying, Q., and Hu, X. M.: Relationships between meteorological parameters and criteria air pollutants in three megacities in China, Environ. Res., 140, 242-254, https://doi.org/10.1016/j.envres.2015.04.004, 2015.

Zhang, H., Yin, S., Xu, Y., Zhang, D., Yu, S., Lu, X., and Xin, K.: Multiple source apportionments, secondary transformation potential and human exposure of VOCs: A case study in a megacity of China, Atmos. Res., 291, https://doi.org/10.1016/j.atmosres.2023.106823, 2023.

Zhao, Y., Chen, L., Li, K., Han, L., Zhang, X., Wu, X., Gao, X., Azzi, M., and Cen, K.: Atmospheric ozone chemistry and control strategies in Hangzhou, China: Application of a 0-D box model, Atmos. Res., 246, https://doi.org/10.1016/j.atmosres.2020.105109, 2020.

Zong, R., Yang, X., Wen, L., Xu, C., Zhu, Y., Chen, T., Yao, L., Wang, L., Zhang, J., Yang, L., Wang, X., Shao, M., Zhu, T., Xue, L., and Wang, W.: Strong ozone production at a rural site in the North China Plain: Mixed effects of urban plumes and biogenic emissions, J. Environ. Sci., 71, 261-270, https://doi.org/10.1016/j.jes.2018.05.003, 2018.

---

## Author Comment (AC2)

**Referee comments:**

The study investigates the characteristics of VOCs and their importance in ozone formation during June in Zhengzhou City, China. The study focuses on the interesting relationship of $O_3$-$NO_x$-VOCs during summer, which is crucial for ozone control strategies. The study compares $O_3$ pollution events and clean days regarding different sources and $O_3$ formation sensitivity. However, the manuscript is poorly written and not up to the mark for consideration for publication in ACP. The authors fail to discuss crucial sections of the manuscript.

The authors do not include basic details about the instrumentation and dataset. Why is only a small part of the VOC measurement included in the PMF analysis? The reason is not mentioned properly in the manuscript or supplementary.

More in-depth comparisons with similar previous studies in Chinese/ Zhengzhou city would enhance the wider impact of the manuscript. Details about common factors and key trace VOC species can also be included. Any new or unique source or marker emerging from the region during the study may provide valuable insights.

Discussion related to the influence of meteorology and the transport of air masses needs to be included and explained.

Some of the statements require supporting references and proper reasoning. Also, VOCs can be changed to NMVOCs throughout the manuscript.

**Response:** Thank you for your careful reading of our paper and valuable comments and suggestions. We believe that we have adequately addressed your comments. To facilitate your review, the comments are in black, and the responses are in blue. The major changes that have been made according to these responses were marked in yellow color in the highlighted copy of the revised manuscript. And our own minor changes were marked in red font. Note that the following line numbers are shown in the corrected version.

We have revised the abbreviation in the manuscript from "VOCs" to "NMVOCs" as recommended. But, in Section 3.3, we further differentiate NMVOCs into AVOC and BVOC. The abbreviations for AVOCs and BVOCs remain unchanged.

**Detailed comments:**

**Lines 114-115:** "The sampling site is surrounded by residential areas, commercial areas, and office buildings, and there are no obvious atmospheric pollution sources nearby, which is a typical urban site.". These lines should be changed to the following for better clarity. "The sampling site is a typical urban site, surrounded by residential areas, commercial areas, and office buildings. There are no point sources of air pollution nearby (mention up to how much radius)."

**Response:** Thanks for the suggestion.

We appreciate your suggestion to improve the clarity of the description of the sampling site. We have revised the relevant lines as per your recommendation:

"The sampling site is a typical urban site, surrounded by residential areas, commercial areas, and office buildings. There are no point sources of air pollution nearby within a radius of 1 meter."

We believe that these changes enhance the clarity of the description and provide a more precise understanding of the sampling site.

We have revised the manuscript according to your request. (Line 112-114)

Line 112-114: The sampling site is a typical urban site, surrounded by residential areas, commercial areas, and office buildings. There are no point sources of air pollution nearby within a radius of 1 meter.

**Line 99** 'heaviest' should be 'highest'

**Response:** Sorry for this mistake.

We have replaced the "heaviest" to "highest".

**Lines 116-117:** "The sampling site is surrounded by roads and vegetation, and the sampling may be affected by motor vehicle emissions and

plant emissions." changed to "The sampling site may be affected by motor vehicle and plant emissions."

**Response:** Thanks for the suggestion.

We have revised the original content according to your request. (Line 115)

Line 115: The sampling site may be affected by motor vehicle and plant emissions.

**Section 2.2. Sample collection and chemical analysis**

More details should be provided about the instruments used for supporting measurements. Details about the input of sampling air should be added. Details about sampling dates, calibration, sampling time resolution, etc, should be added.

**Response:** Thanks for the suggestion.

As per your request, we have added some more details about the instrument. (Line 120-128).

Line 120-128: The time resolution of the instrument is 1 hour, and the flow rate is 60 mL/min. The air sample was collected for the first 5 minutes of each hour and then pre-concentrated through a cold trap to remove $H_2O_2$ and $CO_2$. The sample was captured using an empty capillary column. After pre-concentration, the sample was desorbed by rapid heating and introduced into an analytical

system. After separation by chromatographic column, the sample was detected by FID (for C2-C5 hydrocarbons) and MS (for C5-C12 hydrocarbons, halocarbons and OVOCs). The correlation coefficient of the standard curve of the target compound was greater than or equal to 0.99, and the detection limit of the instrument method was less than or equal to 0.1 nmol/mol. A total of 115 NMVOCs were monitored, including 29 alkanes, 11 alkenes, 1 alkyne, 17 aromatic hydrocarbons, 35 halogenated hydrocarbons, 21 OVOCs and 1 sulfide (carbon disulfide).

**Section 2.3 PMF model**

Only 29 out of 115 VOCs have been used for PMF analysis. Why is that? Multiple studies have used more than ~90 VOCs in the PMF model and shown its advantages. The author mentioned that species with missing samples were excluded. A very high proportion of sampling is missed out. This questions the reliability of the collected samples and dataset. More details about the error matrix and uncertainty should be included in this section. Why were the 5-factor solution, 6-factor solution, and 8- factor solution not considered? Did the author observe any source tracers or markers mixing in these solutions?

**Response:** Thanks for the useful comment and constructive

suggestions. The introduction was reorganized to make it clearer. Next, we will respond to the above questions one by one.

1. First, we chose to analyze 29 out of 115 NMVOCs for PMF analysis based on the specific objectives and data collection methods of our study. Some NMVOCs were excluded due to missing samples, as we aimed to ensure the accuracy and reliability of the data. Second, our research followed three principles in selecting species: (1) species with relatively high proportions of samples missing or with concentration values more than 25% below the MDLs were excluded; (2) typical NMVOCs tracers of emission sources were included; (3) NMVOCs with short atmospheric lifetimes were excluded.

2. We searched the literature and indeed found that some studies have used more than 90 types of NMVOCs in PMF software. Wang et al. and Jain et al. used PMF software to analyze over 90 NMVOCs in the Delhi. Wang et al. analyzed 101 NMVOCs in the Beijing. In Li et al. (2022) 's study, a total of 225 chemicals were used in the PMF model to quantitatively analyze the contribution of possible sources of NMVOCs measurements during CTT movement. However, these studies mention that PMF analysis uses more than 90 ions as an input matrix to identify different emission sources. These studies used ions, not NMVOCs. In addition, we also read some recent literature studies and found that some studies still use less than 50 NMVOCs (Yu et al.,

2022;Pernov et al., 2021;Mishra et al., 2023;Zhang et al., 2023;Zuo et al., 2024). We acknowledge the advantages of using a larger range of volatile organic compounds for PMF model analysis in the study. In future research, we will consider expanding the scope of NMVOCs to gain a more comprehensive understanding of the sources and impacts of air pollutants.

3.  In fact, we tried solutions with different factors. As shown in fig. S2, we explored the number of PMF factors from 3 to 12 to obtain the best solution. Each model is run 20 times. The $Q_{robust}$, $Q_{true}$ , $Q_{theoretical}$ , $Q_{true}/Q_{robust}$ , and Q true / Q theoretical in different solutions are discussed subsequently. Fpeak values from $-2$ to 2 are used in the model. Finally, we adopted a 7-factor solution ($Q_{true}/Q_{theoretical} = 3.42$; and Fpeak $= 0$). In addition, we also add some explanations about the rational selection of factors in the manuscript.

4.  The emission sources of NMVOCs in the atmosphere are complex, and different sources may emit the same substances. Therefore, when using the Positive Matrix Factorization (PMF) software for source apportionment analysis, different sources may have common substances. However, each source has unique tracer substances. Currently, most studies identify different sources based on characteristic substances. In this study, typical tracer substances used for solvent sources include chloromethane, dichloromethane,

tetrachloromethane, 1,2-dichloroethane, 1,2-dichloropropane, ethyl acetate, methylcyclopentane, cyclohexane, TEXs (Toluene, Ethylbenzene, m/p-Xylene, and o-Xylene), 1,2-Dichloroethane, 1,2-Dichloropropane, and Ethyl acetate. Methylcyclopentane and cyclohexane. These substances are commonly used in solvent applications. In solvent sources, we also observed some other substances that cannot be used as solvents. We have reviewed many literatures and found similar issues. However, the proportion of these substances in this factor is very small, so they can be ignored.

**Section 2.4 Conditional bivariate probability function analysis**

More details in the section are required.

**Response:** Thank you for your valuable feedback.

Regarding Section 2.4, we have acknowledged the error in the title "Conditional bivariate probability function analysis" and have made the necessary correction to "Conditional probability function analysis" in the manuscript. Additional details have been included to provide a more comprehensive explanation of the content in this section. We have aimed to enhance the clarity and understanding of our study for the readers. Thank you for guiding us in improving our manuscript.

**Line 192** areas are non-polluting processes (clean days). Remove 'non-polluting processes'

**Response:** Thanks for the suggestion. We have revised the text to remove the phrase "non-polluting processes".

**Line 192-193** During the observation, $O_3$ polluted days were 22 days, accounting for 73%. You mentioned cases 1 (8th-17th Jun.) and 2 (20th-27th Jun.) as pollution events, which is 18 days instead of 22 days. There is a discrepancy. Most of the days are included in ozone pollution events in June.

**Response:** We apologize for the confusion caused by our inappropriate description.

1. As mentioned in line 197, during the entire observation process, apart from Cases 1 (8[th]-17[th] Jun.) and Case 2 (20[th]-27[th] Jun.), $O_3$ pollution also occurred on 6th Jun. and 29[th]-30[th] Jun. $O_3$ pollution lasted a total of 18 days during Cases 1 and Case 2, but throughout the observation process, $O_3$ pollution occurred for a total of 21 days.

2. We apologize for the discrepancy in the statistical methods, where the manuscript originally stated "22 days." We have now corrected it to "21 days."

3. We have modified the "During the observation, $O_3$ polluted days were 22 days, accounting for 73%." to "During the observation, $O_3$

polluted days were 21 days, accounting for 70%". (Line 215-216)

Line 215-216: During the observation, $O_3$ polluted days were 21 days, accounting for 70%.

**Line 198** 'The mean concentrations' change to 'The daily mean concentrations'.

**Response:** Thanks for the suggestion.

We actually use the hourly average concentration. We have replaced "The mean concentrations" to "hourly average concentration". (Line 221-222).

Line 221-222: Hourly average concentration of $SO_2$, $NO_2$, CO, and $PM_{2.5}$ were $4.4 \pm 3.3$ $\mu g/m^3$, $26.5 \pm 17.9$ $\mu g/m^3$, $0.6 \pm 0.2$ $mg/m^3$, $59.6 \pm 26.5$ $\mu g/m^3$ and $22.9 \pm 7.1$ $\mu g/m^3$, respectively.

**199-200:** All were lower than the ambient air quality standard value (National or WHO). Add reference. Also, compare the values with national standards/ WHO standards for pollutant criteria.

The mean of each VOCs species (115 in number) or at least which species have been measured should be added to the supplementary in Table.

The variation of different families or groups of VOCs can be added as a time series in Figure 1. Also, the left and right y-axis are not aligned properly with the text.

**Response:** Thank you for providing valuable feedback. We apologize for this unclear illustration. We have carefully considered your suggestions and made the necessary revisions. Below are our responses to each of the points you raised:

1. We added comparisons of pollutant concentrations with the grade I threshold of the China National Ambient Air Quality Standard (NAAQS-2012) to our manuscript. We have modified the "All of them were lower than the ambient air quality standard value." to "The concentrations of these pollutants were 97%, 87%, 94%, and 35% lower than the grade I threshold of the NAAQS-2012". (Line 222-223).

2. We have added the mean concentration of each of the 115 NMVOCs species that have been measured to the supplementary Table S1 in Supplementary materials.

3. The variation of different groups of NMVOCs has been included as a time series in Figure 1.

4. The alignment of the left and right y-axes with the text has been adjusted to ensure clarity and accuracy of the figures.

   Line 222-223: The concentrations of these pollutants were 97%, 87%, 94%, and 35% lower than the grade I threshold of the NAAQS-2012.

**Section 3.2 Sources of VOCs**

**Figure 4:** The VOCs concentration is given in ug/m$^3$, while it is mentioned as ppbv in the whole manuscript. Please check the discrepancy in the units.

An explanation of each factor concerning the sampling site is required. As you mentioned, there are no point sources nearby, so why is industrial pollution showing 22%? What could be the contributing factors for it?

Have you also performed PMF on clean days and polluted days simultaneously? One suggestion is to check if PMF gives different results in different cases (Case 1 and Case 2 and clean days) and compare the results.

Secondary VOCs, such as OVOCs and amines, play an important role in identifying VOCs sources. The ratio of OVOCs and other family groups should also be analysed in each source to determine their contribution. This will give great insights into the source characterisation.

**Response:** Thank you for valuable suggestions. Below are our responses to each of the points you raised:

1. We apologize for the mistake in Figure 4 where the NMVOCs concentration is incorrectly labeled as μg/m$^3$ instead of ppbv. We will correct this error in the revised manuscript. The correct unit for

NMVOCs concentration in Figure 4 should indeed be ppbv, consistent with the rest of the manuscript.

2. There are indeed no large industrial emission sources near the monitoring site, but the industrial emissions in Zhengzhou city account for a very high proportion. Lu et al. (2024) analyzed the NMVOCs emission inventory of Zhengzhou city in their latest study, and combined with the PMF model to analyze the sources of NMVOCs. Both the NMVOCs emission inventory and PMF simulation results indicate that Zhengzhou is heavily influenced by industrial sources. Liu et al. (2024) established a NMVOCs emission inventory for the Central China region represented by Henan Province. The inventory shows that Zhengzhou is the city with the largest NMVOCs emissions, and industrial emissions are the main contributing source of NMVOCs. Therefore, although there are no large industrial emission sources near the monitoring site, due to the large industrial emissions in the Zhengzhou area, the monitoring site will be affected by the transport of industrial emission NMVOCs.

   Figure 1 shows the distribution of industrial sites around the observation points. It can be seen that there are a large number of industrial sites in Zhengzhou city.

[Figure]

Fig. 1 Distribution of industrial sites around observation sites.

3.  We have performed the PMF analysis on both clean days and polluted days simultaneously. Fig. 2. shows the source profiles and contributions of NMVOCs in Case 1. From Figure 3, we can see that the contributions of combustion, industrial production, biogenic emission, vehicular exhaust, LPG/NG, and solvent usage are 8.9, 23.9, 4.4, 29.7, 7.3, and 25.7%, respectively. Compared with the Case 1 event in the manuscript, the differences in contributions of each factor are not significant. Vehicular exhaust, solvent usage, and biogenic emission decrease by 0.3, 1.3, and 1.2% respectively, while combustion, industrial production, and LPG/NG increase by 0.9, 0.9, and 0.3% respectively. Fig. 4. shows the source profiles and contributions of NMVOCs in Case 2. From Figure 5, we can see that the contributions of combustion, industrial production, biogenic emission, vehicular exhaust, LPG/NG, and solvent usage are 10.7, 20.0, 7.2, 16.6, 19.5, and 26.1%, respectively. Compared with the Case 2

event in the manuscript, industrial, biogenic emission, LPG/NG, and solvent usage decrease by 1.0, 0.8, 1.5, and 1.9% respectively, while combustion, biogenic emission, and vehicular exhaust increase by 3.7% and 2.6%. Fig. 6. shows the source profiles and contributions of NMVOCs in clean days. From Figure 7, we can see that the contributions of combustion, industrial production, biogenic emission, vehicular exhaust, LPG/NG, and solvent usage are 11.4, 19.6, 3.6, 33.6, 5.4, and 26.4%, respectively. Compared with the clean days in the manuscript, industrial and vehicular exhaust decrease by 1.5% and 1.4%, while combustion, biogenic emission, LPG/NG, and solvent usage increase by 0.4, 0.6, 0.4, and 1.4%, respectively. From the above analysis, it can be seen that although performing PMF separately on clean days and polluted days may lead to some differences, the relative contributions of each factor in Case 1, Case 2, and clean days have not changed. Compared to the manuscript, the main conclusions remain consistent, namely that vehicular exhaust, solvent usage, and industrial production were major contributors to both $O_3$ pollution events and clean days.

4. We appreciate the emphasis on the importance of Secondary NMVOCs, such as OVOCs and amines, in identifying NMVOCs sources. According to your request, we reperformed PMF to attempt to analyze the contribution of secondary formation to NMVOCs. However, we

found that the secondary formation source obtained always contain trace substances from other sources, and these substances contribute significantly. Additionally, we reviewed the literature and found that these trace substances from other sources are also present in the secondary formation source in the literature (Zhang et al., 2023;Wen et al., 2024;Zeng et al., 2023), but the authors did not explain this phenomenon. We believe that there may be a significant error in the secondary formation source obtained through PMF analysis in this study. Therefore, we did not further analyze the contribution of secondary formation source to NMVOCs in this study. Once again, we appreciate your suggestions and guidance. In future studies, we will carefully and thoroughly analyze the secondary formation source of NMVOCs.

[Figure]

**Fig. 2.** Source profiles and contributions of NMVOCs in Case 1.

[Figure]

**Fig. 3.** Source contributions to NMVOCs concentration in Case 1.

[Figure]

**Fig. 4.** Source profiles and contributions of NMVOCs s in Case 2.

[Figure]

**Fig. 5.** Source contributions to NMVOCs concentration in Case 2.

[Figure]

**Fig. 6.** Source profiles and contributions of NMVOCs s in Case 2.

[Figure]

**Fig. 7.** Source contributions to NMVOCs concentration in Case 2.

**More explanation is required for Figure 5.**

**Liness 366-369** indicate that similar sources contribute to $O_3$ pollution events and clean days. Does it mean that the emissions of primary

and secondary VOCs do not influence $O_3$ formation? Then, what are the reasons for $O_3$ formation in the area?

**Response:** Thank you for your comments.

1. In this study, the contributions of various pollution sources show relatively minor differences between $O_3$ pollution events and clean days, but there are still some distinctions. For instance, compared to clean days, in Case 1 events, industrial production, biogenic emission, LPG/Ng, and solvent usage increased by 2%, 3%, 2%, and 2% respectively. Compared to clean days, in Case 2 events, solvent usage, biogenic emission, and LPG/Ng increased by 3%, 5%, and 16% respectively. Therefore, the increased contributions of solvent usage, biogenic emission, and LPG/Ng may have a certain impact on the formation of $O_3$ pollution.

2. Although compared with the pollution process, the contribution changes of each pollution source in $O_3$ pollution events are not very obvious. It does not mean that the emissions of primary and secondary NMVOCs do not influence $O_3$ formation. In fact, emissions of both primary and secondary NMVOCs are important factors in $O_3$ formation. The reasons for $O_3$ formation in the area may involve various complex factors, including but not limited to emissions of nitrogen oxides, levels of solar radiation, meteorological conditions, etc.

3. We compared the average concentrations of nitrogen oxides in Case 1,

Case 2, and clean days. The average concentrations of $NO_2$ in Case 1, Case 2, and clean days were $27.4 \pm 19.5$, $24.9 \pm 12.3$, and $24.4 \pm 16.1$ ppbv, respectively, while the average concentrations of NO were $3.9 \pm 3.6$, $3.9 \pm 2.4$, and $4.8 \pm 5.5$ ppbv, respectively. The average concentrations of $NO_2$ in pollution events were higher than those in clean days, while the average concentrations of NO were lower than those in clean days. Higher concentration of $NO_2$ can promote the formation of $O_3$, while the titration reaction between NO and $O_3$ consumes $O_3$. Therefore, the higher concentration of $NO_2$ and lower concentration of NO during pollution events are one of the reasons for the occurrence of $O_3$ pollution events.

4. We further explored the relationship between meteorology and $O_3$ concentration. According to Fig. S3a and Fig. S3b, it can be observed that $O_3$ concentration shows a linear increasing trend with temperature and a linear decreasing trend with RH. $O_3$ has a significant correlation with temperature and RH, with correlation coefficients of 0.7 and -0.61 respectively. Therefore, conditions of high temperature and low RH are more conducive to $O_3$ pollution. Fig. S3c indicates that $O_3$ concentration exceeding the secondary standard mainly occurs under meteorological conditions of high temperature (greater than 30 °C) and low RH (less than 55%). It can be noted that when 35 °C < T < 40 °C and 20% < RH < 40%, the $O_3$ concentration consistently exceeds the

secondary standard. Wang et al. argued that most of the reactions involved in ozone formation increase with temperature, and the rate of ozone production exceeds that of ozone loss by a large margin (Meng et al., 2023).

In conclusion, in addition to the impact of solvent usage, biogenic emission, and LPG/Ng on $O_3$ pollution events, meteorological factors are also significant factors in the occurrence of $O_3$ pollution events. In addition, we have added an analysis of the correlation between $O_3$ and temperature and RH in the manuscript. (Line 235-253)

[Figure]

Fig. S3 Correlation analysis of $O_3$, T, and RH.

Line 235-253: The average concentrations of TNMVOCs, $NO_2$, $PM_{10}$, and $PM_{2.5}$ on clean days were lower than those of the $O_3$ pollution events. The average RH (65 ± 17%) on clean days was higher than those during Case 1 and Case 2 events, while the average temperature (26.0 ± 4.8 °C) was lower than those during Case 1 and Case 2 events. According to the analysis in Fig. S3a and Fig. S3b, $O_3$ has a significant correlation with temperature and RH,

with correlation coefficients of 0.7 and -0.61 respectively. Therefore, conditions of high temperature and low RH are more conducive to $O_3$ pollution. Fig. S3c indicates that $O_3$ concentration exceeding the secondary standard mainly occurs under meteorological conditions of high temperature (greater than 30 °C) and low RH (less than 55%). It can be noted that when 35 °C < T < 40 °C and 20% < RH < 40%, the $O_3$ concentration consistently exceeds the grade II threshold of the NAAQS-2012. High temperature and low RH are more conducive to $O_3$ pollution(Chen et al., 2020;Zhang et al., 2015). Meng et al. (2023) argued that most of the reactions involved in $O_3$ formation increase with temperature, and the rate of $O_3$ production exceeds that of $O_3$ loss by a large margin. Therefore, during the study period, the meteorological conditions of high temperature and low RH are also important factors affecting the occurrence of $O_3$ pollution.

Besides, the average concentration of $NO_2$ in clean days (24.4 ± 16.1 ppbv) was lower than that in Case 1 and Case 2, while the average concentration of NO in clean days (4.8 ± 5.5 ppbv) was higher than that in Case 1 (3.9 ± 3.75 ppbv) and Case 2 (3.9 ± 2.4 ppbv). Higher concentration of $NO_2$ can promote the formation of $O_3$, while the titration reaction between NO and $O_3$ consumes $O_3$ (Sillman, 1999). Therefore, the higher concentration of $NO_2$ and

lower concentration of NO during pollution events are one of the reasons for the occurrence of $O_3$ pollution events.

**Section 3.3.1 $O_3$ sensitivity analysis**

**Line 383** indicates that $O_3$ formation is more sensitive to biogenic emissions. Why is that? Add references to previous studies in urban settings, how AVOCs and BVOCs vary and their effect on ozone levels.

**Response:** Thank you for this valuable comment.

In summer, due to higher solar radiation, biogenic emissions are an important source of NMVOCs. Studies in Yucheng (Zong et al., 2018), Leshan (Xie et al., 2021), and and Nanjing (Fan et al., 2021;Ming et al., 2020) have shown that ozone is highly sensitive to BVOCs. Studies in Zhengzhou (Wang et al., 2022), Hangzhou (Zhao et al., 2020), and Hong Kong (Wang et al., 2017) suggested that ozone exhibits greater sensitivity to BVOCs than AVOCS during hot seasons. Wang et al. (2019) found in their study on $O_3$ source apportionment in Henan Province, where Zhengzhou is located, that BVOCs contribute to approximately 23.9% of the $O_3$ attributed to NMVOCs. Previous studies on $O_3$ sensitivity analysis in Zhengzhou have shown a strong sensitivity of $O_3$ to BVOCs. Wang et al. pointed out that in two $O_3$ pollution events that occurred, $O_3$ exhibited higher sensitivity to BVOCs than AVOCs.

Furthermore, research in other regions has also indicated a higher sensitivity of summer $O_3$ to BVOCs. The time of this study is in the summer months with the highest temperature, which is more conducive to plant emissions.

In response to your query regarding the reasons for this sensitivity, we have incorporated references to previous studies conducted in urban settings that have discussed the variations of AVOCs and BVOCs and their impact on ozone levels. By including this additional information and discussing the relevant literature, we aim to have provided a more comprehensive analysis of the factors influencing $O_3$ formation. More analysis has been provided in the revised manuscript to clarify this aspect more effectively. (Line 449-459)

Line 449-459: Isoprene was the sole BVOC considered in this study. Isoprene is an important tracer to indicate biogenic emissions (Xie et al., 2021;Li et al., 2024;Qin et al., 2023). During the entire period, especially in the pollution events, the RIR of AVOCs was lower than that of BVOCs, indicating that $O_3$ formation was more sensitive to biogenic emissions. This may be due to increased emissions of BVOCs at higher temperatures and solar radiation conditions, as well as their high reactivity and $O_3$ formation potential. Studies in Yucheng (Zong et al., 2018), Leshan (Xie et al.,

2021), and and Nanjing (Fan et al., 2021;Ming et al., 2020) have shown that $O_3$ is highly sensitive to BVOCs. Studies in Zhengzhou (Wang et al., 2022), Hangzhou (Zhao et al., 2020), and Hong Kong (Wang et al., 2017) suggested that $O_3$ exhibits greater sensitivity to BVOCs than AVOCS during hot seasons. Wang et al. (2019) found in their study on $O_3$ source apportionment in Henan Province, where Zhengzhou is located, that BVOCs contribute to approximately 23.9% of the $O_3$ attributed to NMVOCs. Therefore, the contribution of biogenic NMVOCs to $O_3$ is very important.

**Line 382-383** What could be the possible reasons for the RIR of BVOCs being higher than AVOCs? What are the biogenic VOCs species you have included in this analysis? Add these details in the section.

**Response:** Thank you for this valuable suggestion.

The average RIR value for BVOCs was higher than that for AVOCs, primarily due to the elevated BVOCs emissions under conditions of higher temperature and solar radiation, along with their high reactivities and ozone formation potential. Our study focused on the period when summer temperatures and solar radiation are at their highest, resulting in peak biogenic emissions of BVOCs. Furthermore, the monitoring sites were surrounded by abundant vegetation cover. Isoprene was the sole BVOCs considered in this analysis. Isoprene is an important tracer to indicate biogenic

emissions. Currently, many studies use Isoprene to represent BVOCs (Xie et al., 2021;Li et al., 2024;Qin et al., 2023). We have incorporated these details in manuscript. (Line 449-459)

**Figure 6:** The RIR (%) looks similar for every case. Even for clean days, aromatics show higher values than polluted events. I suggest to check the values.

Another section should be added to compare the study results with the source apportionment studies in the city in different periods or seasons. Also, comparison with other Chinese cities studies can add extra value to the analysis.

**Response:** Thank you for the valuable feedback and suggestions. We have carefully checked the original data and confirmed that there are no issues with the results. The RIR values for different species/groups are shown in Table 1. The RIR value for aromatics on clean days is indeed greater than the RIR value for $O_3$ pollution events, indicating that the generation of $O_3$ is more sensitive to alkenes on polluted days, while on clean days, the generation of $O_3$ is more sensitive to aromatics.

Table 1. Average RIR values of the $O_3$ for different species/groups during different processes in Zhengzhou.

|  | AVOCs | BVOCs | CO | $NO_x$ | Alkanes | Alkenes | Aromatics |
|---|---|---|---|---|---|---|---|
| Entire period | 3.44 | 4.48 | 0.74 | -7.9 | 0.67 | 1.13 | 1.17 |
| $O_3$ pollution events | 3.44 | 5.3 | 0.66 | -8.4 | 0.69 | 1.2 | 1.09 |

| | | | | | | | |
|---|---|---|---|---|---|---|---|
| Clean days | 3.88 | 4.57 | 0.93 | -6.9 | 0.65 | 1.17 | 1.47 |

We are a little confused about the suggestion to add a comparison of source apportionment results, because Figure 6 does not cover source apportionment. Source apportionment is in Section 3.2.2, so we have followed your suggestion to add some content in Section 3.2.2 to compare the results of this study with source apportionment studies of different periods or seasons in the city. In addition, we have also considered comparing the study results with studies in other Chinese cities to add value to the analysis. (Line 414-434)

Line 414-434: In summary, the observation sites are significantly influenced by vehicular exhaust, solvent usage, and industrial production. The results of this study show similarities in the source apportionment of NMVOCs in Zhengzhou during the summers of 2018 to 2021 (Yu et al., 2022;Guo et al., 2024). Yu et al. (2022) found that vehicular exhaust and industrial production contributed the most to NMVOCs emissions in Zhengzhou from 2018 to 2020, with the main sources of summer NMVOCs being vehicular exhaust, solvent usage, and industrial production. In contrast to the NMVOCs source apportionment results of Li et al. (2021). for the $O_3$ pollution process in Zhengzhou in May 2018, the difference lies in the higher impact of solvent usage compared to vehicular exhaust

and industrial production. This is mainly attributed to the fact that Li et al. (2021)'s observation site was located within Zhengzhou University, making them more susceptible to the influence of chemical reagent use. In comparison to the source apportionment of NMVOCs in Zhengzhou during winter (Zhang et al., 2021), combustion also becomes an important contributor during winter, attributed to the increased heating demand, while the contribution from solvent usage is relatively lower due to the cold temperatures. In comparison with other cities (Table S2), vehicular exhaust in Zhengzhou contributes the most, higher than in cities such as Qingdao (Wu et al., 2023), Xuchang (Qin et al., 2021)), Guangzhou (Meng et al., 2022), Nanjing (Fan et al., 2021), Shijiazhuang (Guan et al., 2020), and Weinan (Hui et al., 2020), but lower than in Changzhou (Liu et al., 2023) and on par with Beijing (Liu et al., 2020). Solvent usage in Zhengzhou contributes more than in Qingdao (Wu et al., 2023), Xuchang (Qin et al., 2021), Nanjing (Fan et al., 2021), Shijiazhuang (Guan et al., 2020), Weinan (Hui et al., 2020), Changzhou (Liu et al., 2023), and Beijing (Liu et al., 2020), but less than in Guangzhou (Meng et al., 2022). Industrial production in Zhengzhou contributes more than in Xuchang (Qin et al., 2021), Guangzhou (Meng et al., 2022), Nanjing (Fan et al.,

2021), Weinan (Hui et al., 2020), and Changzhou (Liu et al., 2023), but less than in Shijiazhuang (Guan et al., 2020).

**Section 4 Conclusions**:

The authors have just given a summary of the results obtained. No explanation is included about why and how any trend follows the study of how sources influence $O_3$ formation in the area. What could be the driving factors for the presence of any particular source? You should include more details in the section.

**Response:** Thank you for the valuable suggestion regarding our study.

We have revised the conclusion section in the revised manuscript to integrate all results, rather than just summarizing the key findings of each section in turn. We strive to better demonstrate the overall significance and contribution of the research. Thank you once again for your feedback, and we have taken it into consideration and made the necessary changes. (Line 532-551)

Line 532-551: The summer $O_3$ pollution has always been an important environmental issue in Zhengzhou. This study investigated the characteristics and emission sources of $O_3$ precursors from 1st to 30th June 2023. The OBM was used to analyze the influence of precursors on the formation of $O_3$, and the

emission reduction strategy of precursors was proposed to control the concentration of $O_3$. During the entire period, the concentration of TNMVOCs varied from 9.9 to 60.3 ppbv, with an average value of 22.9 ± 8.3 ppbv. The average concentration of TNMVOCs during $O_3$ pollution was higher than that during clean days. Alkanes (44%), OVOCs (20%), and halocarbons (19%) were the most abundant NMVOCs group. Ethane, acetone, and propane were always the most abundant species. The average concentrations of $NO_2$ in pollution events were higher than those in clean days, while the average concentrations of NO were lower than those in clean days. Therefore, the increasing concentration of $O_3$ precursors is one of the reasons for the formation of $O_3$ pollution. At the same time, the unfavorable meteorological conditions of high temperature and low RH in the observation process are also important factors in the formation of $O_3$ pollution. Further analysis of the source of these precursors found that Vehicular exhaust (28%), solvent usage (27%), and industrial production (22%) were the main emission sources of VOCs. The increase of solvent usage, biogenic emission and LPN/NG contribution is an important cause of $O_3$ pollution. Sensitivity analysis of $O_3$ to precursors found that NMVOCs had the highest RIR value, while $NO_x$ had a negative RIR value. Alkenes have the highest RIR value among AVOCs. It

should be noted that the RIR value of BVOCs was greater than that of AVOCs. The local $O_3$ formations were in the AVOCs-limited regimes, which means reducing the concentration of AVOCs was an effective way to reduce $O_3$ concentration. Meanwhile, we suggest that the minimum reduction ratio of AVOCs/$NO_x$ should be no less than 3:1 to reduce $O_3$ production.

**References**

Chen, L., Zhu, J., Liao, H., Yang, Y., and Yue, X.: Meteorological influences on $PM_{2.5}$ and $O_3$ trends and associated health burden since China's clean air actions, Sci. Total. Environ., 744, 140837, https://doi.org/10.1016/j.scitotenv.2020.140837, 2020.

Fan, M., Zhang, Y., Lin, Y., Li, L., Xie, F., Hu, J., Mozaffar, A., and Cao, F.: Source apportionments of atmospheric volatile organic compounds in Nanjing, China during high ozone pollution season, Chemosphere, 263, https://doi.org/10.1016/j.chemosphere.2020.128025, 2021.

Guan, Y., Wang, L., Wang, S., Zhang, Y., Xiao, J., Wang, X., Duan, E., and Hou, L. a.: Temporal variations and source apportionment of volatile organic compounds at an urban site in Shijiazhuang, China, J. Environ. Sci., 97, 25-34, https://doi.org/10.1016/j.jes.2020.04.022, 2020.

Guo, J., Xu, Q., Yu, S., Zhao, B., and Zhang, M.: Investigation of atmospheric VOCs sources and ozone formation sensitivity during epidemic closure and control: A case study of Zhengzhou, Atmos. Pollut. Res., 15, https://doi.org/10.1016/j.apr.2023.102035, 2024.

Hui, L., Ma, T., Gao, Z., Gao, J., Wang, Z., Xue, L., Liu, H., and Liu, J.: Characteristics and sources of volatile organic compounds during high ozone episodes: A case study at a site in the eastern Guanzhong Plain, China, Chemosphere, 265, 129072, https://doi.org/10.1016/j.chemosphere.2020.129072, 2020.

Li, X., Yuan, B., Wang, S., Wang, C., Lan, J., Liu, Z., Song, Y., He, X., Huangfu, Y., Pei, C., Cheng, P., Yang, S., Qi, J., Wu, C., Huang, S., You, Y., Chang, M., Zheng, H., Yang, W., Wang, X., and Shao, M.: Variations and sources of volatile organic compounds (VOCs) in urban region: insights from measurements on a tall tower, Atmos. Chem. Phys., 22, 10567-10587, https://doi.org/10.5194/acp-22-10567-2022, 2022.

Li, Y., Yin, S., Yu, S., Bai, L., Wang, X., Lu, X., and Ma, S.: Characteristics of ozone pollution and the sensitivity to precursors during early summer in central plain, China, J Environ Sci (China), 99, 354-368, https://doi.org/10.1016/j.jes.2020.06.021, 2021.

Li, Y., Wu, Z., Ji, Y., Chen, T., Li, H., Gao, R., Xue, L., Wang, Y., Zhao, Y., and Yang, X.: Comparison of the ozone formation mechanisms and VOCs apportionment in different ozone pollution episodes in urban Beijing in 2019 and 2020: Insights for ozone pollution control strategies, Sci. Total. Environ., 908, https://doi.org/10.1016/j.scitotenv.2023.168332, 2024.

Liu, Y., Song, M., Liu, X., Zhang, Y., Hui, L., Kong, L., Zhang, Y., Zhang, C., Qu, Y., An, J., Ma, D., Tan, Q., and Feng, M.: Characterization and sources of volatile organic compounds (VOCs) and their related changes during ozone pollution days in 2016 in Beijing, China, Environ. Pollut., 257, 113599, https://doi.org/10.1016/j.envpol.2019.113599, 2020.

Liu, Y., Lu, X., Zhang, X., Wang, T., Li, Z., Wang, W., Kong, M., Chen, K., and Yin, S.: The newest emission inventory of anthropogenic full-volatility organic in Central China, Atmos. Res., 300, https://doi.org/10.1016/j.atmosres.2024.107245, 2024.

Liu, Z., Hu, K., Zhang, K., Zhu, S., Wang, M., and Li, L.: VOCs sources and roles in $O_3$ formation in the central Yangtze River Delta region of China, Atmos. Environ., 302, 119755, https://doi.org/10.1016/j.atmosenv.2023.119755, 2023.

Lu, X., Zhang, D., Wang, L., Wang, S., Zhang, X., Liu, Y., Chen, K., Song, X., Yin, S., Zhang, R., Wang, S., and Yuan, M.: Establishment and verification of anthropogenic speciated VOCs emission inventory of Central China, J. Environ. Sci., https://doi.org/10.1016/j.jes.2024.01.033, 2024.

Meng, X., Jiang, J., Chen, T., Zhang, Z., Lu, B., Liu, C., Xue, L., Chen, J., Herrmann, H., and Li, X.: Chemical drivers of ozone change in extreme temperatures in eastern China, Sci. Total. Environ., 874, https://doi.org/10.1016/j.scitotenv.2023.162424, 2023.

Meng, Y., Song, J., Zeng, L., Zhang, Y., Zhao, Y., Liu, X., Guo, H., Zhong, L., Ou, Y., Zhou, Y., Zhang, T., Yue, D., and Lai, S.: Ambient volatile organic compounds at a receptor site in the Pearl River Delta region: Variations, source apportionment and effects on ozone formation, J. Environ. Sci., 111, 104-117, https://doi.org/10.1016/j.jes.2021.02.024, 2022.

Ming, W., Wentai, C., Lin, Z., Wei, Q., Yong, Z., Xiangzhi, Z., and Xin, X.: Ozone pollution characteristics and sensitivity analysis using an observation-based model in Nanjing, Yangtze River Delta Region of China, J. Environ. Sci., https://doi.org/10.1016/j.jes.2020.02.027, 2020.

Mishra, M., Chen, P.-H., Bisquera, W., Lin, G.-Y., Le, T.-C., Dejchanchaiwong, R., Tekasakul, P., Jhang, C.-W., Wu, C.-J., and Tsai, C.-J.: Source-apportionment and spatial distribution analysis of VOCs and their role in ozone formation using machine learning in central-west Taiwan, Environ. Res., 232, https://doi.org/10.1016/j.envres.2023.116329, 2023.

Pernov, J. B., Bossi, R., Lebourgeois, T., Nøjgaard, J. K., Holzinger, R., Hjorth, J. L., and Skov, H.: Atmospheric VOC measurements at a High Arctic site: characteristics and source apportionment, Atmos. Chem. Phys., 21, 2895-2916, https://doi.org/10.5194/acp-21-2895-2021, 2021.

Qin, J., Wang, X., Yang, Y., Qin, Y., Shi, S., Xu, P., Chen, R., Zhou, X., Tan, J., and Wang, X.: Source apportionment of VOCs in a typical medium-sized city in North China Plain and implications on control policy, J. Environ. Sci., 107, 26-37, https://doi.org/10.1016/j.jes.2020.10.005, 2021.

Qin, Z., Xu, B., Zheng, Z., Li, L., Zhang, G., Li, S., Geng, C., Bai, Z., and Yang, W.: Integrating ambient carbonyl compounds provides insight into the constrained ozone formation chemistry in Zibo city of the North China Plain, Environ. Pollut., 324, https://doi.org/10.1016/j.envpol.2023.121294, 2023.

Sillman, S.: The relation between ozone, $NO_x$ and hydrocarbons in urban and polluted rural environments, Atmos. Environ., 33, 1821–1845, https://doi.org/https://doi.org/10.1016/S1352-2310(98)00345-8, 1999.

Wang, P., Chen, Y., Hu, J., Zhang, H., and Ying, Q.: Source apportionment of summertime ozone in China using a source-oriented chemical transport model, Atmos. Environ., 211, 79-90, https://doi.org/10.1016/j.atmosenv.2019.05.006, 2019.

Wang, X., Yin, S., Zhang, R., Yuan, M., and Ying, Q.: Assessment of summertime $O_3$ formation and the $O_3$-$NO_x$-VOC sensitivity in Zhengzhou, China using an observation-based model, Sci. Total. Environ., 813, 152449, https://doi.org/10.1016/j.scitotenv.2021.152449, 2022.

Wang, Y., Wang, H., Guo, H., Lyu, X., Cheng, H., Ling, Z., Louie, P. K. K., Simpson, I. J., Meinardi, S., and Blake, D. R.: Long-term $O_3$–precursor relationships in Hong Kong: field observation and model simulation, Atmos. Chem. Phys., 17, 10919-10935, https://doi.org/10.5194/acp-17-10919-2017, 2017.

Wen, M., Deng, W., Huang, J., Zhang, S., Lin, Q., Wang, C., Ma, S., Wang, W., Zhang, X., Li, G., and

An, T.: Atmospheric VOCs in an industrial coking facility and the surrounding area: Characteristics, spatial distribution and source apportionment, J. Environ. Sci., 138, 660-670, https://doi.org/10.1016/j.jes.2023.04.026, 2024.

Wu, Y., Liu, B., Meng, H., Dai, Q., Shi, L., Song, S., Feng, Y., and Hopke, P. K.: Changes in source apportioned VOCs during high $O_3$ periods using initial VOC-concentration-dispersion normalized PMF, Sci. Total. Environ., 896, https://doi.org/10.1016/j.scitotenv.2023.165182, 2023.

Xie, Y., Cheng, C., Wang, Z., Wang, K., Wang, Y., Zhang, X., Li, X., Ren, L., Liu, M., and Li, M.: Exploration of $O_3$-precursor relationship and observation-oriented $O_3$ control strategies in a non-provincial capital city, southwestern China, Sci. Total. Environ., 800, 149422, https://doi.org/10.1016/j.scitotenv.2021.149422, 2021.

Yu, S., Wang, S., Xu, R., Zhang, D., Zhang, M., Su, F., Lu, X., Li, X., Zhang, R., and Wang, L.: Measurement report: Intra- and interannual variability and source apportionment of volatile organic compounds during 2018–2020 in Zhengzhou, central China, Atmos. Chem. Phys., 22, 14859-14878, https://doi.org/10.5194/acp-22-14859-2022, 2022.

Zeng, X., Han, M., Ren, G., Liu, G., Wang, X., Du, K., Zhang, X., and Lin, H.: A comprehensive investigation on source apportionment and multi-directional regional transport of volatile organic compounds and ozone in urban Zhengzhou, Chemosphere, 334, 139001, https://doi.org/10.1016/j.chemosphere.2023.139001, 2023.

Zhang, D., He, B., Yuan, M., Yu, S., Yin, S., and Zhang, R.: Characteristics, sources and health risks assessment of VOCs in Zhengzhou, China during haze pollution season, J. Environ. Sci., 108, 44-57, https://doi.org/10.1016/j.jes.2021.01.035, 2021.

Zhang, H., Wang, Y., Hu, J., Ying, Q., and Hu, X. M.: Relationships between meteorological parameters and criteria air pollutants in three megacities in China, Environ. Res., 140, 242-254, https://doi.org/10.1016/j.envres.2015.04.004, 2015.

Zhang, Z., Sun, Y., and Li, J.: Characteristics and sources of VOCs in a coastal city in eastern China and the implications in secondary organic aerosol and $O_3$ formation, Sci. Total. Environ., 887, 164117, https://doi.org/10.1016/j.scitotenv.2023.164117, 2023.

Zhao, Y., Chen, L., Li, K., Han, L., Zhang, X., Wu, X., Gao, X., Azzi, M., and Cen, K.: Atmospheric ozone chemistry and control strategies in Hangzhou, China: Application of a 0-D box model, Atmos. Res., 246, https://doi.org/10.1016/j.atmosres.2020.105109, 2020.

Zong, R., Yang, X., Wen, L., Xu, C., Zhu, Y., Chen, T., Yao, L., Wang, L., Zhang, J., Yang, L., Wang, X., Shao, M., Zhu, T., Xue, L., and Wang, W.: Strong ozone production at a rural site in the North China Plain: Mixed effects of urban plumes and biogenic emissions, J. Environ. Sci., 71, 261-270, https://doi.org/10.1016/j.jes.2018.05.003, 2018.

Zuo, H., Jiang, Y., Yuan, J., Wang, Z., Zhang, P., Guo, C., Wang, Z., Chen, Y., Wen, Q., Wei, Y., and Li, X.: Pollution characteristics and source differences of VOCs before and after COVID-19 in Beijing, Sci. Total. Environ., 907, https://doi.org/10.1016/j.scitotenv.2023.167694, 2024.

---

## Editor Decision (ED1)

Corrections/amendments for Zhang et al. "Characteristics and sources of VOCs and the O3-NOx-NMVOCs relationships in Zhengzhou, China"

**Line numbers refer to the track-changes document egusphere-2023-2835-ATC1**

10. "...important precursors of ozone (O3) generation" add "under conditions of sufficient nitrogen oxides"

19-20. The sentence "An observation-based mode was applied ..." is unclear. I suggest "We explore observations of the $O_3$-precursors relationship and propose observation-oriented $O_3$ control strategies."

22. Change "in anthropogenic" to "in the anthropogenic"

44. "NMVOCs concentration varies" should be "NMVOC concentrations vary" (change every occurrence).

90. The more usual symbol for percentile is "%ile", so "8H-90%ile"

114-117. "1 meter" is incredibly close. Please say something more specific

166ff. Section 2.5 is still rather obscure. Please provide the general differential equation being solved by the model (e.g., dX/dt = P – L(X), where X is the chemical species, P is all the production terms and L(X) all the loss terms). Since the model is described as being independent of emissions and (incorrectly – see line 190) meteorological parameters, I imagine it is being used to calculate a steady-state solution appropriate for every hour (cf. ). Is this correct?

168. Change "employed to estimate the effect of changes of what in O3 precursors" to something like "employed to estimate the effect of changes in precursors on $O_3$"

169-170 Change "a good mix" to "a well-mixed atmosphere", if this is what is meant.

190-191. You mean that the concentration of NO was held constant? This seems like a very questionable assumption and would be equivalent to adding a source of NOx to the model.

194. The model will use time steps much smaller than 1 hour. Do you mean the model is stepped forward for 1 hour of simulation?

203. Do you mean that the $O_3$ is initialised to zero? That seems unlikely. I think you have an initial value which is allowed to evolve for the duration of the model run. Since $O_3$ evolves relatively slowly, it may be quite possible to have a good IOA without the model working well.

206. Without a better description of the model set up, you cannot claim that it is reliable.

262. "average" or "mean", not both.

303-306. The diurnal pattern in ozone must be a combination of mixing (especially as the nocturnal boundary layer breaks up) and chemistry, so it is not correct to ascribe high ozone mixing ratios to in-situ production alone. These sentences should be modified.

316. The reference for this sentence is not appropriate. Please cite one or more of the foundational papers on isoprene production from plants from the 1980s.

320-321. Couldn't the bimodal shape also come from reaction with OH, since OH peaks in the middle of the day? Your model should be able to calculate this.

346-350. These sentences are not consistent. You argue that E/X gives information about photochemical age, not source differences.

Fig 4. "ratio" not "ration"

444-445. Pease correct: "Among AVOCs, aromatics had the highest RIR value, followed by alkanes and aromatics."

536. Change "The summer O3 pollution has always been an important environmental issue in Zhengzhou" to something like "Summer $O_3$ pollution remains an important environmental issue in Zhengzhou."

---

## Author Response (AR2)

A letter of Response to Reviewers

**Title:** Characteristics and sources of NMVOCs and the $O_3$-$NO_x$-NMVOCs relationships in Zhengzhou, China

We thank the reviewer for reading our manuscript and providing valuable comments and suggestions. We have carefully considered all comments and revised the manuscript accordingly. We believe that we have adequately addressed your comments. To facilitate your review, the comments are in black, and our responses are in blue. Major changes made in response to these comments are highlighted in yellow in the revised manuscript, while our minor changes are marked in red font. Please note that the line numbers referred to below correspond to the corrected version.

**Referee comments**

10     "…important precursors of ozone ($O_3$) generation" add "under conditions of sufficient nitrogen oxides"

     Response: Thank you for your valuable feedback.

     We have made the suggested modification to our manuscript. The original sentence on line 10, "Nonmethane volatile organic compounds (NMVOCs) are important precursors for the formation of ozone ($O_3$)," has been revised to: "Nonmethane volatile organic compounds (NMVOCs) are important precursors of ozone ($O_3$) formation under conditions of sufficient nitrogen oxides." (Line 10-11)

19-20    The sentence "An observation-based mode was applied …" is unclear. I suggest "We explore observations of the $O_3$-precursors

relationship and propose observation-oriented $O_3$ control strategies."

**Response:** Thank you for your valuable feedback.

We have revised the sentence "An observation-based modeling approach was employed to investigate the $O_3$-precursors relationship." to "We explore observations of the $O_3$-precursors relationship and propose observation-oriented $O_3$ control strategies." to improve clarity. (Line 19-20)

22    Change "in anthropogenic" to "in the anthropogenic"

**Response:** Sorry for the mistake.

We have made the correction by changing "in anthropogenic" to "in the anthropogenic". (Line 21)

44    "NMVOCs concentration varies" should be "NMVOC concentrations vary" (change every occurrence).

**Response:** Sorry for the mistake.

We have made the correction by changing "NMVOCs concentration varies" to "NMVOC concentrations vary" for every occurrence in the manuscript. (Line 43)

90    The more usual symbol for percentile is "%ile", so "8H-90%ile".

**Response:** Thanks for the suggestion.

We have made the correction by changing "($O_3$-8H-90per)" to "($O_3$-8H-90%)." (Line 88-89)

114-117 "1 meter" is incredibly close. Please say something more specific?

**Response:** Thank you for your valuable feedback.

Regarding the mention of "Within a radius of 1 meter" in manuscript, it was indeed an error on my part. I intended to refer to a radius of 1 kilometer, not 1 meter. We have conducted field investigations, and we have corrected this mistake. (Line 114-115)

166 Section 2.5 is still rather obscure. Please provide the general differential equation being solved by the model (e.g., dX/dt = P – L(X), where X is the chemical species, P is all the production terms and L(X) all the loss terms). Since the model is described as being independent of emissions and (incorrectly – see line 190) meteorological parameters, I imagine it is being used to calculate a steady-state solution appropriate for every hour (cf. ). Is this correct?

MCM. Was it used to calculate other factors in your model?

**Response:** Thanks for the comments. We apologize for any obscurity.

1. The OBM used in this study iteratively solves a set of ordinary

differential equations (ODEs) that describe the evolution of species concentrations over time. For species with observation concentrations (normally constituted by primary NMVOCs and NO$_x$), the horizontal convection and emission are normally significant. In a zero-dimensional model, those processes are lumped into R$_{other}$ term. Within each iteration, R$_{other}$ is determined by the Eq (6):

$$R_{other} = \left(\frac{\partial C_i}{\partial t}\right)_{obs} - \left[P_i - L_i C_i - \frac{1}{H}v_d C_i - \frac{1}{H}\frac{dH}{dt}\left(C_i - C_{i,bg}\right) + R_{aero,i} + R_{aq,i}\right]$$

(6)

Where $P_i$ and $L_i C_i$ represents total the represents all the production and loss rate, respectively; $\frac{1}{H}v_d C_i$ represents the sum of mixing and deposition rates; $\frac{1}{H}\frac{dH}{dt}\left(C_i - C_{i,bg}\right)$ accounts for the mass exchange rate with background atmosphere; $R_{aero,i}$ and $R_{aq,i}$ are the rate of aerosol and aqueous processes, respectively; $\left(\frac{\partial C_i}{\partial t}\right)_{obs}$ is the real rate of change in concentration which is interpolated from hourly observed data points.

With the value of R$_{other}$ term explicitly determined from Eq (6), the concentrations of all species are then predicted by integrating the governing equation (7):

$$\frac{\partial C_i}{\partial t} = P_i - L_i C_i - \frac{1}{H}v_d C_i - \frac{1}{H}\frac{dH}{dt}\left(C_i - C_{i,bg}\right) + R_{aero,i} +$$
$$R_{aq,i} + R_{other}$$

(7)

New iterations start with updated $R_{other}$ values based on the concentrations predicted from the previous step, until converged solution is obtained.

Additionally, we have included the above content in the manuscript. (Line 172-186)

2. Although the input data for our study has a resolution of one hour, the actual time step for solving the model is less than 100 seconds, such small time-step guarantees the accuracy/reliability of the model solution (i.e. the results are independent from the integration time-step size). The model performs interpolation on the observed values. The statement in the manuscript has been changed to: "Briefly, OBM is a zero-dimensional model that assumes a well-mixed atmosphere, and combined with atmospheric chemical mechanisms, simulates the $O_3$ production rate and the corresponding $O_3$ concentration at a given time." (Line 168-171)

3. The Master Chemical Mechanism (MCM) is a near-explicit chemical mechanism used to describe gas-phase chemistry. Its latest version, MCMv3.3.1, includes over 5800 species and 17,000 reactions. The MCM was not used to calculate other factors in our model. Instead, it served as a reference mechanism

for the chemical reactions included in our model.

168    Change "employed to estimate the effect of changes of what in $O_3$ precursors" to something like "employed to estimate the effect of changes in precursors on $O_3$".

**Response:** Thanks for the suggestion.

We have revised the sentence "OBM based on the Master Chemical Mechanism (MCM v3.3.1; https://mcm.york.ac.uk/MCM/) was employed to estimate the effect of changes of what in $O_3$ precursors." to "OBM based on the Master Chemical Mechanism (MCM v3.3.1; https://mcm.york.ac.uk/MCM/) was employed to estimate the effect of changes in precursors on $O_3$." to improve clarity. (Line 166-167)

169-170    Change "a good mix" to "a well-mixed atmosphere", if this is what is meant.

**Response:** Thanks for the suggestion.

We have made the correction by changing "a good mix" to "a well-mixed atmosphere." (Line 169)

190-191    You mean that the concentration of NO was held constant? This seems like a very questionable assumption and would be equivalent to adding a source of $NO_3$ to the model.

**Response:** Thank you for your valuable feedback.

In the model, concentration of NO is not held constant, as it is calculated by solving the Eq (6-7). The solution procedure is described in the response to above comment. Regarding your comment about the concentration of NO being held constant.

194    The model will use time steps much smaller than 1 hour. Do you mean the model is stepped forward for 1 hour of simulation?

**Response:** Thank you for your insightful comments.

We apologize for the inaccuracies in our manuscript, which have been removed. Regarding the time steps, we would like to clarify that the model's time steps are less than 100 seconds, not 1 hour. However, the output time step of the model is indeed 1 hour. We have addressed the specific calculations and settings of the OBM in our previous responses.

203   Do you mean that the $O_3$ is initialised to zero? That seems unlikely. I think you have an initial value which is allowed to evolve for the duration of the model run. Since $O_3$ evolves relatively slowly, it may be quite possible to have a good IOA without the model working well.

**Response:** We apologize for the confusion caused by our inappropriate description.

Regarding your query about the initialization of $O_3$, we would like to clarify that the initial concentration of $O_3$ in our model is not zero. Instead, we set the initial $O_3$ concentration to 0.04 ppmv, which represents the background ozone concentration in clean air. Additionally, the meaning in the manuscript is that we do not input the observed $O_3$ concentration into the OBM, but rather simulate the $O_3$ concentration using the OBM.

206     Without a better description of the model set up, you cannot claim that it is reliable.

**Response:** Thanks for the suggestion.

We have provided a detailed description of the OBM's simulation methodology in our response above. Additionally, we have added a table in the supplementary materials that outlines the specific settings of the model (Table S2). We believe these additions enhance the understanding of the model setup and its reliability.

262     "average" or "mean", not both.

**Response:** Sorry for this mistake.

We have revised the sentence as follows: "During the entire period, the concentration of TNMVOCs varied from 10.0 to 60.0 ppbv, with an average concentration of $23.0 \pm 8.0$ ppbv." (Line 275)

303-306   The diurnal pattern in ozone must be a combination of mixing (especially as the nocturnal boundary layer breaks up) and chemistry, so it is not correct to ascribe high ozone mixing ratios to in-situ production alone. These sentences should be modified.

**Response:** Thank you for your comments.

We have referenced relevant literature. Based on this, we have revised the relevant section. The updated text now reads:

"This diurnal pattern is influenced by strong photochemical reactivity, boundary layer processes, and meteorological parameters. Higher $O_3$ production during the day indicates significant contributions from both photochemical reactions and atmospheric mixing processes." (Line 318-321)

316   The reference for this sentence is not appropriate. Please cite one or more of the foundational papers on isoprene production from plants from the 1980s.

**Response:** Thank you for your feedback.

We apologize for the oversight regarding the reference for the sentence on isoprene production from plants. We have revised the manuscript to include the following references:

1. Guenther, A. B., Zimmerman, P. R., Harley, P. C., Monson, R.

K., and Fall, R.: Isoprene and monoterpene emission rate variability: Model evaluations and sensitivity analyses, J. Geophys. Res-Atmos., 98, 12609-12617, https://doi.org/https://doi.org/10.1029/93JD00527, 1993.

2. Sharkey, T. D., Singsaas, E. L., Vanderveer, P. J., and Geron, C.: Field measurements of isoprene emission from trees in response to temperature and light, Tree. Physiol., 16, 649-654, https://doi.org/10.1093/treephys/16.7.649, 1996.

We hope these references meet the standards you suggested. (Line 335-336)

320-321    Couldn't the bimodal shape also come from reaction with OH, since OH peaks in the middle of the day? Your model should be able to calculate this.

**Response:** Thank you for your valuable feedback.

To address your comment, we have now included an analysis of the OH radical concentrations simulated by our model. The daily variation of OH radical indeed shows a peak around noon, which could contribute to the observed decrease in isoprene concentrations. We have included a plot of the daily variation of OH radical in the revised manuscript (Fig. S5) and updated the text to reflect this additional consideration.

Additionally, we have added the following sentence to further explain the influence of OH radical on the bimodal shape of isoprene concentration: "Additionally, the concentration of OH radicals peaks at noon (Fig. S5), leading to the rapid oxidation of isoprene by OH radicals, which further contributes to the observed bimodal pattern (Paulot et al., 2009)." (Line 337-339)

We believe these additions enhance the understanding of the factors influencing the bimodal shape of isoprene concentration. Thank you again for your constructive comments.

[Figure]

Fig. S5. Model-simulated daytime average diurnal variations in OH concentrations.

References

Paulot, F., Crounse, J. D., Kjaergaard, H. G., Kürten, A., St Clair, J. M., Seinfeld, J. H., and Wennberg, P. O.: Unexpected Epoxide Formation in the Gas-Phase Photooxidation of Isoprene, Science, 325, 730-733, https://doi.org/10.1126/science.1172910, 2009.

346-350    These sentences are not consistent. You argue that E/X gives information about photochemical age, not source differences.   Fig 4. "ratio" not "ration"

**Response:** Thank you for your feedback. We appreciate your attention to detail.

We have revised the first sentence of the manuscript as you suggested. It now reads: "Since m/p-xylene and ethylbenzene share a common source but differ in their OH radical reaction rate constants, the E/X ratio can be used to understand source characteristics." (Line 363-364)

Additionally, we have corrected the typographical error in Figure 4, changing "ration" to "ratio".

[Figure]

**Figure 4.** Source profiles and contributions of NMVOCs during the observation period.

444-445 Please correct: "Among AVOCs, aromatics had the highest RIR value, followed by alkanes and aromatics."

**Response:** Thank you for pointing out the error in the sentence. I have corrected it as follows:

The original sentence: "Among AVOCs, aromatics had the highest

RIR value, followed by alkanes and aromatics." has been corrected to: "Among AVOCs, aromatics had the highest RIR value, followed by alkenes and alkanes." (Line 461-462)

536 Change "The summer $O_3$ pollution has always been an important environmental issue in Zhengzhou" to something like "Summer $O_3$ pollution remains an important environmental issue in Zhengzhou."

**Response:** Thank you very much for your valuable feedback. I have revised the sentence as per your suggestion. The original sentence: "The summer $O_3$ pollution has always been an important environmental issue in Zhengzhou" has been changed to: "Summer $O_3$ pollution remains an important environmental issue in Zhengzhou." (Line 554)